# IGFBP5 is an ROR1 ligand promoting glioblastoma invasion via ROR1/HER2-CREB signaling axis

Weiwei Lin[1,2,3,4,8], Rui Niu[1,8], Seong-Min Park[2,5,8], Yan Zou [1,6,8], Sung Soo Kim [2,8], Xue Xia[1], Songge Xing[1], Qingshan Yang[1], Xinhong Sun[1], Zheng Yuan[1], Shuchang Zhou[1], Dongya Zhang[1], Hyung Joon Kwon[7], Saewhan Park[2], Chan Il Kim [2], Harim Koo[2], Yang Liu[1], Haigang Wu[1], Meng Zheng[1], Heon Yoo[2,3], Bingyang Shi [1,6] ✉, Jong Bae Park [1,2,3] ✉ & Jinlong Yin [1,2] ✉

Diffuse infiltration is the main reason for therapeutic resistance and recurrence in glioblastoma (GBM). However, potential targeted therapies for GBM stem-like cell (GSC) which is responsible for GBM invasion are limited. Herein, we report Insulin-like Growth Factor-Binding Protein 5 (IGFBP5) is a ligand for Receptor tyrosine kinase like Orphan Receptor 1 (ROR1), as a promising target for GSC invasion. Using a GSC-derived brain tumor model, GSCs were characterized into invasive or non-invasive subtypes, and RNA sequencing analysis revealed that IGFBP5 was differentially expressed between these two subtypes. GSC invasion capacity was inhibited by IGFBP5 knockdown and enhanced by IGFBP5 overexpression both in vitro and in vivo, particularly in a patient-derived xenograft model. IGFBP5 binds to ROR1 and facilitates ROR1/HER2 heterodimer formation, followed by inducing CREB-mediated *ETV5* and *FBXW9* expression, thereby promoting GSC invasion and tumorigenesis. Importantly, using a tumor-specific targeting and penetrating nanocapsule-mediated delivery of CRISPR/Cas9-based IGFBP5 gene editing significantly suppressed GSC invasion and downstream gene expression, and prolonged the survival of orthotopic tumor-bearing mice. Collectively, our data reveal that IGFBP5-ROR1/HER2-CREB signaling axis as a potential GBM therapeutic target.

Patients with glioblastoma (GBM), the most frequent and aggressive malignant primary brain tumor in adults, have an average overall survival time of merely 14 months[1,2]. The infiltrative nature of GBM enables neoplastic spread and migration into adjacent brain tissue, which makes it very challenging, or even impossible, for all multimodality treatments to achieve complete excision, inevitably leading to recurrence[3–5]. Therefore, elucidating the mechanisms regulating GBM invasion is key for the development of effective therapeutic strategies.

[1]Henan-Macquarie University Joint Centre for Biomedical Innovation, School of Life Sciences, Henan University, Kaifeng, Henan 475004, China. [2]Department of Cancer Biomedical Science, Graduate School of Cancer Science and Policy, National Cancer Center, Goyang, Gyeonggi 10408, Republic of Korea. [3]Research Institute, National Cancer Center, Goyang, Gyeonggi 10408, Republic of Korea. [4]Department of Life Science, Ewha Womans University, Seoul 03760, Republic of Korea. [5]Personalized Genomic Medicine Research Center, KRIBB, Daejeon 34141, Republic of Korea. [6]Centre for Motor Neuron Disease Research, Macquarie Medical School, Faculty of Medicine & Health Sciences, Macquarie University, Sydney, NSW 2109, Australia. [7]Department of Cancer Control and Population Health, Graduate School of Cancer Science and Policy, National Cancer Center, Goyang, Gyeonggi 10408, Republic of Korea. [8]These authors contributed equally: Weiwei Lin, Rui Niu, Seong-Min Park, Yan Zou, Sung Soo Kim. ✉e-mail: bs@henu.edu.cn; jbp@ncc.re.kr; jlyin@henu.edu.cn

GBM stem-like cells (GSCs), also known as GBM-initiating cells, are responsible for disease progression, therapeutic resistance, and tumor recurrence[6–8]. These cells share stem cell markers with neural stem cells, such as Nestin and CD133, as well as their capacity of self-renewal and differentiation[5,9,10]. In contrast to differentiated tumor cells, GSCs can efficiently propagate tumors in orthotopic xenograft mice[6,9,11]. Moreover, GSC-derived orthotopic xenografts closely mirror the phenotype and genotype of primary tumors in patients[11]. In this xenograft mouse model, both invasive and localized orthotopic tumors are established from a series of human GSCs[12–16]. The exhibition of various degrees of invasive model promptly resembles the brain pathological features of patients with GBM, is ideal for the thorough investigation of the molecular mechanism underpinning GBM invasion.

Recently, a comprehensive longitudinal study of GBM tumors classified GBMs into proneural (PN), classical (CL), and mesenchymal (MES) subtypes, and a similar recapitulation can be made for GSCs[13,17]. In particular, according to the Ivy GAP transcriptome data (Ivy Glioblastoma Atlas Project), the PN subtype is predominantly documented in the leading edge of the tumor, as compared to the MES subtype which largely exists in the pseudopalisading region or the tumor core[13,18]. In addition, a recent phenotypic study of patients with GBM revealed that PN- and MES-subtyped GSCs were localized to the invasive edge and core of the tumor, respectively[19]. Nevertheless, the detailed molecular characteristics of these invasive and non-invasive GSCs remain largely unexplored.

In this study, to identify the master regulators of invasive GSCs and to better understand GBM invasion, we perform RNA-seq analysis between invasive and non-invasive GSCs, which are divided according to the hematoxylin and eosin (H&E) staining of GSC-derived orthotopic xenograft models. We demonstrate that IGFBP5 regulates GSC invasion serving as a ligand for ROR1, which triggers formation of ROR1/HER2 (Human Epidermal growth factor Receptor 2) heterodimer to enhance CREB (cAMP Response Element Binding protein) oncogenic signaling. Moreover, both lentivirus-mediated IGFBP5 knockdown and nanocapsule-mediated Cas9/sgIGFBP5 delivery significantly compromise GSC invasion and extend the survival of orthotopic tumor-bearing mice. Collectively, our findings highlight the critical role of IGFBP5 in enhancing GSC invasion and providing a promising therapeutic approach for diffuse GBM.

## Results

### IGFBP5 expression is associated with GSCs invasion and patient survival in glioma

To study GBM invasion, we classified GSCs into invasive or non-invasive by phenotypic characterization using orthotopic xenograft mouse models. In detail, 448 and X01 GSCs formed invasive tumors that spread into the brain through the corpus callosum, whereas 83 and 131 GSCs exhibited strong localization with a clear boundary, indicating non-invasive localization (Fig. 1a). In addition, in vitro transwell invasion assays demonstrated consistent results regarding the behavior of the two GSC subtypes: 448 and X01 GSCs exhibited significantly greater invasive ability than the non-invasive GSCs (83 and 131 GSCs; $P < 0.01$; Fig. 1b).

High-throughput RNA sequencing (RNA-seq) was performed to further elucidate the molecular mechanism underlying the different invasive abilities of the two GSC types. Differentially expressed genes (DEGs) between the invasive and non-invasive GSCs were identified using a criterion combined with a minimum four-fold change in the number of mapped reads per kilobase of transcript per million reads mapped (RPKM) and FDR < 0.1 in the DESeq2 and edgeR of TCC package[20] (Fig. 1c). Through Kyoto Encyclopedia of Genes and Genomes (KEGG) analysis, the upregulated DEGs were identified as significantly associated with GBM cell-related pathways, including GABAergic synapses, signaling pathways regulating pluripotency, and

Wnt signaling (Supplementary Fig. 1a). No KEGG terms related to well-known GBM cell-related pathways were found to be associated with the downregulated DEGs (Supplementary Fig. 1a). Further analysis of the identified DEGs was performed using gene set enrichment analysis (GSEA) (Supplementary Fig. 1b–d), and the results showed that the invasive-type GSCs exhibited high expression of proneural or classic GBM marker genes, whereas the non-invasive GSCs presented high mesenchymal GBM marker gene expressions, which is consistent with the results of previous studies[19]. A total of 36 candidate DEGs regulating GSC invasion were identified from our established 292 DEGs based on corresponding clinical information for 2057 genes obtained from The Cancer Genome Atlas (TCGA), and these DEGs were significantly associated with poor survival (hazard ratio (HR) > 1, $P < 0.01$, Cox proportional hazards analysis) (Fig. 1c). Among these, IGFBP5 showed the highest differential expression between invasive and non-invasive GSCs (Fig. 1c, d). This result was validated in 448, X01, 131, and 83 GSCs at both the RNA (quantitative polymerase chain reaction, RT-qPCR; Fig. 1e) and protein (enzyme-linked immunosorbent assay, ELISA; Fig. 1f) levels. Therefore, IGFBP5 was selected as the top candidate for further study.

IGFBP5 is a secreted protein of the IGFBP family that mainly regulates the specific binding of insulin-like growth factors (IGFs) to IGF receptors[21–24], which showed a high correlation with the migration of breast cancer[25,26]. This protein plays diverse roles in cancer progression, i.e., it serves as a tumor suppressor in melanoma[27], cervical carcinoma[28] and ovarian cancer[21,22] and as an oncogene in prostate[29], breast[23], and pancreatic[30] cancers through either IGF-dependent or IGF-independent signaling pathways. Though a small number of studies have observed high IGFBP5 expression in tissue samples from patients with GBM, the functional significance of IGFBP5 in GBM has yet to be thoroughly investigated. A comparison of IGFBP5 expression between normal and tumor tissues from patients with various types of cancer using data from the Gene Expression across Normal and Tumor Tissue (GENT; http://medical-genome.kribb.re.kr/GENT/) database[31] showed that IGFBP5 is typically expressed at lower levels in tumor tissues than in normal tissues (Supplementary Fig. 2a, b). However, the opposite pattern was found in brain cancer: IGFBP5 expression was higher in tumor tissues than in normal brain tissues (Supplementary Fig. 2a, b), which indicated a potential protumorigenic role of IGFBP5 in brain cancer. To confirm the potential oncogenic role of IGFBP5 in brain cancer, we analyzed public RNA-seq databases with corresponding clinical information, such as the TCGA low-grade glioma (LGG) and the TCGA high-grade GBM datasets, revealed that high IGFBP5 expression was significantly associated with the poor survival of patients with LGG and GBM (Fig. 1g, TCGA-LGG, $P = 0.000431$; Fig. 1h, TCGA-GBM, $P = 0.0031$; log-rank test).

Thus, IGFBP5 was identified as the top DEG in invasive GSCs and was found to be associated with unfavorable patient outcomes, which suggests that this protein might regulate GBM progression by promoting GSC invasion.

### IGFBP5 regulates GSC invasion and tumorigenesis

To determine whether IGFBP5 is important to GSC invasion, we silenced IGFBP5 in invasive GSCs (X01 and 448) with two different shRNAs, performed in vitro transwell invasion assay and generated an orthotopic mouse model (Fig. 2a–j). Successful knockdown of IGFBP5 was validated by RT-qPCR and ELISA (Fig. 2a, b, e, f). Transwell invasion assays revealed that both shIGFBP5 significantly decreased the invasion of invasive GSCs compared to a non-targeting control shRNA (shCtrl) (Fig. 2c, d, g, h). We then examined the effect of IGFBP5 depletion on GSC invasion and tumor progression by intracranial injection into mice with X01 GSCs transduced with shCtrl or shIGFBP5. Knocking down of IGFBP5 expression in X01 GSCs remarkably reduced the intracranial tumor volume (Fig. 2i) and significantly prolonged the survival of the mice (Fig. 2j). Interestingly, H&E staining indicated that

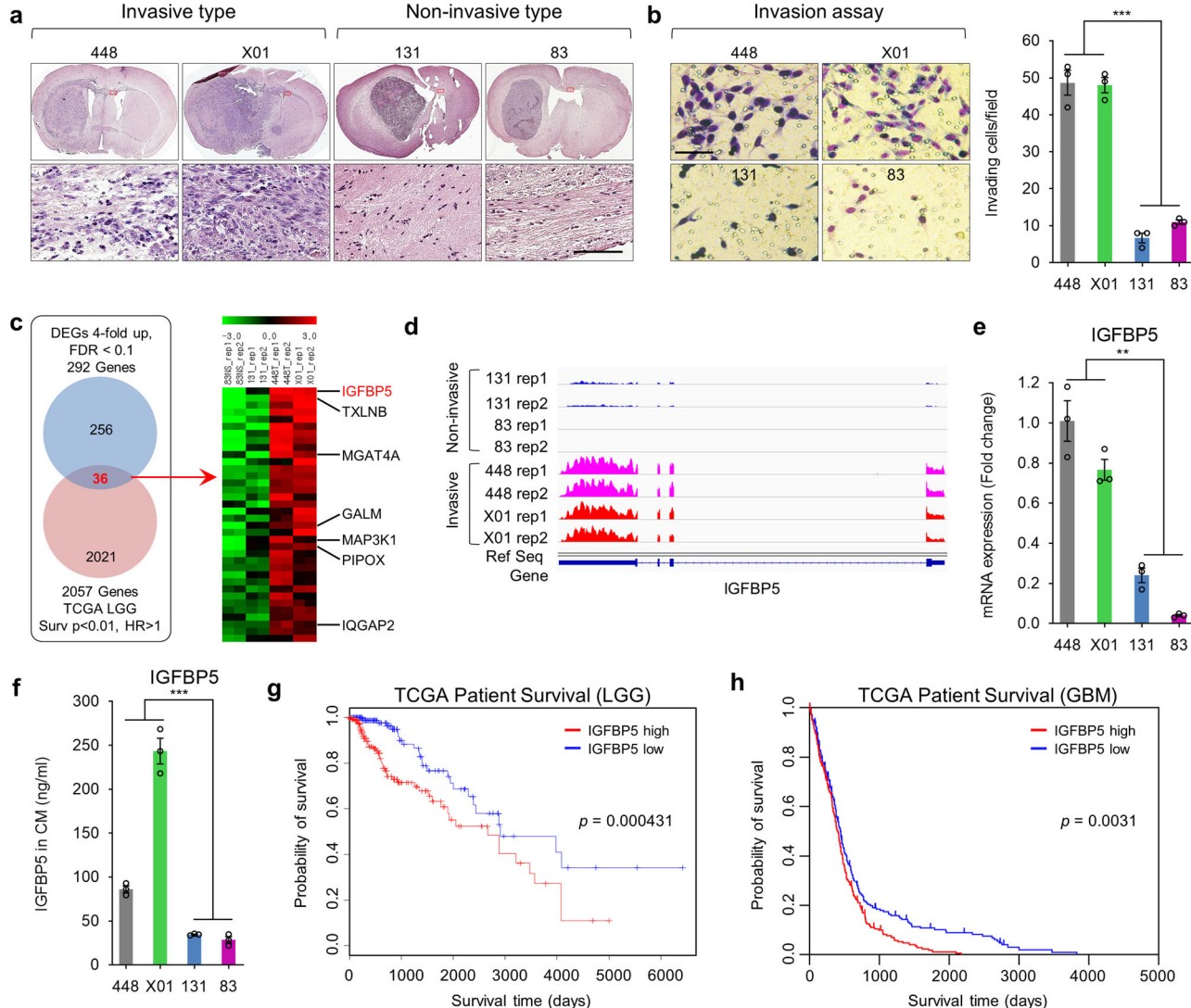

**Fig. 1 | IGFBP5 expression is associated with GSCs invasion and patient survival in glioma. a** Hematoxylin and eosin (H&E) staining of the whole brain from mice implanted with GSCs (448, X01, 131, and 83). Scale bar, 100 μm. **b** Invasion assays of four GSCs (448, X01, 131, and 83) after 48 h. Images are representative of three independent experiments (Scale bar, 100 μm; *n* = 3), the bar graph shows the average numbers of invasive cells. All error bars represent means ± standard error of the mean (SEM), ***P* < 0.001, two-tailed Student's *t*-test. **c** Venn diagram of Differentially-Expressed Genes (DEGs) identified in our RNA-seq analysis and the TCGA low-grade glioma (LGG) RNA-seq dataset (left) and the putative candidate genes associated with GSC invasion and the survival of patients with glioma (right). **d** IGFBP5 read distribution (RNA-seq) in the four types of GSCs. **e** RT-qPCR

validation of IGFBP5 expression in GSCs. Data are represented as mean ± SEM (*n* = 3 independent experiments), ***P* < 0.01, two-tailed Student's *t*-test. **f** ELISA analysis of secreted IGFBP5 protein in conditioned media (CM) from GSCs cultured for 3 days. Data are presented as mean ± SEM (*n* = 3 independent experiments), ****P* < 0.001, two-tailed Student's *t*-test. **g** Kaplan–Meier survival curves for LGG patients with high or low IGFBP5 expression based on the median expression in the TCGA dataset. *P* = 0.000431, log-rank test. **h** Kaplan–Meier survival curves for GBM patients with high or low IGFBP5 expression based on the median expression in the TCGA dataset. *P* = 0.0031, log-rank test. Source data and exact *P* values are provided as the Source Data file.

shCtrl-GSCs exhibited enhanced cell proliferation at its injected hemisphere (Green box, Fig. 2i), their invasion was also observed deeply into the opposite hemisphere through corpus callosum when compared to either shIGFBP5 (Red box, Fig. 2i). These results highlighted that the depletion of IGFBP5 expression reduced the invasive ability of invasive GSCs both in vitro and in vivo.

Next, we modulated IGFBP5 expression in non-invasive GSCs (83 and 131) by treatment with recombinant IGFBP (rIGFBP5) protein and ectopic IGFBP5 overexpression. RT-qPCR and ELISA verified a successful ectopic overexpression of IGFBP5 (Fig. 3a, b, e, f). Both rIGFBP5 and ectopic IGFBP5 overexpression significantly increased the invasiveness of non-invasive GSCs (Fig. 3c, d, g, h and Supplementary Fig. 3a–d). In the orthotopic mouse model, ectopic IGFBP5 overexpression substantially increased the non-invasive 83 GSC tumor

volume (Fig. 3i) and significantly reduced the mice survival (Fig. 3j). Particularly, anti-GFP immunofluorescence staining (enhanced by red fluorescent secondary antibodies) indicated that the mice injected with overexpressed IGFBP5 in 83 GSCs showed readily invasion of a considerable number of GSC cells in the corpus callosum (Red box) (Fig. 3i). These results strongly indicated that IGFBP5 promotes GSC invasion both in vitro and in vivo.

## IGFBP5 binds to ROR1 and triggers ROR1/HER2-CREB signaling

To elucidate the molecular mechanism of IGFBP5 in promoting GSC invasion and tumorigenesis, we explored whether IGFBP5 triggers plasma membrane receptor phosphorylation and activation since the tyrosine kinase cascades are associated with oncogenesis. We performed an unbiased human phospho-receptor tyrosine kinase (RTK)

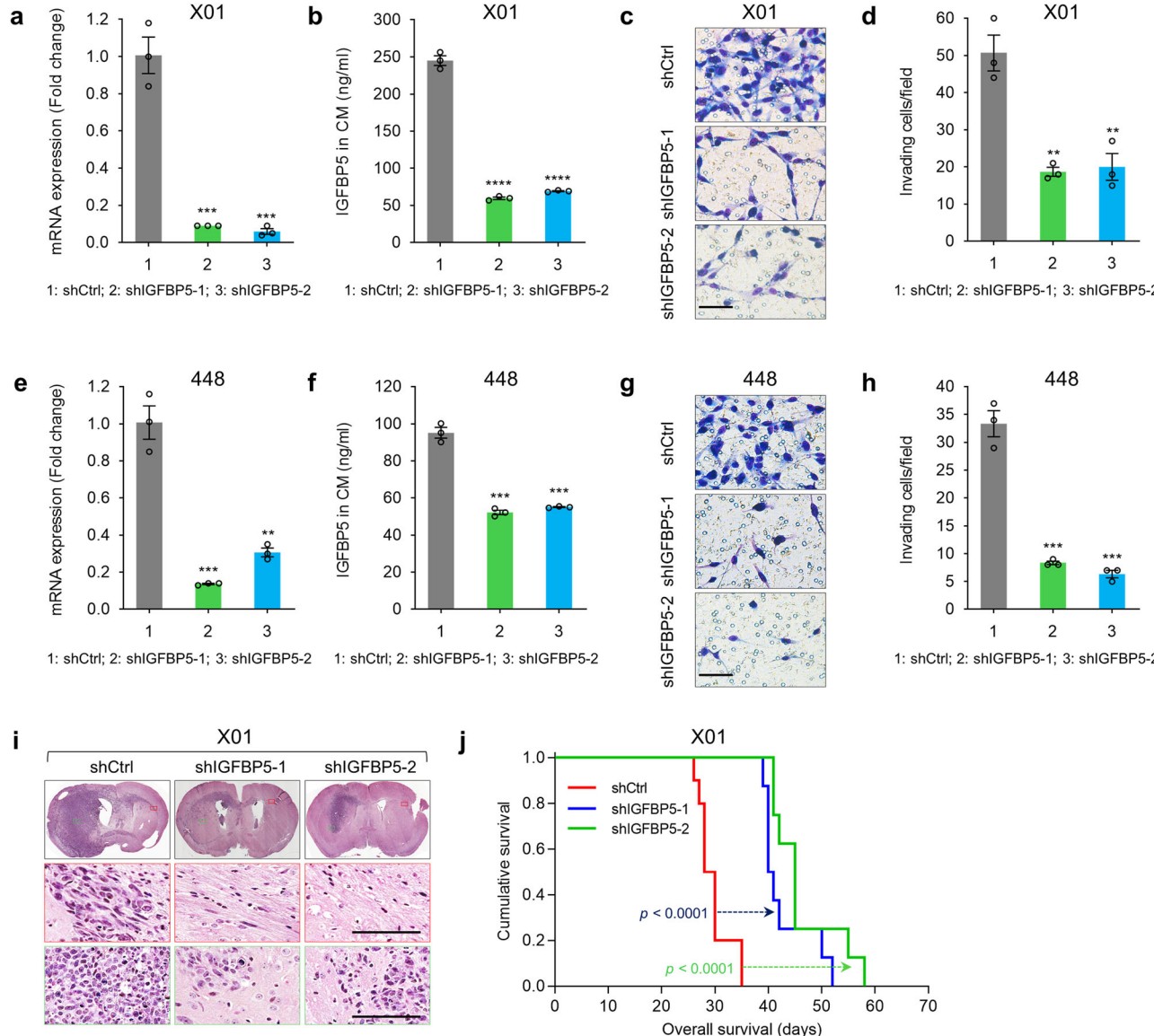

**Fig. 2 | Inhibition of IGFBP5 impairs GSCs invasion and tumorigenesis.**
**a** RT-qPCR analysis of IGFBP5 mRNA expression in X01 GSCs infected with lentivirus expressing shIGFBP5 or shCtrl. Data are presented as mean ± SEM ($n = 3$ independent experiments), two-tailed Student's $t$-test (left to right, $***P = 0.0008$, $***P = 0.0007$). **b** ELISA analysis of IGFBP5 in CM from X01 GSCs infected with shCtrl, shIGFBP5-1, or shIGFBP5-2 lentivirus. Data are presented as mean ± SEM ($n = 3$ independent experiments), two-tailed Student's $t$-test (left to right, $****P = 0.00001$, $****P = 0.00001$). **c, d** Invasion assays with X01 GSCs infected with shCtrl, shIGFBP5-1, or shIGFBP5-2 lentivirus. **c** Images taken after 48 h of invasion are representative of three independent experiments (scale bar, 100 μm; $n = 3$), and **d** the graph shows the mean number of invasive cells ± SEM, two-tailed Student's $t$-test (left to right, $**P = 0.003$, $**P = 0.007$). **e** RT-qPCR analysis of IGFBP5 mRNA expression in 448 GSCs infected with shCtrl, shIGFBP5-1, and shIGFBP5-2 lentivirus. Data are presented as mean ± SEM ($n = 3$ independent experiments), two-tailed Student's $t$-test

(left to right, $***P = 0.0006$, $**P = 0.002$). **f** ELISA analysis of IGFBP5 in CM from 448 GSCs infected with shCtrl, shIGFBP5-1, or shIGFBP5-2 lentivirus. Data are presented as mean ± SEM ($n = 3$ independent experiments), two-tailed Student's $t$-test (left to right, $***P = 0.0002$, $***P = 0.0002$). **g, h** Invasion assays using 448 GSCs infected with shCtrl, shIGFBP5-1, or shIGFBP5-2 lentivirus. **g** Images taken after 48 h of invasion are representative of three independent experiments (scale bar, 100 μm; $n = 3$), and **h** the graph shows the mean number of invasive cells ± SEM, two-tailed Student's $t$-test (left to right, $***P = 0.0004$, $***P = 0.0004$). **i** H&E staining of the whole brains of mice bearing orthotopic xenografts of X01 GSCs infected with shCtrl, shIGFBP5-1, or shIGFBP5-2 lentivirus. Scale bar, 100 μm. **j** Kaplan–Meier survival curves of mice implanted with X01 GSCs infected with shCtrl ($n = 10$), shIGFBP5-1 ($n = 8$), or shIGFBP5-2 ($n = 8$) lentivirus ($1 \times 10^4$ cells/mouse). $P < 0.0001$, log-rank test. Source data are provided as the Source Data file.

antibody array using 83 GSCs treated with human rIGFBP5 or vehicle. Treatment with human rIGFBP5 increased the phosphorylation levels of both HER2 and ROR1 (Fig. 4a and Supplementary Fig. 4a). Furthermore, well-recognized kinases and their substrates downstream of HER2 and ROR1 that are related with tumorigenesis, such as extracellular signal-regulated kinases (ERK1/2), checkpoint kinase 2 (Chk2), CREB, and heat shock protein 60 (HSP60), exhibited notable levels of phosphorylation in response to IGFBP5, as showed in the phosphokinase array analysis (Fig. 4b and Supplementary Fig. 4b). Interestingly,

CREB is associated with tumor progression, therapeutic resistance, and patient prognosis[32,33], and previous studies have shown that CREB activation contributes to HER2-mediated neoplastic cell growth[34] and ROR1-mediated breast cancer cell growth[33]. Thus, we oriented our focus on the ROR1/HER2-CREB signaling axis for further study. We validated these results with immunoblot assays, which showed that rIGFBP5 increased the phosphorylation of HER2, ROR1 and CREB in 83 and 131 GSCs (Fig. 4c and Supplementary Fig. 4c, d). Moreover, the phosphorylation of all three proteins was increased by the ectopic

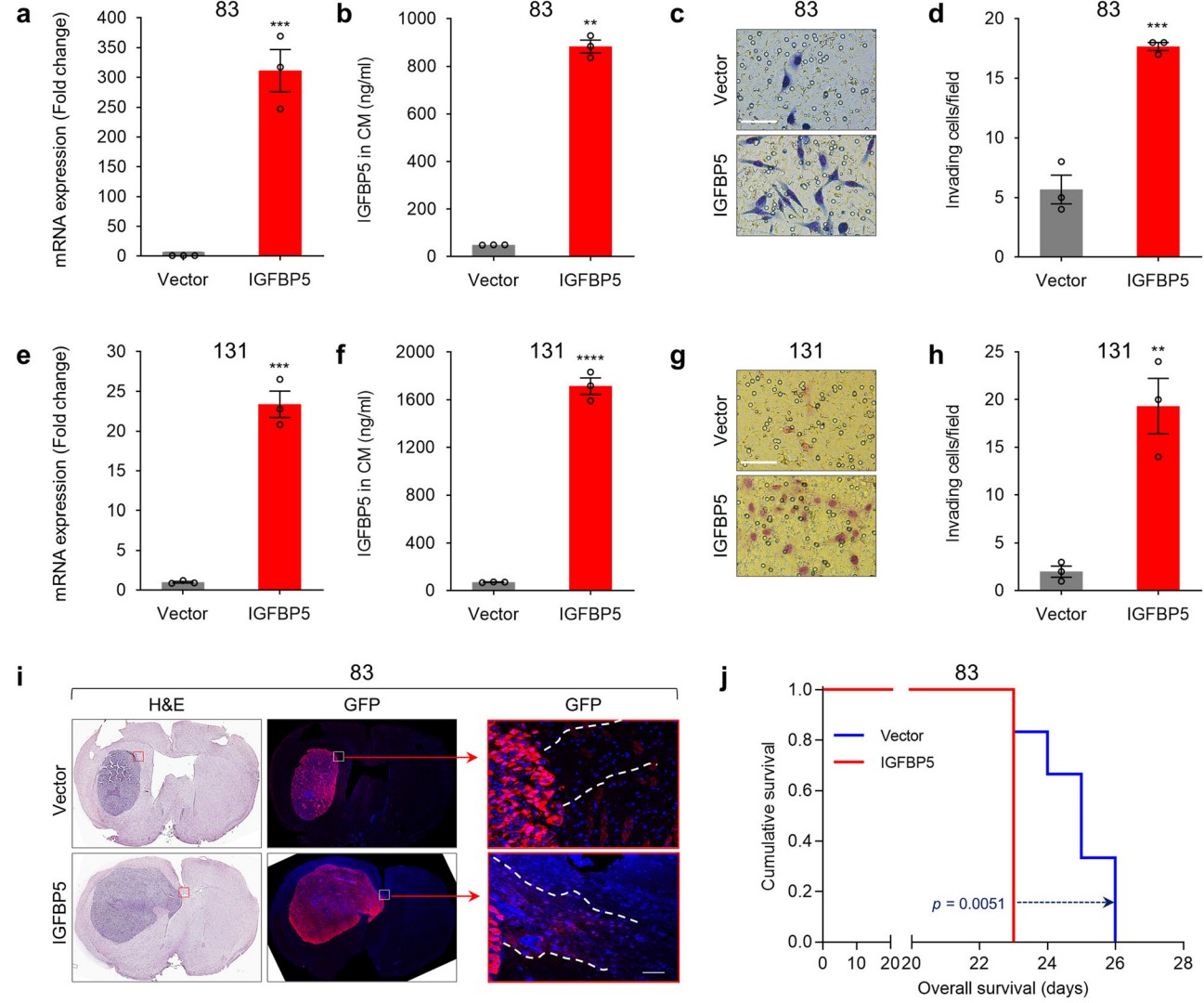

**Fig. 3 | Ectopic IGFBP5 overexpression enhances invasion and tumorigenesis of non-invasive GSCs. a** RT-qPCR analysis of IGFBP5 expression in 83 GSCs infected with IGFBP5 or vector control lentivirus. Data are presented as mean ± SEM ($n = 3$ independent experiments), two-tailed Student's $t$-test ($^{***}P = 0.0009$). **b** ELISA analysis of IGFBP5 in CM from 83 GSCs infected with IGFBP5 or vector control lentivirus. Data are presented as mean ± SEM ($n = 3$ independent experiments), two-tailed Student's $t$-test ($^{**}P = 0.003$). **c, d** Invasion assays with 83 GSCs infected with vector control or IGFBP5 lentivirus. **c** Images taken after 24 h of invasion are representative of three independent experiments (scale bar, 100 μm; $n = 3$), and **d** the graph shows the mean number of invasive cells ± SEM, two-tailed Student's $t$-test ($^{***}P = 0.0007$). **e** RT-qPCR analysis of IGFBP5 expression in 131 GSCs infected with IGFBP5 or vector control lentivirus. Data are presented as mean ± SEM ($n = 3$ independent experiments), two-tailed Student's $t$-test ($^{***}P = 0.0002$). **f** ELISA analysis of IGFBP5 in CM from 131 GSCs infected with IGFBP5 or vector control lentivirus. Data are presented as mean ± SEM ($n = 3$ independent experiments), two-tailed Student's $t$-test ($^{***}P = 0.00002$). **g, h** Invasion assays using 131 GSCs infected with vector control or IGFBP5 lentivirus. **g** Images taken after 24 h of invasion are representative of three independent experiments (scale bar, 100 μm; $n = 3$), and **h** the graph shows the mean number of invasive cells ± SEM, two-tailed Student's $t$-test test ($^{**}P = 0.004$). **i** H&E staining of the whole brains of mice bearing orthotopic xenografts of 83 GSCs infected with vector control or IGFBP5 lentivirus. Scale bar, 50 μm. **j** Kaplan–Meier survival curves of mice implanted with 83 GSCs infected with vector control or IGFBP5 lentivirus ($n = 6$ in each group, $1 \times 10^3$ cells/mouse). $P = 0.0051$, log-rank test. Source data are provided as the Source Data file.

overexpression of IGFBP5 (Fig. 4d) but substantially reduced by the knockdown of IGFBP5 (Fig. 4e).

The above results prompted us to ask whether IGFBP5 acts as a ROR1 or HER2 ligand to transduce downstream CREB signaling. To test this hypothesis, we performed co-immunoprecipitation (co-IP) using ROR1 or HER2 antibody in X01 and 448 cells. Interactions between endogenous ROR1 and HER2 were observed (Fig. 4f, g and Supplementary Fig. 4e, f). Moreover, we found that IGFBP5 interacted with ROR1 but not HER2 (Fig. 4f, g and Supplementary Fig. 4e, f), which suggest that IGFBP5 may serves as a ligand for ROR1 to transduce downstream signaling. To further confirm the above results, we conducted co-IP assay in 293 T cells by exogenous overexpression of Flag-tagged IGFBP5 and HA-tagged ROR1. Co-IP using Flag as the

precipitating antibody showed interactions only between IGFBP5 and ROR1 (Fig. 4h). However, ROR1 interacted with both IGFBP5 and endogenous HER2 when using HA as the precipitating antibody (Fig. 4i). Furthermore, to determine the direct interaction between IGFBP5 and ROR1, the binding affinity and the dissociation constant (Kd) were examined by microscale thermophoresis (MST) assay in vitro. The results showed a clear binding curve to IGFBP5-ROR1 with a Kd of 157.5 nM (Fig. 4j), but no binding affinity between IGFBP5 and HER2 (Supplementary Fig. 4g), suggesting that IGFBP5 serves as a ligand for ROR1 to trigger downstream signaling transduction.

To determine whether IGFBP5 transduces downstream signaling through ROR1 and HER2 heterodimer, we knocked down HER2 or ROR1 using two different shRNAs, and found that the inhibitions of

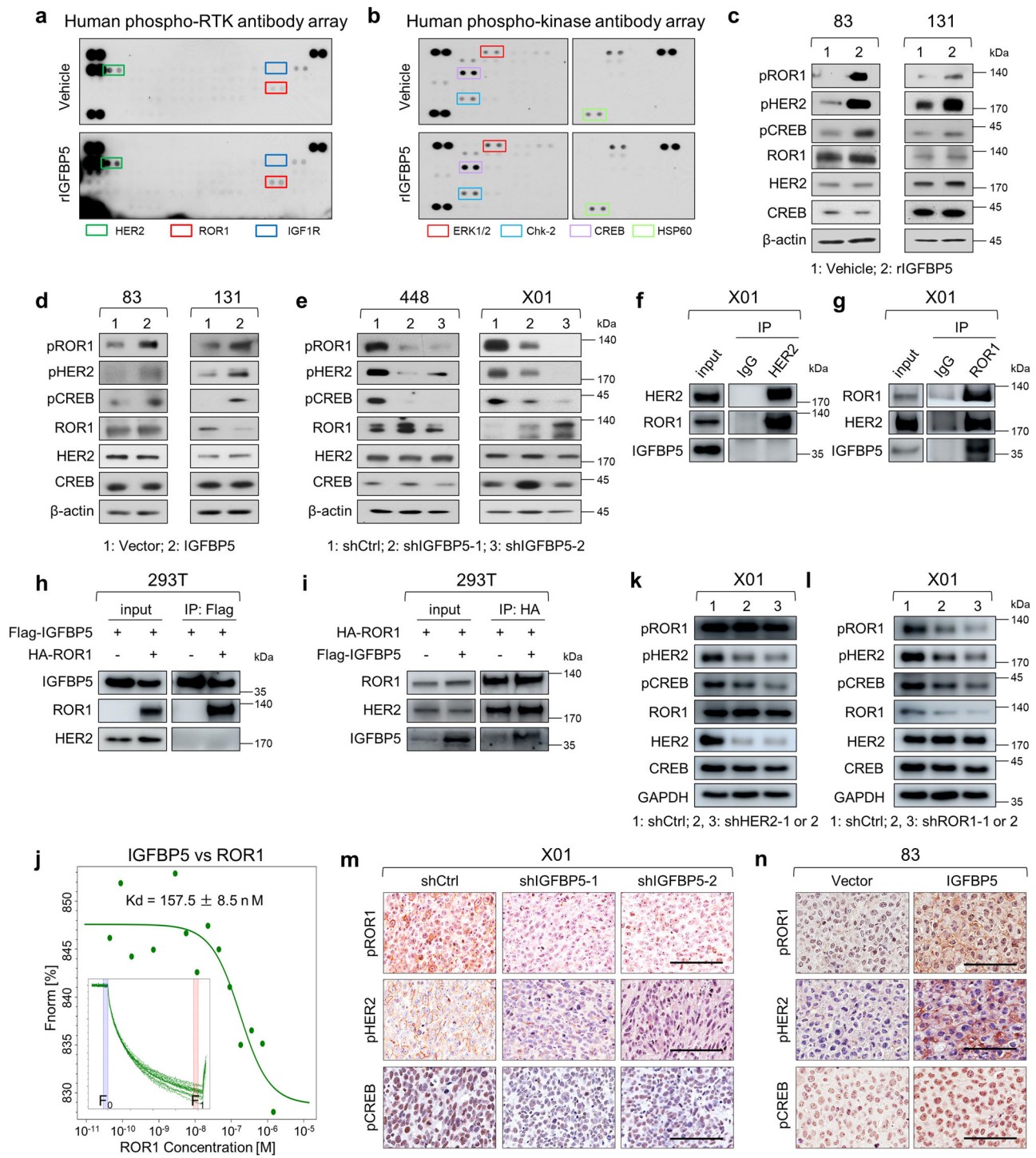

HER2 or ROR1 blocked CREB phosphorylation (Fig. 4k, l and Supplementary Fig. 4h, i) and reduced GSC invasiveness (Supplementary Fig. 4j–m). Moreover, our data suggested that HER2 knockdown did not alter ROR1 phosphorylation (Fig. 4k and Supplementary Fig. 4h), while ROR1 knockdown decreased HER2 phosphorylation (Fig. 4l and Supplementary Fig. 4i). Furthermore, we also modulated ROR1 activation by overexpressing ROR1 wild type (ROR1 WT) and kinase deletion type (ROR1 mut) in IGFBP5 knockdown X01 GSCs. The results showed that overexpression of ROR1 WT but not ROR1 mut rescued knockdown of IGFBP5 mediated repression of HER2 and CREB phosphorylation (Supplementary Fig. 4n), and GSCs invasion (Supplementary Fig. 4o, p), indicating that phosphorylation of ROR1 is a key step in IGFBP5-ROR1/HER2 signal transduction. To confirm the molecular

mechanism regulating this IGFBP5-mediated signaling axis, we examined the activation status of protein downstream of IGFBP5 in GSCs orthotopic mouse tissue based on immunohistochemistry. The phosphorylated ROR1, HER2, and CREB levels were lower in IGFBP5-knockdown tumors (Fig. 4m) and higher in IGFBP5-overexpressing tumors (Fig. 4n) than in control X01 or 83 tumors. Taken together, these results suggest that IGFBP5 serves as a ligand for ROR1 and functions through a ROR1/HER2-CREB downstream signaling axis.

## IGFBP5 activates transcription of *ETV5* and *FBXW9* via CREB

Since IGFBP5 increases the phosphorylation of the transcription factor (TF) CREB, we then studied which genes downstream of CREB are modulated by IGFBP5. Invasive X01 and 448 GSCs were subjected to

**Fig. 4 | IGFBP5 activates HER2 and ROR1 signaling. a** Human phospho-RTK array of 83 GSCs treated with 100 ng/ml recombinant IGFBP5 (rIGFBP5) or vehicle control for 6 h. **b** Human phospho-kinase array of 83 GSCs treated with 100 ng/ml rIGFBP5 or vehicle control for 6 h. **c–e** Immunoblot (IB) analysis of pROR1, pHER2, pCREB, ROR1, HER2, and CREB in **c** non-invasive GSCs (83 and 131) treated with rIGFBP5 (100 ng/ml) or vehicle control for 6 h, **d** 83 and 131 cells infected with vector control or IGFBP5 lentivirus and **e** 448 and X01 GSCs infected with shIGFBP5 or shCtrl lentivirus. β-actin was used as a loading control. **f, g** Co-IP of X01 cells with antibodies targeting HER2, ROR1 or normal IgG. **h** Co-IP analysis for the interaction of IGFBP5 and ROR1 in 293 T cells transfected with Flag-tagged IGFBP5 and HA-tagged ROR1. Cell lysates were precipitated with anti-Flag antibody. **i** Co-IP analysis for the interaction of IGFBP5 and ROR1 in 293 T cells transfected with HA-tagged ROR1 and Flag-tagged IGFBP5. Cell lysates were precipitated with anti-HA antibody. **j** In vitro binding affinity between IGFBP5 and ROR1 tested by MST assay. The concentration of IGFBP5 proteins is kept constant at 50 nM, while the ROR1 concentration varies from 1.45 μM to 0.04 nM. The binding curve yields a Kd of 157.5 nM. Inset, thermophoretic movement of fluorescently labeled proteins. Fnorm = $F_1/F_0$ (Fnorm: normalized fluorescence; $F_1$: fluorescence after thermodiffusion; $F_0$: initial fluorescence or fluorescence after T-jump). Kd, dissociation constant. **k** IB analysis of pROR1, pHER2, pCREB, ROR1, HER2, and CREB in X01 GSCs infected with shCtrl, shHER2-1, or shHER2-2 lentivirus. GAPDH was used as a loading control. **l** IB analysis of pROR1, pHER2, pCREB, ROR1, HER2, and CREB in X01 GSC infected with shCtrl, shROR1-1, or shROR1-2 lentivirus. GAPDH was used as a loading control. **m** IHC analysis of pHER2, pROR1, and pCREB in orthotopic xenografts of X01 GSCs infected with shCtrl, shIGFBP5-1, or shIGFBP5-2 lentivirus. Scale bar, 100 μm. **n** IHC analysis of pHER2, pROR1, and pCREB in orthotopic xenografts of 83 GSCs infected with vector control or IGFBP5 lentivirus. Scale bar, 100 μm. All the immunoblots were representative data from three independent experiments. Source data are provided as the Source Data file.

RNA-seq analysis after IGFBP5 knockdown. To identify specific genes associated with IGFBP5 and GSC invasion, we selected 22 DEGs shared between the upregulated DEGs in invasive vs non-invasive GSCs and the downregulated DEGs in IGFBP5-knockdown vs control 448 and X01 GSCs (Fig. 5a, b). In addition to IGFBP5, ETS variant transcription factor 5 (ETV5), which is highly correlated with HGG and gliogenesis[35], and F-box and WD repeat domain containing 9 (FBXW9), a member of the F-box protein family[36], were positively associated with a poor patient survival rate (Fig. 5c). We then confirmed that IGFBP5 positively regulated ETV5 and FBXW9 in our GSC model and found that the expression of both ETV5 and FBXW9 was increased by ectopic IGFBP5 overexpression (Fig. 5d, e) and decreased by IGFBP5 knockdown (Fig. 5f, g). Moreover, the expression levels of ETV5 and FBXW9 were higher in invasive GSCs than in non-invasive GSCs (Fig. 5h, i). To determine whether ETV5 and FBXW9 were involved in regulating GSC invasion, we suppressed ETV5 and FBXW9 with small interfering RNA (siRNA) (Supplementary Fig. 5a–f). Downregulated ETV5 or FBXW9 significantly decreased the invasion capacities in both X01 and 448 GSCs, indicating that ETV5 and FBXW9 regulate GSCs invasion (Supplementary Fig. 5b, c, e, f). To examine whether ETV5 and FBXW9 are directly regulated by CREB at the transcriptional level, we examined the binding of CREB to the promoter regions of these two genes after IGFBP5 knockdown/overexpression. Chromatin immunoprecipitation (ChIP)-qPCR assays were executed using a CREB-specific antibody and primer sets positioned in the promoter region of the *ETV5* gene (*ETV5*-Prom 1: chr3: 185827212-185827368) and near the transcription start site (TSS) of the *FBXW9* gene (*FBXW9*-TSS: chr19: 12807377-12807602) based on the hg19 genome. The ChIP-qPCR results indicated significant enrichment of *ETV5* and *FBXW9* binding in the CREB-immunoprecipitated samples compared with the control IgG-immunoprecipitated samples from X01 GSCs (Supplementary Fig. 6a, b). Furthermore, the CREB occupancy at the promoter or TSS was significantly decreased by IGFBP5 knockdown and increased by IGFBP5 overexpression (Fig. 5j–l and Supplementary Fig. 6c). These results suggest that IGFBP5 activates the transcription of *ETV5* and *FBXW9* via CREB activation.

To determine whether IGFBP5 transduces downstream signaling through CREB, we first overexpressed CREB in non-invasive GSC and investigated the role of CREB in regulating the downstream genes expression and GSC invasion. Immunoblot analysis verified a successful ectopic CREB overexpression in non-invasive GSCs (83 and 131) (Supplementary Fig. 6d, e). CREB overexpression significantly increased the expression of downstream *ETV5* and *FBXW9* genes, and the invasion capacity of GSCs (Supplementary Fig. 6f–k). To investigate the downstream role of CREB in the IGFBP5-ROR1/HER2 signaling pathway, we ectopically overexpressed CREB in IGFBP5 knockdown GSCs, and found that CREB sufficiently restored GSC invasion (Supplementary Fig. 6l–n). Furthermore, CREB overexpression also rescued HER2 or ROR1 knockdown mediated repression of GSC invasion

(Supplementary Fig. 7a–h). These results indicate that CREB is a key molecule in IGFBP5-mediated ROR1/HER2 signaling axis.

Given the potential involvement of IGFBP5, ETV5, and FBXW9 in the invasive potential of GSCs, we subsequently aimed to identify associations between the expression of these three genes and patient survival using TCGA glioma datasets. By analyzing the patient survival rates in TCGA glioma datasets, we elucidated the clinical significance of the regulation of ETV5 and FBXW9 by IGFBP5. Glioma patients with high ETV5 or FBXW9 expression experienced worse survival than those with low expression (Fig. 5c), and high IGFBP5 expression in combination with high ETV5 or FBXW9 expression was significantly associated with the worst survival rate among all the groups (Fig. 5m). Therefore, the increase ETV5 and FBXW9 expression in response to IGFBP5 has a negative impact on the survival of patients with glioma.

Collectively, our results suggest that IGFBP5 promotes *ETV5* and *FBXW9* transcription through the binding of CREB to the promoters of these two target genes, and the altered expression patterns of these three genes are correlated with poor patient survival.

## IGFBP5 supports GSC invasion and tumor progression in patient-derived xenograft model

To further address whether IGFBP5 promotes GSC invasion and tumor progression in vivo, we knocked down the expression of IGFBP5 in a patient-derived GSC (PDC) and tested the resulting antitumor activity in PDC-derived orthotopic xenograft models (PDXs), which generally recapitulate both the genetic and histological profiles of donor patient-derived tumors. The results showed that PDX tumors derived from 772 GSC, which was isolated directly from tumor of patient with GBM (National Cancer Center, Korea) demonstrated invasive properties (Fig. 6a). Further RNA-seq analysis cataloged 772 GSCs into the PN subtype, which is consistent with our invasive GSC subtype characterization (Fig. 6b). The knockdown of IGFBP5 expression in 772 GSCs (Fig. 6c) suppressed GSC invasion (Fig. 6d and Supplementary Fig. 8a), as well as the expression of the target genes *ETV5* and *FBXW9* (Supplementary Fig. 8b, c), and tumor volume (Fig. 6e), while significantly prolonged the survival of the mouse (Fig. 6f). Interestingly, the mice injected with control GSCs showed myelin-associated invasion in the corpus callosum and deeply invaded into opposite hemisphere compared with both shIGFBP5 (Red box, Fig. 6e). Moreover, the phosphorylation levels of ROR1, HER2, and CREB were lower in mice with IGFBP5-knockdown tumors than in those with control 772 tumors (Fig. 6g). Taken together, the results from our original patient-derived 772 GSCs strongly suggest that IGFBP5 promotes GSC invasion, as well as HER2, ROR1, and CREB phosphorylation in vivo.

## IGFBP5-targeting CRISPR/Cas9 delivered by tumor-penetrating nanocapsules decreases GSC invasion and tumor progression

Since our results suggest that IGFBP5 promotes GSC invasion and tumor progression, we subsequently assessed the potential

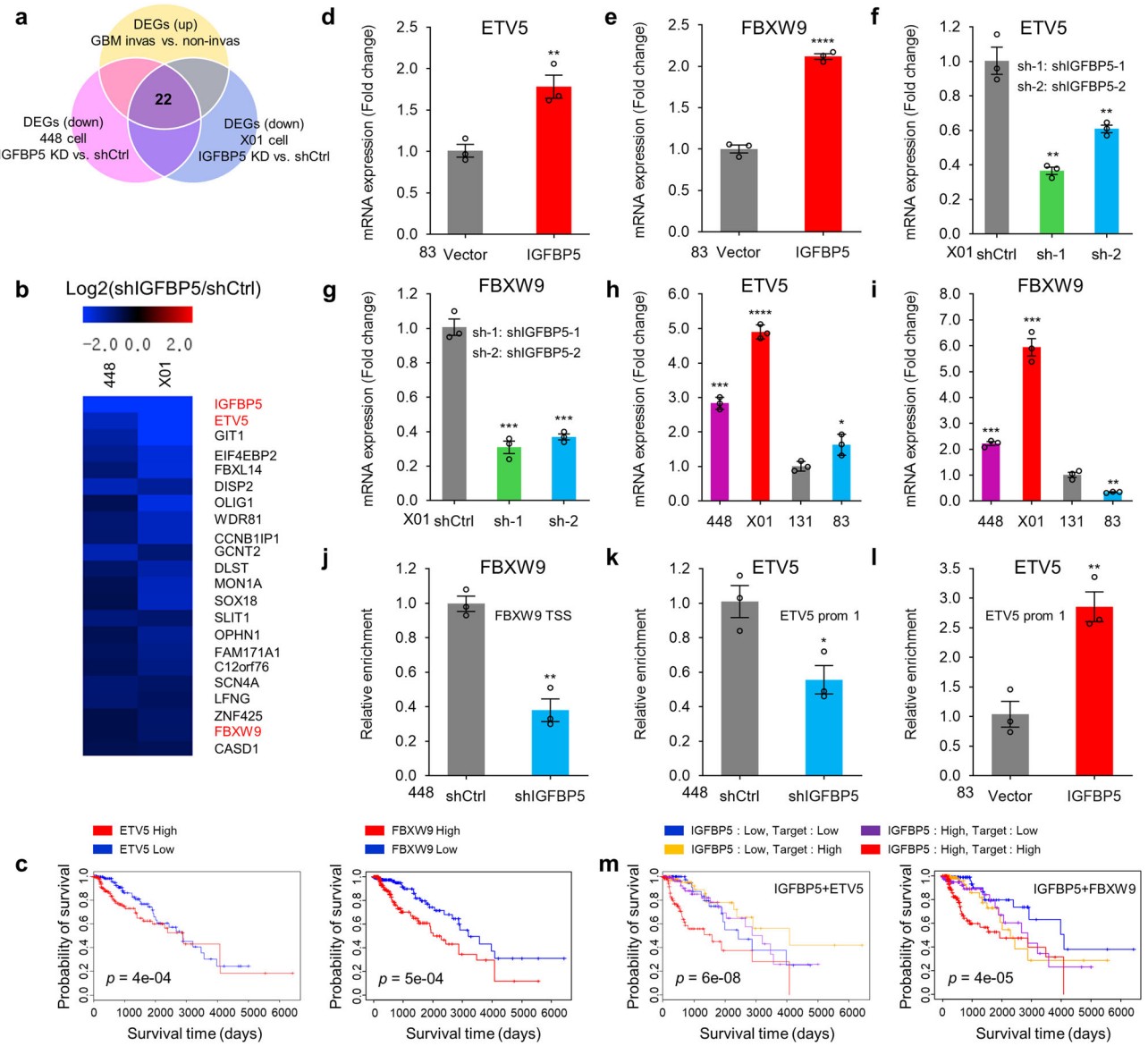

**Fig. 5 | IGFBP5 activates transcription of *ETV5* and *FBXW9* via CREB. a** Venn diagram showing the overlap of upregulated DEGs in invasive vs non-invasive GSCs and downregulated DEGs in 448 and X01 GSCs infected with shIGFBP5 versus GSCs infected with shCtrl based on RNA-seq data. The 22 selected genes are putative invasion-related target genes of IGFBP5. **b** Heatmap of the log2-fold change in putative invasion-related target genes of IGFBP5 identified by RNA-seq analysis of invasive GSCs infected with shIGFBP5 or shCtrl. **c** Kaplan–Meier survival curves for LGG patients with high or low ETV5 (left, *P* = 0.0004) or FBXW9 expression (right, *P* = 0.0005), log-rank test. **d, e** RT-qPCR analysis of ETV5 and FBXW9 expression in 83 GSCs infected with vector control or IGFBP5 lentivirus. Data are presented as mean ± SEM (*n* = 3 independent experiments), two-tailed Student's *t*-test (**d, e**, \*\**P* = 0.008, \*\*\*\**P* = 0.00004). **f, g** RT-qPCR analysis of ETV5 and FBXW9 expression in X01 GSCs infected with shCtrl, shIGFBP5-1, or shIGFBP5-2 lentivirus. Data are presented as mean ± SEM (*n* = 3 independent experiments), two-tailed Student's *t*-test (left to right, (**f**) \*\**P* = 0.001, \*\**P* = 0.008; (**g**) \*\*\**P* = 0.0003,

\*\*\**P* = 0.0002). **h, i** RT-qPCR analysis of ETV5 and FBXW9 expression in invasive GSCs (X01 and 448) and non-invasive GSCs (131 and 83). Data are presented as mean ± SEM (*n* = 3 independent experiments), two-tailed Student's *t*-test (compare with 131 group, left to right, \*\*\**P* = 0.0002, \*\*\*\**P* = 0.00001, \**P* = 0.03; \*\*\**P* = 0.0006, \*\*\**P* = 0.0001, \*\**P* = 0.003). **j** ChIP-qPCR analysis of CREB binding to the *FBXW9* TSS in 448 GSCs infected with shCtrl or shIGFBP5-1 lentivirus. Data are presented as mean ± SEM (*n* = 3 independent experiments), two-tailed Student's *t*-test (\*\**P* = 0.002). **k, l** ChIP-qPCR analysis of CREB binding to the *ETV5* promoter in 448 GSCs infected with shCtrl or shIGFBP5-1 lentivirus or in 83 GSCs infected with vector control or IGFBP5 lentivirus. Data are presented as mean ± SEM (*n* = 3 independent experiments), two-tailed Student's *t*-test (**k, l**, \**P* = 0.02, \*\**P* = 0.006). **m** Kaplan–Meier survival curves for LGG patients stratified by IGFBP5 and ETV5 (left, *P* = 0.00000006) or IGFBP5 and FBXW9 expression (right, *P* = 0.00004), log-rank test. Source data are provided as the Source Data file.

therapeutic target role of IGFBP5 in GBM. To this end, we developed Angiopep-2 decorated, reduction-sensitive CRISPR-Cas9 ribonucleo-protein (RNP)-based nanocapsules for IGFBP5 gene editing (denoted as "Ang-SS-Cas9/sgIGFBP5"), accordingly constructed to our previous reported[37,38] (Fig. 7a). The averaged diameter of Ang-SS-Cas9/sgIGFBP5 nanocapsules was 32 nm, consistent with the previously reported thickness of the polymerization layer[39,40] (Fig. 7b). Transmission electron microscopy (TEM) revealed that Ang-SS-Cas9/sgIGFBP5 is

spherical shaped under physiological conditions (Fig. 7c, left panel). Given the employment of -SS- as a responsive linker in the nanostructure design, reductive agents, including glutathione (GSH), in the intracellular microenvironment (2–10 mM GSH in tumor cell cytoplasm) can be easily disassemble the nanocapsules, leading to the reduction-triggered release of loaded Cas9/sgIGFBP5 (Fig. 7c, right panel). In addition, because both endothelial cells in the blood-brain barrier (BBB) and glioma cells show high expression of receptor-

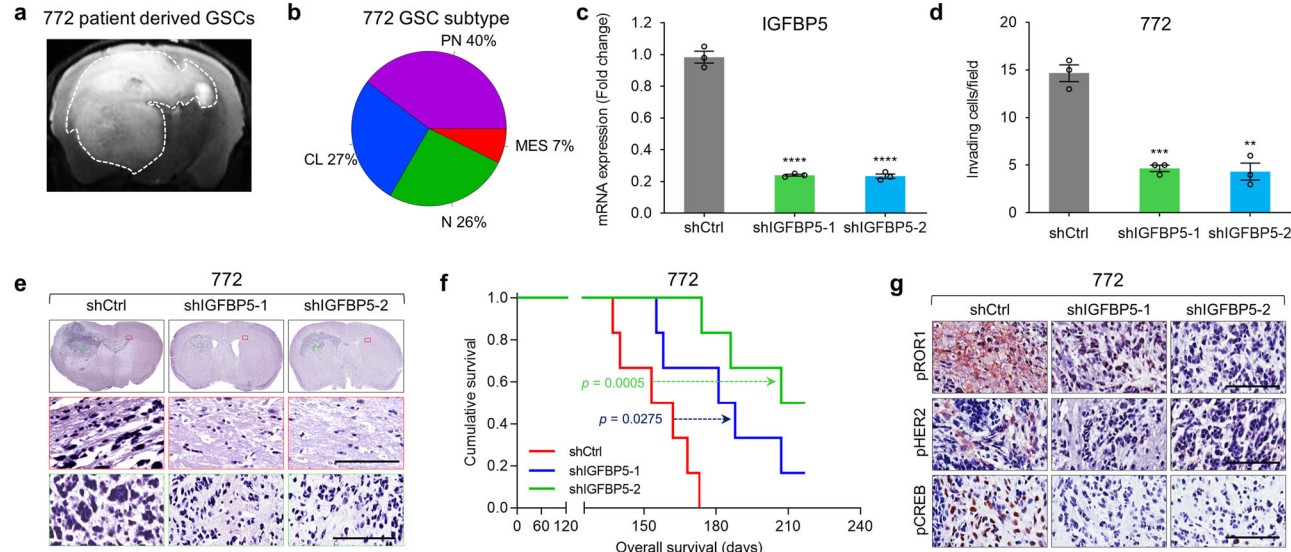

**Fig. 6 | IGFBP5 knockdown suppresses GSC invasion and tumorigenesis in patient-derived xenografts. a** Magnetic resonance imaging (MRI) of mice bearing orthotopic xenografts of patient-derived 772 GSCs. **b** Categorization of patient-derived 772 GSCs based on the RNA-seq analysis. **c** RT-qPCR analysis of IGFBP5 expression in patient-derived 772 GSCs infected with shCtrl, shIGFBP5-1, or shIGFBP5-2 lentivirus. Data are presented as mean ± SEM ($n = 3$ independent experiments), two-tailed Student's $t$-test (left to right, ***$P = 0.00004$, ****$P = 0.00005$). **d** Invasion assays with patient-derived 772 GSCs infected with shCtrl, shIGFBP5-1, or shIGFBP5-2 lentivirus. The graph shows the mean number of invasive cells ± SEM ($n = 3$ independent experiments), two-tailed Student's $t$-test

(left to right, ***$P = 0.0004$, **$P = 0.001$). **e** H&E staining of the whole brains of mice bearing orthotopic xenografts of patient-derived 772 GSCs infected with shCtrl, shIGFBP5-1, or shIGFBP5-2 lentivirus. Scale bar, 100 μm. **f** Kaplan–Meier survival curves of mice implanted with patient-derived 772 GSCs infected with shCtrl, shIGFBP5-1, or shIGFBP5-2 lentivirus ($n = 6$ in each group, $1 \times 10^5$ cells/mouse). shCtrl vs shIGFBP5-1, $P = 0.0275$; shCtrl vs shIGFBP5-2, $P = 0.0005$; log-rank test. **g** IHC analysis of pHER2, pROR1, and pCREB in orthotopic xenografts of patient-derived 772 GSCs infected with shCtrl, shIGFBP5-1, or shIGFBP5-2 lentivirus. Scale bar, 100 μm. Source data are provided as the Source Data file.

related protein 1 (LRP-1), which is high binding affinity to its ligand Angiopep-2[37], conjugated Angiopep-2 is then able to efficiently drive the developed Ang-SS-Cas9/sgIGFBP5 nanocapsules across the BBB and subsequently actively target GSCs in the brain, towards effective IGFBP5 gene editing via reductive-responsive Cas9/sgIGFBP5 release.

The therapeutic potential of IGFBP5 gene editing was evaluated in both in vitro and in vivo models. T7 endonuclease I (T7EI) cleavage assays showed that Ang-SS-Cas9/sgIGFBP5 lead to 65.9% and 31.8% gene disruption in X01 and 448 cells, respectively (Fig. 7d, e). Compared with Ang-SS-Cas9/sgScramble, the Ang-SS-Cas9/sgIGFBP5-loaded nanocapsules significantly reduced invasive ability in both X01 and 448 GSCs (Fig. 7f–i). Subsequently, the ability of Ang-SS-Cas9/sgIGFBP5 to restrict GSC tumor progression was evaluated in vivo. Mice harboring orthotopic X01-Luc (stable luciferase-expressing X01 GSCs) tumors were intravenously injected with Cas9/sgIGFBP5-loaded nanocapsules every second day for a total of 10 days (Fig. 7j). Both tumor bioluminescence intensity and mice survival were monitored. The Ang-SS-Cas9/sgIGFBP5 treated mice showed significant tumor inhibition than that of Ang-SS-Cas9/sgScramble and PBS (Fig. 7j). Notably, two out of five Ang-SS-Cas9/sgIGFBP5 treated mice showed complete fluorescence abolishment at day 24 post-implantation (Fig. 7j). Furthermore, the IGFBP5 protein level and phosphorylation of IGFBP5 downstream proteins (i.e., ROR1, HER2, and CREB), were remarkably reduced in mice injected with Cas9/sgIGFBP5-loaded nanocapsules (Fig. 7k). Particularly, the body weight of mice treated with Ang-SS-Cas9/sgIGFBP5 exhibited little changes across the treatment period, in comparison to the Ang-SS-Cas9/sgScramble and PBS (Fig. 7l). Last but not the least, in vivo nanocapsule-mediated IGFBP5 disruption dramatically extended mouse survival (Fig. 7m). To further confirm the specificity of IGFBP5 regulatory role for invasive GSCs, we performed Ang-SS-Cas9/sgIGFBP5 in a non-invasive 83-Luc (stable luciferase-expressing 83 GSCs) GSC-bearing mice model. The results showed the identical outcomes of tumor growth and mouse survival among Ang-SS-Cas9/sgIGFBP5, Ang-SS-Cas9/sgScramble and PBS

(Supplementary Fig. 9a–c). In summary, these results highlight the powerful diminishment of tumorigenicity via Ang-SS-Cas9/sgIGFBP5 mediated in vivo IGFBP5 gene editing, proposing the potential therapeutic target role of IGFBP5 in invasive GBM.

## Discussion

Due to the highly invasive nature of GSCs, these cells readily spread and migrate to neighboring brain tissue, leading to the key challenges in achieving complete GBM resection. Thus, there is an urgent need to elucidate the invasion mechanism and identify therapeutic targets. To date, the genetic variations or molecular mechanisms underpinning the invasive and non-invasive phenotypes of GSCs have yet been elucidated. Therefore, in this study, we aimed to investigate the invasive behavior of GSCs from molecular, mechanistic, clinical, and therapeutic perspectives. IGFBP5 was identified as the top upregulated gene in invasive vs non-invasive GSCs that was significantly associated with the poor prognosis of patients with glioma. The results showed that IGFBP5 is a ligand for ROR1, and activates the ROR1/HER2-CREB signaling axis and the transcription of *ETV5* and *FBXW9*. In glioma patients, high IGFBP5 expression alone or in combination with high ETV5 or FBXW9 expression is significantly associated with a poor prognosis. Finally, we showed that IGFBP5 can be employed as an effective therapeutic target using a nanocapsule-based platform to decrease tumor progression and improve the survival rate.

Three IGFBP isoforms, IGFBP2, IGFBP3, and IGFBP5, are reportedly overexpressed in clinical biopsies of high-grade diffuse glioma[41–43]. IGFBP2 regulates glioma progression by activating EGFR-STAT3 signaling and promoting cell proliferation and chemoresistance via the integrin β1-ERK pathway, whereas IGFBP3-STAT1 activation is associated with neoplastic cell invasion and poor patient outcomes[44–46]. However, few studies have illustrated the molecular mechanisms of IGFBP5. Importantly, we found that IGFBP5 showed remarkable higher expression in brain tumor tissue than normal brain

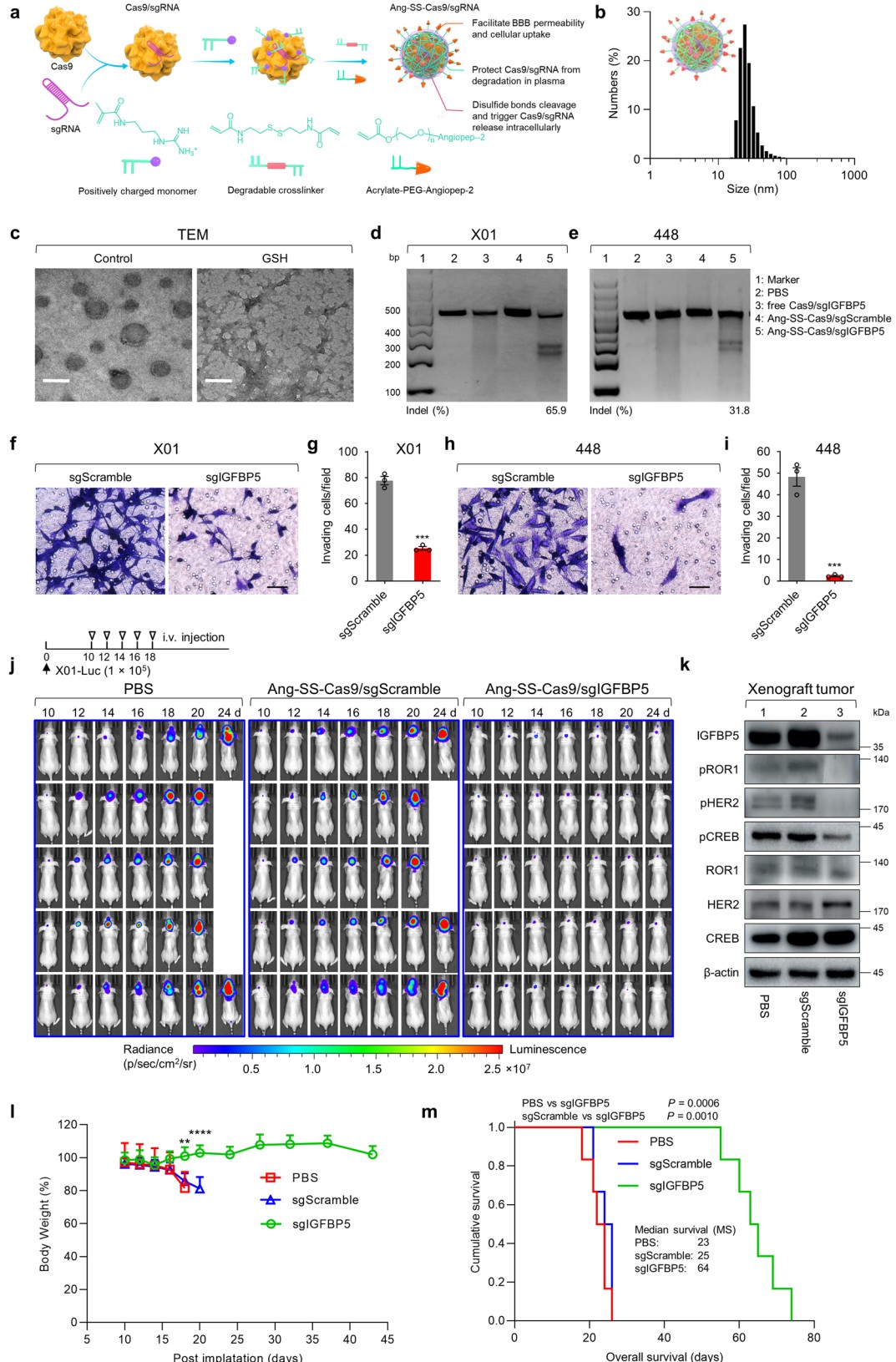

tissue, and our RNA-seq data revealed increased IGFBP5 expression in invasive GSCs compared with non-invasive GSCs, which suggests that the IGFBP5 signaling axis regulates GSC invasion. To the best of our knowledge, this study provides the first demonstration of IGFBP5 as the key regulator of the invasive nature of GSCs both in vitro and in vivo.

Even though IGFBPs can affect the action of IGF, these cells can also initiate IGF-independent actions[47]. In our study, the results suggested that IGFBP5 serves as a ligand for ROR1 and triggers IGF-independent oncogenic signaling. Since we observed no IGF1R phosphorylation in GSCs treated with human rIGFBP5 protein in phospho-RTK array, and immunoblot analysis confirmed that human rIGFBP5

**Fig. 7 | Nanocapsule-mediated delivery of Cas9/sgIGFBP5 suppresses GSC invasion and tumorigenesis. a** Schematic illustration of Ang-SS-Cas9/sgRNA nanocapsule preparation. **b** Size distribution of Ang-SS-Cas9/sgRNA nanocapsules. **c** TEM images of Ang-SS-Cas9/sgRNA nanocapsules treated with or without 10 mM GSH at pH 7.4 for 12 h. Scale bar, 50 nm. **d, e** Indel detection by T7 endonuclease I (T7EI) of X01 and 448 GSCs treated by Ang-SS-Cas9/sgIGFBP5. T7EI cleavage assays were representative data from three independent experiments. **f, g** Invasion assays with X01 GSCs treated with Ang-SS-Cas9/sgIGFBP5 and Ang-SS-Cas9/sgScramble. **f** Images taken after 48 h of invasion are representative of three independent experiments (scale bar, 100 μm; $n = 3$), and **g** the graph shows the mean number of invasive cells ± SEM, two-tailed Student's $t$-test (***$P = 0.0001$). **h, i** Invasion assays with 448 GSCs treated with Ang-SS-Cas9/sgIGFBP5 and Ang-SS-Cas9/sgScramble. **h** Images taken after 48 h of invasion are representative of three independent experiments (scale bar, 100 μm; $n = 3$), and **i** the graph shows the mean number of invasive cells ± SEM, two-tailed Student's $t$-test (***$P = 0.0004$). **j** Luminescence images of orthotopic X01-Luc human glioblastoma tumor-bearing nude mice following treatment with Ang-SS-Cas9/sgIGFBP5, Ang-SS-Cas9/sgScramble or PBS. Mice were intravenously injected at a dose of 1.5 mg Cas9 equiv./kg on day 10, 12, 14, 16, and 18 post tumor implantations. **k** IB analysis of IGFBP5, pROR1, pHER2, pCREB, ROR1, HER2 and CREB in tumor tissues taken from mice treated with Ang-SS-Cas9/sgIGFBP5, Ang-SS-Cas9/sgScramble or PBS. β-actin was used as the loading control. The immunoblots were representative data from three independent experiments. **l** Body weight changes in mice following treatment with Ang-SS-Cas9/sgIGFBP5, Ang-SS-Cas9/sgScramble or PBS. Data are presented as mean ± SEM ($n = 6$ in each group), one-way ANOVA (day 18, ***$P = 0.0006$) and two-tailed Student's $t$-test (day 20 sgIGFBP5 vs sgScramble, ***$P = 0.00008$). **m** Kaplan–Meier survival curves of mice implanted with $1 \times 10^5$ X01-Luc GSCs and treated with Ang-SS-Cas9/sgIGFBP5, Ang-SS-Cas9/sgScramble or PBS ($n = 6$ in each group). PBS vs sgIGFBP5, $P = 0.0006$, sgScramble vs sgIGFBP5, $P = 0.001$, log-rank test. Source data are provided as the Source Data file.

protein does not impact IGF1R phosphorylation (Supplementary Fig. 10a), and IGF1R blockade has no apparent effect on the IGFBP5 mediated ROR1/HER2-CREB signaling axis (Supplementary Fig. 10b). Subsequently, the increased phosphorylation of HER2 and ROR1 observed in same array suggests their crucial roles in the signaling axis. Our results suggested that IGFBP5 serves as a ligand for ROR1 and triggers IGF-independent oncogenic signaling. Several potential kinase targets are reportedly involved in glioma invasion or proliferation, including ERK1/2[48], Chk-2[49,50], CREB[51,52], HSP60[53,54], were identified. In addition, CREB, which is a protooncogenic "master" transcription factor[55,56] that promotes tumorigenesis in many cancers (i.e., non–small-cell lung carcinoma, breast cancer, acute myeloid leukemia, hepatocellular carcinoma[57–60], and glioma[51,52]), is also tightly associated with HER2-mediated neoplastic cell growth and ROR1-mediated leukemia cell growth[34,61], supporting our findings that IGFBP5 increases the phosphorylation of HER2 and ROR1 during GSC invasion. Further analysis revealed that the knockdown of IGFBP5 expression decreased the level of CREB phosphorylation, which suggests a potential signaling pathway involving IGFBP5, ROR1, HER2, and CREB that cooperatively supports the invasive behavior of GSCs. Thus, in contrast to its role in other types of cancer, IGFBP5 may serve as an oncogene by activating ROR1/HER2-CREB signaling axis in glioma.

To further investigate the IGFBP5 pathway, we sought to identify the downstream transcriptional targets. ChIP assays confirmed that the *ETV5* and *FBXW9* genes, which were found to be downregulated in shIGFBP5-treated invasive GSCs, are direct targets of CREB, and the levels of these genes were negatively associated with patient survival. ETV5 is highly expressed in glioma cells, and the upregulation of ETV5 enhances ovarian cancer cell survival;[35,62] however, no previous study has identified the function of FBXW9 in solid tumors[63]. Although our analysis revealed a considerable level of correlation between high ETV5 or FBXW9 expression and poor prognosis in patients with glioma, further studies are required to reveal the mechanical role of ETV5 and FBXW9 in GSC invasion and tumorigenesis.

As an undruggable target, conventional therapies via in vivo targeting of IGFBP5 are limited. To this end, we developed a CRISPR/Cas9-based nanocapsule that can be intravenously injected and actively target to the brain tumor for in vivo IGFBP5 gene editing, based on our previous report[37,38,64]. The employment of disulfide bond (SS) and Angiopep-2, empowered the nanocapsules with BBB-penetrating and tumor-targeting abilities[37]. Upon uptake inside GBM cells, the high level of GSH in the tumor microenvironment triggers the site-specific breakage of disulfide bonds and the local release of the Cas9/sgIGFBP5 cargo[65].

Our results reveal the Cas9/sgIGFBP5-loaded nanocapsules has led to a significant reduction in tumor growth and profoundly extended longevity. This treatment is firstly highlighted with intravenous injection application decorated with brain tumor target ability, can thusly be considered as a more clinical approachable and patient-friendly therapy, in comparison to the intratumoral injection[66,67]. More importantly, our first establishment of CRISPR/Cas9-based gene therapy against the above-identified GBM invasion regulator, IGFBP5, showed a much preferable outcome on tumor growth as well as mouse life expectancy, comparing to the well-known siRNA-based therapies targeting the conventional GBM targets (i.e., PLK1[37] or STAT3[68]). Although the current PDXs model only can be established in immunodeficient mice such as nude, NOD-SCID, or NOD-SCID-gamma mice. However, these preclinical data offer a proof-of-concept for the treatment of invasive GBM by targeting IGFBP5.

Although we focused on IGFBP5, which encodes an easily targetable extracellular molecule, we believe that further study of the other identified genes is warranted to ascertain their involvement in GSC invasion. The DEGs identified in the RNA-seq analysis are associated with various functions related to invasive GSC characteristics, including GABAergic synapses, signaling pathways regulating pluripotency and basal cell carcinoma, which implies their functional importance in GSC invasion. Thus, future studies should evaluate the roles of these DEGs in GSC invasion.

In conclusion, IGFBP5 increases GBM invasion and promotes tumor growth through the ROR1/HER2-CREB signaling axis. As targeting IGFBP5 significantly prolonged the survival of mice bearing orthotopic GBM, we may shine more light on a potentially promising therapeutic approach to greatly increase the life expectancy of patients with GBM, particularly those with a poor prognosis related to GBM invasion or reoccurrence.

## Methods

### Cell culture

X01[69], 448[70], 131[71], 83[13], and 772 GSCs were maintained in DMEM/F12 (Welgene) supplemented with EGF (10 ng/ml; R&D Systems), bFGF (5 ng/ml; for X01, 448, 131 and 83; 10 ng/ml for 772; R&D Systems), B27 (Invitrogen), and 1% penicillin/streptomycin (P/S; HyClone). The 772 GSC was newly isolated using Patel's method and supplied additional N2 supplement (Gibco)[72]. The 293 T cells purchased from American Type Culture Collection (ATCC) were maintained in high-glucose DMEM (HyClone) containing 10% FBS (HyClone) and 1% P/S. All cell cultures were incubated at 37 °C in a humidified incubator with 5% $CO_2$ and repeatedly screened for mycoplasma.

### Plasmids

pCDH-Flag-IGFBP5, pCDH-Flag-CREB, pCDH-HA-ROR1 were subcloned for co-IP and overexpression experiments using pCDH-RFP vector (Genechem). PCR amplified oligomers were as follows: IGFBP5, sense 5'-GCTCTAGAGCCACCATGGATTACAAGGATGACGACGATAAGGTGTTGCTCACCGCGG-3' and antisense 5'-CGGGATCCCTCAACGTTGCTGCTGT-3'; CREB, sense, 5'-GCTCTAGAATGGATTACAAGGATGACGACGATAAGATGACCATGGAATCTGG-3' and antisense 5'-GGAATTCGCGGCCGCTTAATCTGATTTGTGGC-3'; ROR1 wild type, sense 5'-GCTCTAGAA

TGTACCCATACGATGTTCCAGATTACGCTATGCACCGGCCGCG-3′ and antisense 5′-AAGGAAAAAAGCGGCCGCTTACAGTTCTGCAGAAATCAT AGATTC-3′; ROR1 kinase domain deletion type (ROR1 mut), sense 5′-CCAAGAGCAAGGCTAAAGAGGGACTCTCAAGTCA-3′ and antisense 5′-TGACTTGAGAGTCCCTCTTTAGCCTTGCTCTTGG-3′. Negative Control Vector were purchased from Sino Biological (Beijing, China). For the generation of HRST-IGFBP5-IRES-GFP construct for the lentiviral transduction (83, 131 overexpression), PCR was performed with the following oligomers; sense 5′-GATCTCGACGCGGCCGCATGGTGTT GCTCACCGCG-3′ and antisense 5′-GGGCGGAATTGGATCCTCAC TCAACGTTGCTGCTG-3′. The amplified DNA fragments were digested with NotI-BamHI and subcloned into HRST-IRES-GFP treated with NotI-BamHI-CIP. All shRNA-expressing lentiviral constructs targeting IGFBP5, ROR1, and HER2 were constructed by ligating annealed oligomers with AgeI-EcoRI-digested pLKO.1 puro (Addgene). The nucleotide sequences used for shRNA are as follows: shIGFBP5-1, 5′-GCAAGTCAAGATCGAGAGAGA-3′; shIGFBP5-2, 5′-CGACGAGAAG CCCTCTCCAT-3′; shROR1-1, 5′-CAAGATCAAATCCCATGATTC-3′; shROR1-2, 5′-GCACCGTCTATATGGAGTCTT-3′; shHER2-1, 5′-TGTCAG-TATCCAGGCTTTGTA-3′; shHER2-2, 5′-CAGTGCCAATATCCAGGAGTT-3′. All oligomers were purchased from Tsingke (Beijing, China). All constructs were verified by DNA sequencing (Tsingke).

## Conditioned medium preparation and recombinant protein treatment

To collect conditioned medium (CM) from GSCs (X01, 448, 131 and 83), $5 \times 10^5$ GSCs (X01, 448, 131 and 83) were plated in bovine fibronectin (R&D System) coated 60 mm dishes, and after 24 h, the medium was replaced with medium with/without additional lentivirus. CM was collected after 48 h of culture, centrifuged at 1000 g for 20 min at 4 °C to remove the debris, and kept in aliquots at −20 °C for further use.

For recombinant human IGFBP5 protein treatment, $8 \times 10^5$ 131 or 83 GSCs were plated in 60 mm dishes and incubated overnight. Recombinant human IGFBP5 protein (100 ng/ml; R&D System) was added to the cultures of 131 and 83 cells for 6 h, and harvested cell pellets were used for further study. For IGF1R inhibitor treatment, $8 \times 10^5$ 83 GSCs were plated in 60 mm dishes and incubated overnight. NVP-AEW541 (10 μM; TargetMol) was added into the culture medium for 3 h, with or without pre-treatment of recombinant human IGFBP5 protein for 6 h, and harvested cell pellets were used for further study.

## Lentivirus production and infection

Lentiviruses were produced as previously reported[8]. Briefly, $3-4 \times 10^6$ 293 T cells were plated on 100 mm culture dishes, incubated for 24 h, and then cotransfected with 4.5 μg of lentiviral constructs (pLKO-shCtrl, pLKO-shIGFBP5-1, pLKO-shIGFBP5-2, pLKO-shHER2-1, pLKO-shHER2-2, pLKO-shROR1-1, pLKO-shROR1-2, pHRST-vector, pHRST-IGFBP5, pCDH-vector, pCDH-Flag-CREB, and pCDH-HA-ROR1 (WT, mut), 3 μg of psPAX2 (Addgene), and 1.5 μg of pMD2.G (Addgene) using 27 μL of Lipofectamine 2000 (Invitrogen). The medium was changed 6 h after transfection, and 48 h after transfection, medium containing lentivirus was harvested. The viral particles were concentrated and purified using a Lenti-X concentrator (Clontech). Cells were infected with lentivirus in the presence of 6 μg/ml polybrene.

## RNA interference

The siRNA targeting ETV5 and FBXW9 were designed by Bioneer (Korea). The siRNA sequences were as follows: siETV5-1, 5′-CACAAG-CUUAGAUUCUCUA-3′; siETV5-2, 5′-CACAGAUCUGGCUCACGAU-3′; siFBXW9-1, 5′-AGUCCAACCAGGUUCUGAU-3′; siFBXW9-2, 5′-GUUGA-GAGCGAAGGAGAAA-3′. X01 and 448 GSCs were seeded in 60 mm culture plates and then transfected with 100 pmole siRNA with Lipofectamine 2000 (Invitrogen) according to the manufacture's instruction. The cells harvested at 48 h after transfection were used for further experiments.

## ELISA

The level of IGFBP5 in the CM from GSCs (X01, 448, 131, and 83) and lentiviral-infected GSCs was measured by sandwich ELISA using an IGFBP5 human ELISA Kit (Abcam #ab100543).

## Invasion assay

Transwell chambers (Corning) were coated with Matrigel Basement Membrane Matrix (BD). GSCs were suspended in DMEM/F12 medium supplemented with 1% P/S media and seeded into the upper chamber at a density of 3 to $5 \times 10^4$ cells per well, and GSC full medium was placed in the lower chamber. After incubation for 6–48 h, the cells that penetrated the pores were stained with Diff-Quik staining solution (Sysmex) and observed under bright-field microscope.

## Chromatin immunoprecipitation assay

Chromatin immunoprecipitation (ChIP) was performed using Protein A and G Dynabeads (Thermo Fisher Scientific). For each ChIP reaction, $5 \times 10^6$ GSCs were cross-linked with 1% formaldehyde for 10 min at 37 °C, and genomic DNA was fragmented into ~100-to-300-bp pieces by sonication (truChIP Chromatin Shearing Reagent Kit, Covaris). DNA-bound CREB was immunoprecipitated using a CREB-specific antibody (Cell Signaling Technology). The associated DNA was then purified and analyzed by RT-qPCR to detect specific DNA sequences within the ETV5 or FBXW9 promoter bound by the CREB protein. The data were normalized to the input levels, and at least three independent biological replicates of each ChIP-qPCR were performed. An antibody against IgG (Abcam) was used as a nonspecific control. The primers were as follows: ETV5 prom 1, sense 5′- GACCTGAGGGGGAAGCTTAG-3′ and antisense 5′-TTTGCTGGATGGAGAAGTGG-3′; FBXW9 TSS, sense 5′-G CCCTAGGGGAAGCTCCATT-3′ and antisense 5′-AAAAGCGCAGAAAC AGGAAC-3′.

## Quantitative RT-PCR (RT-qPCR)

Total RNA from GSCs was isolated using a RNeasy Mini Kit (QIAGEN), and 1 μg of total RNA was used as the template to synthesize cDNA using the AMPIGENE cDNA Synthesis Kit (Enzo). Quantitative RT-PCR analysis was performed with a LightCycler 480II real-time detection system (Roche) using AMPIGENE qPCR Green Mix Hi- ROX enzyme (Enzo). The expression levels of the target genes were normalized to that of β-actin. The primers were as follows: β-actin, sense 5′-GAGGC ACTCTTCCAGCCTTC-3′ and antisense 5′-GGATGTCCACGTCACA CTTC-3′; IGFBP5, sense 5′-CGTGCTGTGTACCTGCCCAA-3′ and antisense 5′-GCTGTCGAAGGTGTGGCACT-3′; ETV5, sense 5′-GGCTCACG ATTCTGAAGAGC-3′ and antisense 5′-AAGACGACAGCTCAGAGGAG-3′; FBXW9, sense 5′-ACCCAGTGGTGGAAGAGAAG-3′ and antisense 5′-C AGCACTGAGTCAACGGAAG-3′.

## Immunoblot analysis

Proteins (GSCs or tumors) were extracted with RIPA buffer in the presence of complete protease inhibitors (Roche), separated by electrophoresis, transferred to PVDF membranes (Millipore), and blocked with 5% skim milk (BD). The membranes were incubated with primary antibodies against IGFBP5 (1:200, Santa Cruz), HER2 (1:500, Cell Signaling Technology), pHER2 Y1248 (1:500, R&D System), ROR1 (1:500, Cell Signaling Technology), pROR1 Tyr786 (1:500, Thermo Fisher), CREB (48H2) (1:500, Cell Signaling Technology), pCREB Ser133 (1:500, Cell Signaling Technology), IGF1R (1:1000, Sangon Biotech), pIGF1R Tyr1165/1166 (1:1000, Sangon Biotech), GAPDH (1:1000, Cell Signaling Technology), and β-actin (1:1000, Santa Cruz) overnight at 4 °C. The immunoreactive bands were visualized using peroxidase-labeled affinity-purified secondary antibodies (KPL) and the Amersham ECL Prime Western Blotting detection reagent (GE Healthcare) or Miracle-Star Western Blot detection system (iNtRON Biotechnology).

## Co-immunoprecipitation

For co-immunoprecipitation (Co-IP) experiments with endogenous proteins, X01 or 448 GSCs were washed twice with ice-cold PBS and fixed with 1 mM dithiobis (succinimidyl propionate, DSP; TOPSCIENCE) at room temperature for 30 min, followed by lysis with IP buffer (Thermo Fisher Scientific) in the presence of 1% protease inhibitor. Then, 4% protein lysate was used as input and the remaining cell lysates were mixed with protein A/G Dynabeads (Thermo Fisher Scientific) preincubated with normal rabbit IgG (Merck) or the indicated antibody at 4 °C overnight. For co-IP of exogenous proteins, 293 T cells were transfected with the indicated plasmids 48 h before harvest, fixation, and lysis. Again, 4% protein lysate was used as input and the remaining cell lysates were mixed with Flag/HA-Magnetic (Bimake) protein A/G Dynabeads at 4 °C overnight. The beads were then washed 5–7 times with IP buffer supplemented with 1% protease inhibitor. Immunoprecipitates were boiled for 10 min at 98 °C in protein loading buffer (EpiZyme) for further immunoblotting analysis.

## Human phospho-RTK array and phosphor-kinase array

To investigate the activation/phosphorylation of RTKs, we employed human phospho-RTK and phospho-kinase arrays, in which the capture and control antibodies were spotted in duplicate onto nitrocellulose membranes. To conduct a proteome profile array experiment, cell lysates were prepared from vehicle and human rIGFBP5 (100 ng/ml, 6 h)-treated 83 GSCs using lysis buffer containing protease and phosphatase inhibitors. For each cell lysate, 300–500 μg of total protein was analyzed using the phospho-RTK array (R&D Systems) and a phospho-kinase array (R&D Systems).

## Microscale thermophoresis (MST)

Purified recombinant His-IGFBP5 was labelled with a RED-tris-NTA protein labelling kit (NanoTemper) according to standard protocol. The protein and RED-tris-NTA 2nd generation dye was incubated for 30 min at room temperature in the dark, followed by centrifuge for 10 min, 15,000 $g$ at 4 °C and then transfer the supernatant to a fresh tube. The ligand protein ROR1 or HER2 was then diluted at a constant concentration with ddH$_2$O. Equal volumes of binding reactions solution were mixed by pipetting and loaded into the instrument (Monolith NT.115, NanoTemper, Germany). Measurements were performed as previously described[73].

## Mouse model

All animal experiments were conducted in accordance with protocols approved by the Institutional Animal Care and Use Committee of the National Cancer Center, Republic of Korea (NCC-17-402); and the Animal Care and Use Committee of Laboratory Animal Center, Henan University, China (HUSOM2021-0105). Mice were group-housed in ventilated cages under controlled temperature and humidity with a 12 h light-dark cycle. Every animal was randomized by body weight before the experiments. For the orthotopic mouse model, GSCs were first resuspended, and then transplanted into the left striatum of 5-week-old female BALB/c nude mice via stereotactic injection. The injection coordinates were 2.2 mm to the left of the midline and 0.2 mm posterior to the bregma at a depth of 3.5 mm. The mice were sacrificed with carbon dioxide when a 20% reduction in weight or severe neurological symptoms were observed, and the brain of each mouse was harvested and fixed in 4% paraformaldehyde for 24 hours at 4 °C for the further using. Their survival was analyzed using GraphPad PRISM software (version 7; GraphPad PRISM, La Jolla, CA, USA).

## Magnetic resonance imaging (MRI)

To capture the tumor structure in mouse brain, MRI experiments were performed with a Bruker BioSpec 7 T system (BioSpec 70/20 USR; Bruker, Germany) using a mouse brain array coil. Mice were isoflurane anesthetized (1–1.5%) and T2-weighted images were acquired using a RARE sequence: repetition time (TR) = 2500 ms; echo time (TE) = 35 ms; slice thickness = 0.7 mm; RARE factor = 8; number of average (NEX) = 4; acquisition matrix size = 256 × 192; and field of view (FOV) = 20 × 20 mm. The MRI data were processed with Para Vision 5.1 software (Bruker, Germany).

## Histology, Immunohistochemistry (IHC) and immuno-fluorescence staining

For histological observations, the brains were removed, fixed with 4% paraformaldehyde for 24 h at 4 °C, sectioned at a thickness of 4 μm using an essential microtome (Leica RM2125 RTS). For histological observations, haematoxylin and eosin (H&E) stains were carried out with 1% hematoxylin (DaKo) and 0.25% eosin (Merck). Stained sections were dried and mounted in an organic mounting medium.

Prior to immunohistochemical (IHC) staining for pHER2 Y1248 (1:100, R&D Systems), pROR1 Tyr786 (1:100, Thermo Fisher) and pCREB Ser133 (1:100, Cell Signaling Technology), the sections were subjected to an antigen retrieval process using citrate buffer (pH 6.0), and endogenous peroxidase was blocked by incubating with 3% hydrogen peroxide. The tissue sections were then incubated overnight at 4 °C in a humidified chamber with the primary antibody, diluted with antibody diluent buffer (IHC World). The tissue sections for 3,3′-diaminobenzidine (DAB) staining were developed using DAB (Vector Laboratories) as the chromogen.

For immunofluorescence staining against GFP, the tissue sections were subjected to an antigen retrieval process using citrate buffer (pH 6.0). Then incubated overnight at 4 °C with anti-GFP antibody (1:500, Abcam) in antibody diluent buffer (IHC World). Secondary staining was performed at RT for 2 h with fluorochrome-conjugated antibody (Alexa 568, 1:500, Thermofisher Scientific) and 4′,6-diamidino-2-phenylindole (DAPI; Sigma, 1:5,000). Fluorescence images were acquired with Zeiss LSM 780 confocal laser-scanning microscope (Carl Zeiss, Thornwood, NY, USA).

## Nanocapsule materials

The Cas9/sgIGFBP5 nanocapsules were prepared as in our previous work[37,38]. The sgRNA core of human IGFBP5 and the scramble control sgRNA were synthesized by Tsingke Biotechnology Co. Ltd. (Beijing, China). The sgRNA core sequences were as follows: sgRNA-IGFBP5, 5′-TACCGCGAGCAAGTCAAGAT-3′; sgRNA-Scramble, 5′-CACGGG-CAGCTTGCCGG-3′.

## Nanocapsule characterization

The size and zeta potential of the Cas9/sgRNA nanocapsules were determined at 25 °C using dynamic light scattering (DLS; Zetasizer Nano-ZS, Malvern Instruments). All measurements were conducted in triplicates. The structure of the nanocapsule was examined under a transmission electron microscope (TEM). Ten microliters of the nanocapsule solution were deposited onto a glow-discharged carbon-coated grid. After 10 min, the grid was washed with two drops of distilled water, and a drop of 1% uranyl acetate stain was then added to the grid. The grid was subsequently dried and visualized under a TEM (JEM-2010HT, Japan).

## Redox-responsive nature of Ang-SS-Cas9/sgIGFBP5

Ang-SS-Cas9/sgIGFBP5 nanocapsules were treated with 10 mM glutathione (GSH) mimicking an intracellular reduction environment of tumor. After 12 h of incubation, the morphology of the Ang-SS-Cas9/sgIGFBP5 nanocapsules was observed using TEM microscopy as described above.

## In vitro gene editing

X01 or 448 cells were seeded in 6-well plates (1 × 10$^5$ cells/well) and cultured for 24 h, cells were then incubated with Ang-SS-Cas9/sgIGFBP5 or Ang-SS-Cas9/sgScramble (Cas9: 20 nM) overnight,

followed by additional 1 ml fresh media added per well. The cells were then incubated at 37 °C for another 48 h. Genomic DNA was subsequently collected using Universal Genomic DNA Kit (CWBIO, China). The sgRNA-targeted genomic locus was amplified with High Fidelity Kod-Plus-Neo (TOYOBO, Japan). After purification by gel extraction (CWBIO, China), T7EI cleavage assays were conducted according to the manufacturer's instructions. In brief, 200 ng of the purified PCR product was denatured and reannealed in 2 μL of NEB buffer 2 (10×) as following: 95 °C, 5 min; 95-85 °C, −2 °C/s; 85-25 °C, −0.1 °C/s; then held at 4 °C. Then 1 μL of T7EI (NEB, USA) was added to the annealed PCR products and incubated at 37 °C for 1 h. Products were finally analyzed on 2% agarose gels and imaged with a Gel Doc gel imaging system (BioRad).

### Treatment with Ang-SS-Cas9/sgRNA nanocapsules
The antitumor efficacy of Ang-SS-Cas9/sgIGFBP5 was studied using an orthotopic X01-Luc bearing BALB/c nude mouse model. The mice were weighed and randomly divided into three groups: Ang-SS-Cas9/sgIGFBP5, Ang-SS-Cas9/sgScramble, and PBS. The mice were intravenously injected with nanocapsules or PBS every second day for a total of 5 times, started on day 10. Kaplan–Meier survival curves were determined for each treatment group ($n = 6$), and the body weights of mice were measured individually.

### High-throughput RNA sequencing (RNA-seq)
The library was prepared with 1 μg of total RNA for each sample by a TruSeq Stranded mRNA LT Sample Prep Kit (Illumina, Inc., San Diego, CA, USA). The first step in the workflow involves purifying the poly-A-containing mRNA molecules using poly-T-attached magnetic beads. Following purification, the mRNA was fragmented into small pieces using divalent cations under elevated temperatures. The cleaved RNA fragments were copied into first-strand cDNA using SuperScript II reverse transcriptase (Invitrogen) and random primers. This was followed by second-strand cDNA synthesis using DNA Polymerase I, RNase H and dUTP. These cDNA fragments were then subjected to an end-repair process, the addition of a single 'A' base, and adapter ligation. The products were then purified and enriched with PCR to create the final cDNA library. Indexed libraries were then paired-end sequenced with an Illumina HiSeq 4000 (Illumina, Inc., San Diego, CA, USA) at Macrogen Incorporated.

### RNA-seq data analysis
Raw fastq files were aligned to the human reference genome (hg19) using the STAR program (https://github.com/alexdobin/STAR). The aligned sam files were converted to bam files and sorted by coordinate using the Samtools program (http://www.htslib.org). Duplicate reads were removed using Picard (https://github.com/broadinstitute/picard). The gene expression values were calculated based on the read per kilobase of exon per million (RPKM) value. Genes that had greater than 30% missing values were discarded. The expression levels of the filtered genes were globally normalized with the Quantile normalization method using the R limma package. The enrichment score (ES) of four GBM subtypes[74] with each molecular signatures in 772 GSCs was analyzed using single sample gene set enrichment analysis (ssGSEA).

### Public data analysis
The TCGA LGG and GBM dataset was downloaded from the International Cancer Genome Consortium (ICGC) data portal (https://dcc.icgc.org/). The RNA-seq data were normalized based on the RPKM (reads per kilobase per million mapped reads) values, and the microarray data were globally normalized using the robust multiarray average (RMA) method. Statistical tests were performed using R language (https://www.r-project.org/), and graphs and heatmaps were prepared using the Microsoft Excel, R and MeV (http://www.tm4.org/mev.html) programs. The RNA-seq read distributions were visualized using the Integrative Genomics Viewer (IGV) program (http://software.broadinstitute.org/software/igv/). Functional annotations were performed using Gene Set Enrichment Analysis (GSEA, https://www.gsea-msigdb.org/gsea/index.jsp, GENT, http://medical-genome.kribb.re.kr/GENT/) and DAVID (https://david.ncifcrf.gov/) software.

### Statistics and reproducibility
The Kaplan-Meier method was used to plot survival curves. In the case of patients who were alive at the time of last follow-up, survival records were censored in our analysis. The statistical analyses were performed with GraphPad PRISM software (v8.0, CA, USA). In the case of mouse experiments, results of multiple datasets were compared by analysis of variance (ANOVA) using the log-rank (Mantel-Cox) test. The results of two-dataset experiments were compared using a two-tailed Student's $t$-test. $P$ values < 0.05 were considered statistically significant.

### Reporting summary
Further information on research design is available in the Nature Portfolio Reporting Summary linked to this article.

## Data availability
The sequencing data generated in this study have been deposited into the Sequence Read Archive (SRA) under the project accession number PRJNA732258. All data are available in the main article, supplementary information. The raw data supporting the finding from this study are provided in the Source Data file. Source data are provided with this paper.

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

## Acknowledgements

This work was supported by grants from the National Natural Science Foundation of China (82173228), the National Key Technologies R&D Program of China (2018YFA0209800), NHMRC Investigator Grant (GNT1194825), and the National Research Foundation of Korea (NRF) grant funded by the Korea government (MSIT) (2018R1C1B6004768, 2020R1C1C1011205, 2021R1A2C3013315, and 2021M3F7A1083230). We thank the animal molecular imaging facility at the National Cancer Center Korea for contributing to the in vivo animal molecular imaging. We also appreciate past J. Yin lab members for technical supports of experimental procedures.

## Author contributions

W.L., S.-M.P., B.S., J.B.P. and J.Y. designed the experiments. W.L., R.N., Y.Z., S.S.K., S.X., Q.Y., X.S., Z.Y., S.Z., D.Z., H.J.K., S.P., C.I.K. and J.Y. performed the experiments. S.-M.P. and H.K. performed the bioinformatics analysis. X.X., Y.L., H.W. and M.Z. provided intellectual support to this study. H.Y. provided clinical samples and advice. W.L., S.-M.P., Y.Z., S.S.K., X.X., B.S., J.B.P. and J.Y. wrote the manuscript.

## Competing interests

The authors declare no competing interests.
