## [Peer Review File · Nature Communications]

IGFBP5 is an ROR1 ligand promoting glioblastoma invasion via ROR1/HER2-CREB signaling axisEditorial Note: Parts of this Peer Review File have been redacted as indicated to remove third-party material where no permission to publish could be obtained.

REVIEWER COMMENTS

Reviewer #1 (Remarks to the Author):

In the manuscript by Weiwei Lin et. al., RNA sequencing analysis revealed IGFBP5 was differentially expressed between invasive or non-invasive subsets of GBM primary cell cultures. They further show through knockdown and gain of function studies that IGFBP5 drives GBM cell invasion (in vitro and in vivo), potentially via ROR1/HER2 heterodimer formation and subsequent activation of CREB. in invasive patient derived GBM models. Finally, nanoparticle delivery of CRISPR/Cas9 targeting IGFBP5 was able to suppress GBM cell invasion and improve the survival of orthotopic tumor-bearing mice. Specific targeting of IGFBP5 via IGFBP5 has recently been shown as having an important role in GBM cell invasion (Dong et. al, 2020). Therefore, the novelty is primarily based on the proposed mechanism of action and the in vivo experiments. Yet, there are numerous areas that need to be addressed to support these conclusions.

Major Points

1) Mechanism of Action: The proposed ROR1/HER2/CREB mechanism is not fully supported by the data. If they are proposing that IGFBP5 mediated ROR1 activation leads ROR1-mediated HER2 phosphorylation, they need to demonstrate this through the introduction of various gain of function and loss of function mutants (e.g., kinase dead ROR1 and/or HER2). Similarly, they also need to show that downstream CREB is necessary and/or sufficient for ROR1/HER2 mediated invasion (e.g., gain of function CREB under loss of function ROR1/HER2). As of now, the data support changes in activity and function of these players correlate with invasion and not necessarily that these factors are all mechanistically linked. Finally, the targets of CREB from Fig 5 have not been confirmed for their functional relevance.

2) Additional in vivo models to support conclusions: The authors should show efficacy of the NPs in at least one additional sensitive model. Moreover, the authors should show whether targeting IGFBP5 is ineffective in a model predicted to be insensitive to this approach.

Other points

1) The use of glioma stem cells for the patient derived neurosphere culture is not appropriate. The simple formation of neurosphere cultures cannot by itself define a putative "glioma stem cell"

2) It is unclear what is the difference in the point from Fig 6 in relation to Fig 2. Are these not all patient-derived cells?

3) Related to above, are there models predicted to be insensitive to targeting IGFBP5. For example, shRNAs against IGFBP5 in the other 2 models, 131 and 83 further reduce invasiveness and improve survival?

4) The authors identified DEGs by a 4-fold change in RPKM. However, directly comparing RPKM is inaccurate for cross-sample comparisons and the determination of DEGs. The authors should use a statistical method of identifying DEGs such as DESeq.

Zhao, Y., Li, MC., Konaté, M.M. et al. TPM, FPKM, or Normalized Counts? A Comparative Study of Quantification Measures for the Analysis of RNA-seq Data from the NCI Patient-Derived Models Repository. *J Transl Med* 19, 269 (2021). doi:10.1186/s12967-021-02936-w

Zhao S, Ye Z, Stanton R. Misuse of RPKM or TPM normalization when comparing across samples and sequencing protocols. *RNA*. 2020;26(8):903-909. doi:10.1261/rna.074922.120

5) For figure 1c, why was the comparison between DEGs identified by the authors' RNAseq analysis compared with TCGA low-grade glioma dataset but not the high-grade glioma dataset

6) For figure 1e, what is the mRNA expression fold change relative to?

7) For figure 1f, possibly perform the same statistical comparison performed in 1b and 1e by comparing the non-invasive vs the invasive models?

8) For figure 2, did the silencing of IGFBP5 with shRNAs also affect cell growth in vitro? Could that have also contributed to the differences seen in invasiveness? Some of this has been shown prior in: Dong, C., Zhang, J., Fang, S. et al. IGFBP5 increases cell invasion and inhibits cell proliferation by EMT and Akt signaling pathway in Glioblastoma multiforme cells. *Cell Div* 15, 4 (2020). <https://doi.org/10.1186/s13008-020-00061-6>

9) For figure 4, do naturally invasive models (e.g., 448 and X01) have higher basal ROR1, HER2,

and CREB activation relative to non-invasive models not stimulated by IGFBP5?

10) For figure 4, what are the quantifications of the RTK and kinase array?

11) For figure 7, does the Cas9/CRISPR reduce protein production?

12) Figure 3j, all 6 mice with IGFBP5 died on the same day? Very unlikely to see all mice reach endpoints on the exact same day. How were endpoints established for this experiment?

Reviewer #2 (Remarks to the Author):

This MS by Lin et al. IGFBP5, a secreted protein known for its role of binding to IGFs, can act as a ROR1 ligand and promote glioblastoma stem cell (GSC) invasion. The authors provided good evidence indicating that IGFBP5 expression affects GSC migration/invasion. This finding is not surprising because it has been reported by several groups that IGFBP5 can stimulate the migration/invasion of normal and tumor cells, including glioblastoma cells. What is potentially new is the idea that IGFBP5 binds to ROR1, causes ROR1/HER2 heterodimer formation, and increases CREB-mediated gene expression. If proven correct, this would be a new and interesting finding. There are, however, a number of issues and major concerns. To convincingly support the notion that IGFBP5 causes ROR1/HER2 heterodimer formation and increases gene expression and cell invasion vis CREB, more rigorous, complimentary genetic (dominant negative & constitutive active) and pharmacological experiments are needed.

Major comments:

1) In this MS, critical technical details are often missing, making it difficult to assess the quality of the data. To give a few examples, I cannot find information on how different GSC cells (448, X01, 83, are these mouse #?) were isolated and what were the objective criteria to classify them into invasive vs. non-invasive GSCs? There is no description in the Methods section about the RNA-seq. The raw data of RNA-seq data set were not available. In vivo xenograft data in Fig. 3i. There is no description in the Methods section about the xenograft experiment. It is unclear what were the biological replica. The quantification data is not presented.

2) The conclusion that IGFBP5 acts as ROR1 ligand was based on the data shown in Fig. 4. Unfortunately, all biochemical assays lacked quantified results (e.g., Fig. 4a-4h). Were these experiments performed only one time? Quantitative and reproducible biochemical data are needed. Same question with Fig. 7k.

3) It is also troublesome that contradictory results (e.g., Fig. 4d, different effects on ROR1 levels; Fig. 4e different effects ROR1 levels) were ignored. How was the staining specificity validated?

4) Fig. 4k, Fig. 6e, 6g: Only one section image is shown per group. What were the biological replica? Are these statistically different changes? How was the staining specificity validated? Are these GSC cells and how do you know?

5) To convincingly support the notion that IGFBP5 causes ROR1/HER2 heterodimer formation and increases gene expression and cell invasion vis CREB, more rigorous, independent genetic (dominant negative & constitutive active) and pharmacological experiments are needed. It will also be critical to demonstrate the specificity of IGFBP5 binding and the Kd value.

6) Fig. 7m – I could not see much difference between the scrambled control and sgIGFBP5 group. Yet, it was stated that in vivo nano capsule-mediated IGFBP5 disruption dramatically increase mouse survival. Did I miss something? Did the authors actually detect any indel in the IGFBP5 gene in these mice?

Minor comments:

1) Fig. 7. Did the Cas9/sgIGFBP5 treatment actually reduced IGFBP5 levels in X01 and 448 cells? Is yes, how much?

2) Fig. 7k, what was analyzed here, GSCs or the entire tumor?

3) The effects of IGFBP5 on cell migration/invasion have been reported since the late 1990s. These papers should be cited.

4) For in vitro invasion assays using shRNA-mediated knockdown, the possible off-target effects were not addressed. A simple experiment would be to test if adding back rIGFBP5 to the knocked down cells can rescue the cell invasion/gene expression.

Reviewer #3 (Remarks to the Author):

This interesting study shows a clear effect of IGFBP-5 in promoting GBM stem cell invasion and provides a plausible mechanism involving ROR1, HER2, and CREB activation, together with the CREB-dependent expression of ETV5 and FBXW9. Strengths: In general, the experiments are clearly presented and well interpreted, and the development of an IGFBP-5-targeting nanocapsule that significantly impairs GBM tumor growth and prolongs survival in mice with orthotopic GBM tumors is very impressive. Weaknesses: Although IGFBP-5 is immunoprecipitated using an anti-ROR1 antibody its direct binding to ROR1, as proposed, is not demonstrated, and the existence of a functional HER2-ROR1-IGFBP-5 complex is unclear since it is not precipitated using an anti-HER2 antibody (see specific comment #4). Therefore the precise mechanism of IGFBP-5 action is in some doubt, and a role for IGF1R phosphorylation is not explicitly excluded by the single negative phosphoarray experiment (see specific comment #7). Nevertheless this work adds significantly to current knowledge of the role of IGFBP-5 in GBM.

Specific comments

1. Line 116: "a small number of studies". Since high IGFBP-5 expression in GBM has been reported previously (as far back as 2006, doi: 10.1177/153303460600500303), some of these prior studies should be cited here.
2. Fig. 4c. The pROR1 and pCREB responses seem weaker in 131 cells than 83 cells. These data would benefit from quantitation of phospho/total ratios from several independent assays, and statistical evaluation.
3. Fig. 4f, g: Some IP bands are stronger than the input. State what fraction of the total lysate was analyzed in the input lanes.
4. Lines 185-186. If a ROR1-IGFBP-5 complex interacts with HER2, IP with a HER2 antibody should precipitate both ROR1 and IGFBP-5, but this is not seen in Fig. 4f. One interpretation is that IGFBP-5 can bind to ROR1, but this binary complex does not interact with HER2. This is different from the model shown in Fig. 4m. Further interpretation of the coIP data is required, as well as some coIPs using anti-IGFBP-5 as the precipitating Ab.
5. Fig. 4h suggests that both ROR1 and HER2 have a role in CREB phosphorylation, but it provides no information about the role of IGFBP-5 in this process. Neither do Figs. 4i and j provide information about the role of IGFBP-5. So line 190, ".. implying that IGFBP-5 promotes ...", overinterprets the data.
6. Lines 193-194. "... HER2 knockdown did not alter ROR1 phosphorylation", etc. It is difficult to be confident about this statement in the absence of a quantitative analysis of Fig. 4h. Phospho/total data should be analyzed for several repeat experiments.
7. Lines 332-333. Rather than introducing a new result in the Discussion, the authors should indicate the position of pIGF1R in Fig. 4a to illustrate that IGF1R phosphorylation is not promoted by IGFBP-5. Given that IGF1R phosphorylation may have occurred transiently and not been detected on this array, it would be informative to check the acute pIGF1R response to IGFBP-5 (e.g. a time-course starting at 5 or 10 min) by immunoblot and to ensure that the addition of an IGF1R tyrosine kinase inhibitor such as NVP-AEW541 did not block any downstream effects of IGFBP-5.

Minor point

Line 112: "... adverse roles": I believe the authors mean "diverse" roles.

Reviewer #4 (Remarks to the Author):

Lin et al. report a comprehensive study into the role of Insulin-like Growth Factor-Binding Protein 5 (IGFBP5) on glioblastoma infiltration in orthotopic mouse models, in particular, as it relates to GSCs. In a GSC-derived brain tumor model, they conducted RNA seq analysis and found that IGFBP5 was differentially expressed between invasive or non-invasive subtypes. The authors then engaged in detailed mechanistic studies to establish the importance of IGFBP5 for GSC invasion. Finally, they report data suggesting that the nanoparticle-based delivery of CRISPR/Cas9-based

IGFBP5 gene editing results in impressive therapeutic benefits, such as significant increase in survival in a GBM mouse model.

Overall, this is a large research study with an impressive attention to details, in particular on the mechanistic aspects. The initial hypothesis related to IGFBP5 is motivated by a comparative study between invasive and non-invasive GBM subtypes. They follow up with both silencing and knock down experiments that clearly establish the link between IGFBP5 and GSC invasion. While this study has clearly merits – in particular as it relates to GBM pathobiology - there are also significant shortcomings that dampen the potential impact. The following aspects should be addressed prior to publication:

- 1) IGFBP5 is a member of the IGFBP family, and, contrary to what is suggested in the introduction and the discussion sections, a significant bulk of literature clearly links these proteins to GBM infiltration and poor clinical prognosis. The same is also true for IFGBP5, see for example (references not cited by authors): <https://www.ncbi.nlm.nih.gov/pmc/articles/PMC7885861/> or <https://celldiv.biomedcentral.com/articles/10.1186/s13008-020-00061-6>.
- 2) There are fundamental limitations of using orthotopic GBM systems to model GBC invasion. For this particular study, genetically engineered spontaneous glioblastoma mouse models would be a better choice. The authors should at least discuss the limitations of the model they selected and how this could affect the scientific premise of their study.
- 3) IGFBP5 is overexpressed in a wide range of tissues, and it is unclear how CRISPR/Cas9-based IGFBP5 gene editing would affect these tissues systematically, see, e.g., <https://www.proteinatlas.org/ENSG00000115461-IGFBP5/tissue>. This seems an important aspect that should be addressed by the authors.
- 4) The nanoparticle delivery system provides clear benefits in survival, but a more complete characterization would be warranted. What is the biodistribution of the nanoparticles? Where are they accumulating? If they go into liver and spleen, is there any toxicity or off-target activity? In particular, it would be interesting to know if they accumulate in any of the tissues that overexpress IGFBP5.
- 5) The CRISPR/Cas9-based IGFBP5 gene editing experiments have only been done in the X1 subtype, but at least one non-invasive subtype should be included for comparison.
- 6) What was the dose used for the CRISPR/Cas9-based IGFBP5 gene editing experiments? A dose escalation study should be included. What is the fraction of the total dose delivered to the brain tumor? How much was delivered to the brain? What is the GBM/brain ratio?
- 7) The authors claim for their CRISPR-Cas9 system “a much preferable outcome on tumor growth as well as mouse expectancy, comparing to the well-known siRNA-based therapies targeting the conventional GBM targets (i.e., PLK134 or STAT3)”. However, when consulting their cited references, it appears as the opposite is the case, i.e., their survival results, also significant, don't match up with the one achieved by the other therapies.

<Responses to reviewers' comments>

Reviewer #1 (Remarks to the Author):

In the manuscript by Weiwei Lin et. al., RNA sequencing analysis revealed IGFBP5 was differentially expressed between invasive or non-invasive subsets of GBM primary cell cultures. They further show through knockdown and gain of function studies that IGFBP5 drives GBM cell invasion (in vitro and in vivo), potentially via ROR1/HER2 heterodimer formation and subsequent activation of CREB. in invasive patient derived GBM models. Finally, nanoparticle delivery of CRISPR/Cas9 targeting IGFBP5 was able to suppress GBM cell invasion and improve the survival of orthotopic tumor-bearing mice. Specific targeting of IGFBP5 via IGFBP5 has recently been shown as having an important role in GBM cell invasion (Dong et. al, 2020). Therefore, the novelty is primarily based on the proposed mechanism of action and the in vivo experiments. Yet, there are numerous areas that need to be addressed to support these conclusions.

Major Points

1) Mechanism of Action: The proposed ROR1/HER2/CREB mechanism is not fully supported by the data. If they are proposing that IGFBP5 mediated ROR1 activation leads ROR1-mediated HER2 phosphorylation, they need to demonstrate this through the introduction of various gain of function and loss of function mutants (e.g., kinase dead ROR1 and/or HER2).

Response: We truly appreciate the reviewer's comment. As suggested by reviewer, we have strengthened ROR1/HER2/CREB mechanism by investigating the role of each molecule in GSCs invasion by i) knocking down HER2 using shRNA or pharmacological inhibitor, Irbinitinib (Response Fig. 1); ii) knocking down ROR1 using shRNA (Response Fig. 2a-d); iii) modulating ROR1 activation by overexpressing ROR1 wild type (ROR1 WT) and kinase domain deletion type (ROR1 mut) (Response Fig. 2e-l) in GSCs, including the rescue of ROR1 WT or ROR1 mut expression in IGFBP5 knockdown GSCs. The results are as follows:

- i) To determine the functional role of HER2 and ROR1 in GSC invasion, we silenced HER2 and ROR1 in invasive GSCs 448 and X01 with shRNA. As shown in response Fig. 1 and 2, both CREB phosphorylation and the invasive capacity of GSCs 448 and X01 were downregulated by knocking down HER2 (Response Fig. 1a-d) and ROR1 (Response Fig. 2a-d).
- ii) The pharmacological HER2 inhibitor, Irbinitinib reduced the downstream CREB phosphorylation (Response Fig. 1e, f) and the GSCs invasion (Response Fig. 1g, h) as well.
- iii) We also modulated ROR1 activation by overexpressing ROR1 wild type (ROR1 WT) and kinase deletion type (ROR1 mut) in the non-invasive GSCs 83 and 131. Our results showed

that overexpression of ROR1 (ROR1 WT) increased phosphorylation of CREB (Response Fig. 2e, f) and CREB downstream genes including ETV5 and FBXW9 (Response Fig. 2g-j), but not in ROR1 mut. More importantly, overexpression of ROR1 (ROR1 WT) but not ROR1 mut rescued knockdown of IGFBP5 mediated repression of HER2 and CREB phosphorylation (Response Fig. 2k) and GSCs invasion (Response Fig. 2l).

Taken together, these results demonstrated the essential role of IGFBP5/ROR1/HER2-CREB signaling axis in resulting GSC invasion.

Response Fig. 1 (Revised Fig. 4k and New Extended Data Fig. 4d, f-i). Inhibition of HER2 expression suppresses GSCs invasion and the downstream CREB signaling. **a-b**, Immunoblot (IB) analysis of pROR1, pHER2, pCREB, ROR1, HER2, and CREB in 448 (**a**) and in X01 (**b**) GSCs infected with shCtrl, shHER2-1, or shHER2-2 lentivirus. GAPDH was used as a loading controls. **c**, Invasion assays of 448 GSCs infected with shCtrl, shHER2-1, or shHER2-2 lentivirus. Images (left) taken after 24 h of invasion are representative of three independent experiments (scale bar, 100 µm), and the graph (right) shows the mean number of invasive cells ± SEM (n = 3). ** $P < 0.01$, *t*-test. **d**, Invasion assays of X01 GSCs infected with shCtrl, shHER2-1, or shHER2-2 lentivirus. Images (left) taken after 24 h of invasion are representative of three independent experiments (scale bar, 100 µm), and the graph (right) shows the mean number of invasive cells ± SEM (n = 3). *** $P < 0.001$, *t*-test. **e-f**, IB analysis of pROR1, pHER2, pCREB, ROR1, HER2, and CREB in 448 (**e**) and in X01 (**f**) GSCs after 24 h Irbitinib treatment. GAPDH was used as a loading controls. **g**, Invasion assays of 448 GSCs treated with Voronoi with 1 µM, 2 µM or vehicle control for 24 h. Images (left) taken after 24 h of invasion are representative of three independent experiments (scale bar, 100 µm), and the graph (right) shows the mean number of invasive cells ± SEM (n = 3). *** $P < 0.001$, *t*-test. **h**, Invasion assays of X01 GSCs treated with Irbitinib with 1 µM, 2 µM or vehicle control for 24 h. Images (left) taken after 24 h of invasion are representative of three independent experiments (scale bar, 100 µm), and the graph (right) shows the mean number of invasive cells ± SEM (n = 3). *** $P < 0.001$, *t*-test.

Response Fig. 2 (Revised Fig. 4l and New Extended Data Fig. 4e, f-l). ROR1 regulates GSCs invasion and the downstream CREB signaling. **a-b**, IB analysis of pROR1, pHER2, pCREB, ROR1, HER2, and CREB in 448 (**a**) and in X01 (**b**) GSCs infected with shCtrl, shROR1-1, or shROR1-2 lentivirus. GAPDH was used as a loading control. **c-d**, Invasion assays of 448 (**c**) and X01 (**d**) GSCs infected with shCtrl, shROR1-1, or shROR1-2 lentivirus. Images (left) taken after 24 h of invasion are representative of three independent experiments (scale bar, 100 μ m), and the graph (right) shows the mean number of invasive cells \pm SEM (n = 3). ** $P < 0.01$, *** $P < 0.001$, *t*-test. **e-f**, IB analysis of pROR1, pHER2, pCREB, ROR1, HER2, and CREB in 83 (**e**) and in 131 (**f**) GSCs infected with vector control, ROR1 WT, or ROR1 mut lentivirus. GAPDH was used as a loading control. **g-h**, RT-qPCR analysis of CREB targets ETV5 (**g**) and FBXW9 (**h**) expression in 83 GSCs infected with vector control, ROR1 WT, or ROR1 mut lentivirus. Data are presented as mean \pm SEM (n = 3). * $P < 0.05$, *t*-test. **i-j**, RT-qPCR analysis of CREB targets ETV5 (**i**) and FBXW9 (**j**) expression in 131 GSCs infected with vector control, ROR1 WT, or ROR1 mut lentivirus. Data are presented as mean \pm SEM (n = 3). * $P < 0.05$, ** $P < 0.01$, *t*-test. **k**, IB analysis of pROR1, pHER2, pCREB, ROR1, HER2, and CREB in X01 GSCs infected with shCtrl, shIGFBP5-1 lentivirus, and then infected with vector control, ROR1 WT, or ROR1 mut lentivirus. GAPDH was used as a loading control. **l**, Invasion assays of X01 GSCs infected with shCtrl, shIGFBP5-1 lentivirus, and then infected with vector control, ROR1 WT, or ROR1 mut lentivirus. Images (left) taken after 24 h of invasion are representative of three independent experiments (scale bar, 100 μ m), and the graph (right) shows the mean number of invasive cells \pm SEM (n = 3). ** $P < 0.01$, *t*-test.

Similarly, they also need to show that downstream CREB is necessary and/or sufficient for ROR1/HER2 mediated invasion (e.g., gain of function CREB under loss of function ROR1/HER2). As of now, the data support changes in activity and function of these players correlate with invasion and not necessarily that these factors are all mechanistically linked.

Response: We thank the reviewer for this comment. As the reviewer suggested, we first overexpressed CREB in non-invasive GSC and investigated the role of CREB in regulating GSC invasion. Next, we also investigated their function by overexpressing CREB under loss-of-function (LOF) of IGFBP5,

ROR1 or HER2. Immunoblot analysis verified a successful ectopic CREB overexpression in non-invasive GSCs (83 and 131) (Response Fig. 3a-b). CREB overexpression significantly increased downstream genes *ETV5* and *FBXW9* expression, and the invasive capacity of GSCs (Response Fig. 3c-h).

To investigate the downstream of CREB in IGFBP5-ROR1/HER2 signaling pathway, we ectopically overexpressed CREB in IGFBP5 knockdown GSCs, and CREB sufficiently restored GSC invasion (Response Fig. 3i, j). Furthermore, overexpression of CERB also rescued HER2 or ROR1 knockdown-mediated repression of GSC invasion (Response Fig. 4a-h). These results indicated that CREB is a key molecule in IGFBP5-mediated ROR1/HER2 signaling axis.

Response Fig. 3 (Revised Extended Data Fig. 6d-n). Overexpression of CREB enhances GSCs invasion. **a-b**, IB analysis of pCREB, and CREB in 83 (**a**) and in 131 (**b**) GSCs infected with vector control, or CREB OE lentivirus. GAPDH was used as a loading control. **c-d**, RT-qPCR analysis of CREB targets *ETV5* (**c**) and *FBXW9* (**d**) expression in 83 GSCs infected with vector control, CREB OE lentivirus. Data are presented as mean ± SEM (n = 3). ** $P < 0.01$, t -test. **e-f**, RT-qPCR analysis of CREB targets *ETV5* (**e**) and *FBXW9* (**f**) expression in 131 GSCs infected with vector control, CREB OE lentivirus. Data are presented as mean ± SEM (n = 3). * $P < 0.05$, ** $P < 0.01$, t -test. **g-h**, Invasion assays of 83 (**g**) and 131 (**h**) GSCs infected with vector control, or CREB OE lentivirus. Images (left) taken after 48 h of invasion are representative of three independent experiments (scale bar, 100 μ m), and the graph (right) shows the mean number of invasive cells ± SEM (n = 3). * $P < 0.05$, t -test. **i**, IB analysis of pROR1, pHER2, pCREB, ROR1, HER2, and CREB in X01 GSCs infected with shCtrl, shIGFBP5-1 lentivirus, and then infected with vector control, CREB OE lentivirus. GAPDH was used as a loading control. **j**, Invasion assays of X01 GSCs infected with shCtrl, shIGFBP5-1 lentivirus, and then infected with vector control, CREB OE lentivirus. Images (left) taken after 24 h of invasion are representative of three independent experiments (scale bar, 100 μ m), and the graph (right) shows the mean number of invasive cells ± SEM (n = 3). ** $P < 0.01$, t -test.

Response Fig. 4 (New Extended Data Fig. 7). Overexpression of CREB rescues HER2 or ROR1 knockdown-mediated repression of GSC invasion. **a-b**, IB analysis of pROR1, pHER2, pCREB, ROR1, HER2, and CREB in 448 (**a**) and X01 (**b**) GSCs infected with shCtrl, shHER2-1 lentivirus, and then infected with vector control, CREB OE lentivirus. GAPDH was used as a loading control. **c-d**, Invasion assays of 448 (**c**) and X01 (**d**) GSCs infected with shCtrl, shHER2-1 lentivirus, and then infected with vector control, CREB OE lentivirus. Images (left) taken after 24 h of invasion are representative of three independent experiments (scale bar, 100 μ m), and the graph (right) shows the mean number of the invasive cells \pm SEM (n = 3). * $P < 0.05$, ** $P < 0.01$, *t*-test. **e-f**, IB analysis of pROR1, pHER2, pCREB, ROR1, HER2, and CREB in 448 (**e**) and X01 (**f**) GSCs infected with shCtrl, shROR1-1 lentivirus, and then infected with vector control, CREB OE lentivirus. GAPDH was used as a loading control. **g-h**, Invasion assays of 448 GSCs (**g**) and X01 (**h**) GSCs infected with shCtrl, shROR1 lentivirus, and then infected with vector control, CREB OE lentivirus. Images (left) taken after 24 h of invasion are representative of three independent experiments (scale bar, 100 μ m), and the graph (right) shows the mean number of invasive cells \pm SEM (n = 3). * $P < 0.05$, ** $P < 0.01$, *t*-test.

Finally, the targets of CREB from Fig 5 have not been confirmed for their functional relevance.

Response: As suggested by the reviewer, to examine the functional relevance of CREB downstream elements (ETV5 and FBXW9) in our study, we suppressed ETV5 and FBXW9 by employing small interfering RNA (siRNA). As shown in response Fig. 5, downregulated ETV5 or FBX9 significantly decreased invasion capacities in X01 and 448. Please see our new Extended Data Fig. 5 and the added accompanying description in the revised manuscript as below.

➤ Page 11 lines 14--18

“To determine whether ETV5 and FBXW9 were involved in regulating GSC invasion, we suppressed ETV5 and FBXW9 with small interfering RNA (siRNA) (Extended Data Fig. 5a-f).

Downregulated ETV5 or FBXW9 significantly decreased the invasion capacities in both X01 and 448 GSCs, indicating that ETV5 and FBXW9 regulate GSCs invasion (Extended Data Fig. 5b, c, e and f).”

Response Fig. 5 (New Extended Data Fig. 5). Inhibition of ETV5 and FBXW9 reduce GSCs invasion. **a**, RT-qPCR analysis of ETV5 (left) and FBXW9 (right) expression in 448 GSCs transfected with siRNA of ETV5 (left) and FBXW9 (right). Data are presented as mean \pm SEM ($n = 3$). ** $P < 0.01$, t -test. **b-c**, Invasion assays of 448 GSCs transfected with siRNA of ETV5 (**b**) and FBXW9 (**c**). Images (left) taken after 24 h of invasion are representative of three independent experiments (scale bar, 100 μ m), and the graph (right) shows the mean number of invasive cells \pm SEM ($n = 3$). *** $P < 0.001$, t -test. **d**, RT-qPCR analysis of ETV5 (left) and FBXW9 (right) expression in X01 GSCs transfected with siRNA of ETV5 (left) and FBXW9 (right). Data are presented as mean \pm SEM ($n = 3$). ** $P < 0.01$, *** $P < 0.001$, t -test. **e-f**, Invasion assays of X01 GSCs transfected with siRNA of ETV5 (**e**) and FBXW9 (**f**). Images (left) taken after 24 h of invasion are representative of three independent experiments (scale bar, 100 μ m), and the graph (right) shows the mean number of invasive cells \pm SEM ($n = 3$). *** $P < 0.001$, t -test.

2) Additional in vivo models to support conclusions: The authors should show efficacy of the NPs in at least one additional sensitive model. Moreover, the authors should show whether targeting IGFBP5 is ineffective in a model predicted to be insensitive to this approach.

Response: As suggested by the reviewer, we performed antitumor evaluation in a non-invasive 83 GSC mice model. The course of treatment was identical to that with X01 GSC model. The tumor bioluminescence intensity in mice treated with Ang-SS-Cas9/sgIGFBP5 increased dramatically, comparable to that with PBS and non-specific sequence control Ang-SS-Cas9/sgScramble (Response Fig. 6a). In addition, the body weight in all the three groups show similar reduction during the treatment period (Response Fig. 6b). Importantly, the Ang-SS-Cas9/sgIGFBP5 nanocapsules did not prolong the survival time of 83 GSCs-bearing mice (Response Fig. 6c). All these results demonstrated that Ang-

SS-Cas9/sgIGFBP5 was ineffective in a non-invasive mice model, supporting that the IGFBP5 plays a key role in the invasion of GSCs.

Accordingly, we have added relevant sentences in our revised manuscript as follows:

➤ Page 15 lines 19-- 23

“To further confirm the specificity of IGFBP5 regulatory role for invasive GSCs, we performed Ang-SS-Cas9/sgIGFBP5 in a non-invasive 83-Luc (stable luciferase-expressing 83 GSCs) GSC-bearing mice model. The results showed the identical outcomes of tumor growth and mouse survival among Ang-SS-Cas9/sgIGFBP5, Ang-SS-Cas9/sgScramble and PBS (Extended Data Fig. 9a-c).”

Since reviewer 4 (comment 7) asked us whether our CRISPR-Cas9 system is better than siRNA-based therapy, also, considering the high cost of Cas9 protein. We evaluated the therapeutic efficacy of Ang-SS-siIGFBP5 (identical nanocapsules loaded with siIGFBP5 instead of Cas9/sgIGFBP5) in another sensitive CSC-2 (CSC-2, invasive subtype) mice model (Response Fig. 7). Compared with vehicle (negative control siRNA), these siIGFBP5-loaded nanocapsules significantly reduced IGFBP5, and ETV5 expression levels (Response Fig. 7a, b) as well as CSC-2 GSCs invasion (Response Fig. 7c, d). Subsequently, whether Ang-SS-siIGFBP5 is capable of restricting GSC invasion and tumor progression were evaluated *in vivo*. Mice harboring orthotopic CSC-2 tumors were intravenously injected with siIGFBP5-loaded nanocapsules every second day for a total of 10 days. Both tumor invasion and volume were monitored. The Ang-SS-siIGFBP5 treated mice showed significantly lower GSC invasion and tumor volume, as observed by H&E staining (Response Fig. 7e). Furthermore, phosphorylation of IGFBP5 downstream proteins (i.e. HER2, ROR1, and CREB), was remarkably reduced in mice injected with siIGFBP5-loaded nanocapsules (Response Fig. 7e). Finally, *in vivo* nanocapsule-mediated siIGFBP5 knockdown significantly improved mouse survival (Response Fig. 7f).

Response Fig. 6 (New Extended Data Fig. 9). Effect of Cas9/sgIGFBP5 nanocapsules on tumorigenesis in 83-Luc GSCs-bearing mice. a, Luminescence images of orthotopic 83-Luc GSCs-bearing mice following treatment with Ang-SS-Cas9/sgIGFBP5, Ang-SS-Cas9/sgScramble or PBS. Mice were intravenously injected at a dose of 1.5 mg Cas9 equiv./kg on day 10, 12, 14, 16, and 18 post tumor implantations. **b**, Body weight changes in mice following different treatments. **c**, Kaplan-Meier survival curves of mice implanted with 1×10^4 83-Luc GSCs and treated with Ang-SS-Cas9/sgIGFBP5, Ang-SS-Cas9/sgScramble or PBS ($n = 7$). ns $P > 0.05$, log-rank test.

Response Fig. 7. Nanocapsule-mediated delivery of siIGFBP5 suppresses GSCs invasion. **a-b**, RT-qPCR analysis of IGFBP5 (**a**) and ETV5 (**b**) expression in CSC2 GSCs treated with Ang-SS-siIGFBP5 nanocapsules (1 ng/ml) or vehicle control for 72 h. Data are represented as mean \pm SEM ($n = 3$). ** $P < 0.01$, t -test. **c-d**, Invasion assay in CSC2 GSCs treated with Ang-SS-siIGFBP5 nanocapsules or vehicle control for 48 h. (**c**) Images are representative of three independent experiments (scale bar, 100 μ m), and (**d**) the graph shows the mean number of invasive cells \pm SEM ($n = 3$). ** $P < 0.01$, t -test. **e**, H&E staining of the whole brain and IHC analysis of pHER2, pROR1, and pCREB in orthotopic xenografts of CSC2 GSCs treated with Ang-SS-siIGFBP5 nanocapsules every other day for 10 days (scale bar, 100 μ m). **f**, Kaplan-Meier survival curves of mice implanted with 1×10^4 CSC2-Luc cells and treated with Ang-SS-siIGFBP5 nanocapsules (1 mg/kg) or vehicle control ($n = 5$). $P = 0.0035$, log-rank test.

Other points

1) The use of glioma stem cells for the patient derived neurosphere culture is not appropriate. The simple formation of neurosphere cultures cannot by itself define a putative “glioma stem cell”

Response: We thank the reviewer’s comment. The glioma stem cells (GSCs) cultured in neurosphere under neurobasal serum-free medium with bFGF and EGF is the most renowned cultivation method¹. We obtained GSCs (448, X01, 131 and 83) from our collaborative labs where those cells were established as GSCs in the following references. Additionally, we have detected the well-known stemness markers by immunoblot analysis. Comparing with the normal astrocyte cell line (HA1800), GSCs (448, X01, 131 and 83) overexpressed well-known stemness marker Nestin and downregulated astrocyte marker GFAP (Response Fig. 8). Accordingly, we have also added the related references in our revised manuscript.

1) X01 GSC was isolated by Dr. Naoki Oka in 2007, they had identified X01 GSCs express CSC

marker Nestin and exhibited extensive infiltrative capacity².

- 2) 448 GSC was dissociated by Dr. Do-Hyun Nam in 2013³, which was classified as proneural subtype.
- 3) 131 GSC was obtained from Dr. Do-Hyun Nam⁴, which was firstly published as mesenchymal type of glioblastoma by upregulation of the mesenchymal related genes.
- 4) 83 GSC was isolated by Dr. Ichiro Nakano, they published 83 GSC as mesenchymal subtype with enrichment of stemness marker CD44⁵. Though CD44 showed high expression level in HA1800, compared with proneural type GSCs (X01, 448), 83 and 131 exhibited higher CD44 expression (Response Fig. 8).

Response Fig. 8. Stemness marker expression in X01, 448, 83, and 131 GSCs. **a**, IB analysis of Nestin, GFAP, and CD44 protein expression in X01, 448, 83, 131 GSCs and astrocyte cell HA1800. GAPDH was used as a loading control. **b-d**, RT-qPCR analysis of Nestin (**b**), GFAP (**c**), CD44 (**d**) mRNA expression in X01, 448, 83, 131 GSCs and astrocyte cell HA1800. Data are presented as mean \pm SEM (n = 3). * $P < 0.05$, ** $P < 0.01$, *** $P < 0.001$, *t*-test.

2) It is unclear what is the difference in the point from Fig 6 in relation to Fig 2. Are these not all patient-derived cells?

Response: We thank the reviewer's comments. All of these 5 GSCs which we showed in Fig. 2 (448, X01, 131 and 83) and Fig. 6 (772) are patient-derived GSCs. The discovery of IGFBP5 differently

expressed between invasive and non-invasive GSCs was firstly identified using 448, X01, 131, and 83 GSCs which were generously donated by our collaborative labs. For the purpose in revealing that IGFBP5 is a universal regulator for invasive GBM, we applied the same assays to our newly isolated 772 GSC. We evaluated the stemness of 772 GSCs by confirming the notable high expression level of stemness markers CD133 and Nestin in comparison to the normal astrocyte cell line HA1800 (Response Fig. 9). As we described in the main text, IGFBP5 showed the same regulatory capacity towards GBM invasion in 772 and other GSCs (original Fig. 6 vs Fig. 2), further confirming the translation potential of IGFBP5 as a key regulator in GBM invasion.

Response Fig. 9. Stemness marker expression in 772 GSCs. a-b, RT-qPCR analysis of CD133 (a) and Nestin (b) mRNA expression in 772 GSCs and astrocyte cell HA1800. Data are presented as mean \pm SEM (n = 3). *** $P < 0.001$, t -test.

3) Related to above, are there models predicted to be insensitive to targeting IGFBP5. For example, shRNAs against IGFBP5 in the other 2 models, 131 and 83 further reduce invasiveness and improve survival?

Response: We thank the reviewer for raising this point. Following your comment, we firstly silenced IGFBP5 in non-invasive GSCs (83 and 131 GSCs) by shRNA and evaluated the invasive ability *in vitro*. The results showed that shRNA suppressed IGFBP5 mRNA expression in 83 and 131 GSCs (Response Fig. 10a, b), there was no significant differences on the invasiveness between shCtrl and shIGFBP5 in 83 or 131 GSCs (Response Fig. 10c, d), because of the low basal (Response Fig. 11) and secreted IGFBP5 level (Original Fig. 1f).

Furthermore, as described in your major point 2, 83 GSCs-bearing mice treated with Ang-SS-Cas9/sgIGFBP5 nanocapsules did not show any improvements in reducing tumors or prolonging the survival time (Response Fig. 6).

Response Fig. 10. Inhibition of IGFBP5 in non-invasive GSCs. **a-b**, RT-qPCR analysis of IGFBP5 mRNA expression in 83 (**a**) and 131 GSCs (**b**) infected with shCtrl, shIGFBP5-1, or shIGFBP5-2 lentivirus. Data are presented as mean \pm SEM (n = 3). ** $P < 0.01$, *t*-test. **c-d**, Invasion assays of 83 (**c**) and 131 (**d**) GSCs infected with shCtrl, shIGFBP5-1, or shIGFBP5-2 lentivirus. Images (left) taken after 48 h of invasion are representative of three independent experiments (scale bar, 100 μ m), and the graph (right) shows the mean number of invasive cells \pm SEM (n = 3). ns $P > 0.05$, *t*-test.

Response Fig. 11. IGFBP5 expression in GSCs. RT-qPCR analysis of IGFBP5 mRNA expression in X01, 448, 83 and 131 GSCs. Data are presented as mean \pm SEM (n = 3). *** $P < 0.001$, *t*-test.

4) The authors identified DEGs by a 4-fold change in RPKM. However, directly comparing RPKM is inaccurate for cross-sample comparisons and the determination of DEGs. The authors should use a statistical method of identifying DEGs such as DESeq.

Zhao, Y., Li, MC., Konaté, M.M. et al. TPM, FPKM, or Normalized Counts? A Comparative Study of Quantification Measures for the Analysis of RNA-seq Data from the NCI Patient-Derived Models Repository. *J Transl Med* 19, 269 (2021). doi:10.1186/s12967-021-02936-w

Zhao S, Ye Z, Stanton R. Misuse of RPKM or TPM normalization when comparing across samples and sequencing protocols. *RNA*. 2020;26(8):903-909. doi:10.1261/rna.074922.120

Response: We would like to thank the reviewer's constructive suggestion. As requested, we performed DEG selection again with statistical methods, DESeq2 and edgeR in the TCC package (Bioconductor), and selected 1724 DEGs (Response Fig. 12). Comparing the DEGs with the previous DEGs in our manuscript, we found that most of the previous DEGs by a 4-fold change in RPKM were included in

newly selected 1724 DEGs except 7 genes (ZNF714, GAREM2, KIF5C, C11orf96, CCDC152, ANKRD44, SDHAF3), and IGFBP5 is ranked on top 137th DEGs. Applying the statistical method and excluding the 7 genes, we strictly selected DEGs by 4-fold change and FDR < 0.1 as the top-ranked DEGs. Following your suggestion, we excluded the 7 genes (ZNF714, GAREM2, KIF5C, C11orf96, CCDC152, ANKRD44, SDHAF3) from 299 DEGs by a 4-fold change in RPKM from Fig. 1c (Fig. 1c left-up side), so that 292 DEGs were selected by both methods (4-fold change in RPKM and FDR < 0.1). Indicating that 292 Genes can be used for the selection of top-ranked DEGs (we revised the 299 Genes to be 292 Genes in revised Fig. 1c). The analysis results suggest that IGFBP5 is ranked on top 27th in these 292 DEGs, which as the top 1 in the 36 DEGs candidates in Fig. 1c.

➤ Page 5 lines 13--16

“Differentially expressed genes (DEGs) between the invasive and non-invasive GSCs were identified using a criterion combined with a minimum four-fold change in the number of mapped reads per kilobase of transcript per million reads mapped (RPKM) and FDR < 0.1 in the DESeq2 and edgeR of TCC package²⁰ (Fig. 1c).”

Response Fig. 12. Differentially expressed genes (DEGs) selection by DESeq (MA plot). DEGs selection was used DESeq2 and edgeR in TCC package (Bioconductor). 1724 DEGs are selected.

5) For figure 1c, why was the comparison between DEGs identified by the authors' RNAseq analysis compared with TCGA low-grade glioma dataset but not the high-grade glioma dataset

Response: We thank the reviewer for this comment. For the high-grade glioma, we performed prognosis analyses with GBM microarray from the TCGA dataset. The results demonstrated that patients with high IGFBP5 expression showed significantly poorer prognosis than patients with low IGFBP5 expression (Response Fig. 13a). Next, we analyzed the Differentially Expressed Genes (DEGs) regulating GSC invasion using our RNA-seq and TCGA GBM microarray dataset. DEGs between the

invasive and non-invasive GSCs were identified using a criterion combined with a minimum four-fold change in the number of mapped reads per kilobase of transcript per million reads mapped (RPKM) and $FDR < 0.1$ in the DESeq2 and edgeR analysis (Response Fig. 13b). A total of 9 candidate DEGs regulating GSC invasion were identified from our established 292 DEGs (Please see revised Fig. 1c) based on corresponding clinical information for 309 genes obtained from the TCGA GBM microarray dataset, and these DEGs were significantly associated with poor survival (hazard ratio (HR) > 1 , $P < 0.01$, Cox proportional hazards analysis) (Response Fig. 13b). Among them, IGFBP5 showed the 2nd highest differential expression between invasive and non-invasive GSCs, while just S100B (a well-known astrocyte marker) was ahead of IGFBP5 (Response Fig. 13b).

Taken together, we found that IGFBP5 plays a potential oncogenic role in both invasive high-grade and low-grade glioma (see details in revised Fig. 1 in our manuscript). Since that invasion is considered the most common and pivotal event across the entire tumor development from low-grade to high-grade glioma⁶. We believe that finding out the key driver genes or biomarkers for invasion at the early time point of low-grade glioma is important to glioma therapy and translational research.

We revised Fig. 1h and described this part in the revised manuscript.

Please see page 7, lines 1--6 in our revised manuscript.

Response Fig. 13. Analysis using the GBM microarray from the TCGA dataset. **a**, Kaplan-Meier survival analysis of GBM patients with higher or lower IGFBP5 expression based on the median expression in the TCGA dataset. $P = 0.0031$, log-rank test. **b**, Venn diagram of Differentially-Expressed Genes (DEGs) identified in our RNA-Seq analysis and the TCGA-GBM dataset (left) and the expression of the candidate genes in our RNA-Seq analysis (right).

6) For figure 1e, what is the mRNA expression fold change relative to?

Response: We calculated the relative mRNA expression of fold change (\log_2) of IGFBP5 gene by RT-PCR, normalized by house-keeping control gene β -actin. The related mRNA expression (fold change) of IGFBP5 in X01, 131, and 83 were compared with that of 448 GSC.

7) For figure 1f, possibly perform the same statistical comparison performed in 1b and 1e by comparing the non-invasive vs the invasive models?

Response: We thank the reviewer for this comment. As suggested, we revised Fig. 1f using the same statistical comparison performed in Fig. 1b and 1e (Response Fig. 14).

Response Fig. 14 (Revised Fig. 1f). Analysis of secreted IGFBP5 level. ELISA analysis of secreted IGFBP5 protein in conditioned media (CM) from GSCs cultured for 3 days. Data are presented as mean \pm SEM (n = 3). * $P < 0.05$, t -test of the significant differences between invasive GSCs (448, X01) and non-invasive GSCs (131, 83).

8) For figure 2, did the silencing of IGFBP5 with shRNAs also affect cell growth in vitro? Could that have also contributed to the differences seen in invasiveness? Some of this has been shown prior in: Dong, C., Zhang, J., Fang, S. et al. IGFBP5 increases cell invasion and inhibits cell proliferation by EMT and Akt signaling pathway in Glioblastoma multiforme cells. Cell Div 15, 4 (2020). <https://doi.org/10.1186/s13008-020-00061-6>

Response: We thank the reviewer for this nice comment. In general, the GSC culture condition of invasive assay is different from cell proliferation assay. Complement medium DMEM/F12 supplemented with growth factors (EGF, bFGF, B27) were used for the cell proliferation assay. DMEM/F12 without any growth factors, that provide insufficient supplements for the GSC growth, was used in invasive assay to avoid the cell growth contributed invasive differences.

As for your concern that Dong et al.⁷ demonstrated that knockdown IGFBP5 inhibited GBM cell invasion. Actually, their results that IGFBP5 knockdown slightly increased cell growth suggested the effect of IGFBP5 on cell proliferation does not contribute to cell invasion.

9) For figure 4, do naturally invasive models (e.g., 448 and X01) have higher basal ROR1, HER2, and

CREB activation relative to non-invasive models not stimulated by IGFBP5?

Response: Following your comment, we performed immunoblot to detect ROR1, HER2, and CREB expression in the 4 GSCs (448, X01, 131 and 83) with a normal astrocyte cell line HA1800 as a control (Response Fig. 15). The result of immunoblot analysis showed the expression of CREB and ROR1 were enriched in invasive GSCs (448, X01) than non-invasive GSCs (131, 83), but the expressions of total HER2 did not show same pattern with the subtype of GSCs.

Response Fig. 15. ROR1, HER2, and CREB basic expression in GSCs. IB analysis of ROR1, HER2, and CREB expression level in astrocyte (HA1800) cell, X01, 448, 83 and 131 GSCs. GAPDH was used as a loading control.

10) For figure 4, what are the quantifications of the RTK and kinase array?

Response: We quantified the spot intensity and normalized it through the reference spots using ImageJ software. After normalized, the expression (mean pixel density) differences of phosphorylated HER2 and ROR1 were shown as rIGFBP5/Vehicle ratio (Response Fig. 16a).

Similarly, in the kinase array, after normalized the 2 reference spots, the expression (mean pixel density) differences of phosphorylated CREB, ERK1/2, ChK2, and HSP60 were shown as rIGFBP5/Vehicle ratio (Response Fig. 16b).

Taken together, all the candidates of RTK (HER2, ROR1) and kinase (CREB, ERK1/2, ChK2, HSP60) we selected in rIGFBP5 treated 83 GSCs showed a significant difference compared to Vehicle.

Response Fig. 16. Quantification of RTK/Kinase Array. Quantitative analysis of the mean pixel density ratio of vehicle control (Veh, blue) to rIGF (100ng/ml recombinant IGFBP5 protein treated 6 h, red) for RTK array from manuscript Fig.4a (a), for Kinase array from Fig.4b (b) in 83 GSCs by using ImageJ software. * $P < 0.05$, ** $P < 0.01$, *t*-test. Refer Spots were used as loading controls.

11) For figure 7, does the Cas9/CRISPR reduce protein production?

Response: Yes, Cas9/sgIGFBP5 not only reduced the IGFBP5 protein expression in X01 and 448 GSCs (Response Fig. 17), but also depressed IGFBP5 expression level in the xenograft tumor tissue (Original Fig. 7k). Successful IGFBP5 knockdown was validated by ELISA analysis after Cas9/sgIGFBP5 treatment (Response Fig. 17a, b). Immunoblot (IB) results showed that knockdown of IGFBP5 decreased pROR1, pHER2 and pCREB expression level (Response Fig. 17c, d).

Response Fig. 17. Cas9/sgIGFBP5 treatment reduces IGFBP5 expression. a-b, ELISA analysis of IGFBP5 in conditioned media (CM) from 448 (a) and X01 (b) GSCs after Cas9/sgIGFBP5 treatment. Data are presented as mean \pm SEM (n = 3). * $P < 0.05$, ** $P < 0.01$, *t*-test. c-d, IB analysis of IGFBP5, pROR1, pHER2, pCREB, ROR1, HER2, and CREB in 448 (c) and X01 (d) GSCs after Cas9/sgIGFBP5 treatment. GAPDH was used as a loading control.

12) Figure 3j, all 6 mice with IGFBP5 died on the same day? Very unlikely to see all mice reach endpoints on the exact same day. How were endpoints established for this experiment?

Response: We thank the reviewer for this comment. To explain the reasons why all mice reach endpoints on the exact same day, we would like to show the body weight and survival data sheets of 83 overexpressing IGFBP5 (Response Fig. 18). We established to sacrifice the mouse when it has decreased around 20% of body weight loss from 0 day of injection. Moreover, if the mice show neurological symptoms such as hunched back and body trembling, we decided to sacrifice the mouse. As you can see in the table, we could observe all 4 mice had more than 20% decrease in body weight compared to 0 day except for two mice (number 1 mouse in cage 1, 097392 and number 5 in cage 2, 097393) (Response Fig. 18c). Two mice (number 1 mouse in cage 1, 097392 and number 5 in cage 2,

097393) showed severe neurological symptom. This is why we did sacrifice all 6 mice on the same day.

Response Fig. 18. Survival and bodyweight datasheet of orthotopic xenografts of 83 GSCs infected with vector control or IGFBP5 OE lentivirus. a, The survival days of a mouse in each orthotopic xenografts of 83 GSCs infected with vector control or IGFBP5 OE lentivirus (n = 6). **b,** Summary of survival data of orthotopic xenografts using 83 GSCs infected with vector control or IGFBP5 OE lentivirus analyzed by Prism software. The median survival days were 25 days for the control vector group, and 23 days for IGFBP5 OE group. $P = 0.0051$, log-rank test. **c,** The datasheet for body weight of orthotopic xenografts of 83 GSCs infected with vector control or IGFBP5 OE lentivirus.

Reviewer #2 (Remarks to the Author):

This MS by Lin et al. IGFBP5, a secreted protein known for its role of binding to IGFs, can act as a ROR1 ligand and promote glioblastoma stem cell (GSC) invasion. The authors provided good evidence indicating that IGFBP5 expression affects GSC migration/invasion. This finding is not surprising because it has been reported by several groups that IGFBP5 can stimulate the migration/invasion of normal and tumor cells, including glioblastoma cells. What is potentially new is the idea that IGFBP5 binds to ROR1, causes ROR1/HER2 heterodimer formation, and increases CREB-mediated gene expression. If proven correct, this would be a new and interesting finding. There are, however, a number of issues and major concerns. To convincingly support the notion that IGFBP5 causes ROR1/HER2 heterodimer formation and increases gene expression and cell invasion vis CREB, more rigorous, complimentary genetic (dominant negative & constitutive active) and pharmacological experiments are needed.

Major comments:

1) In this MS, critical technical details are often missing, making it difficult to assess the quality of the data. To give a few examples, I cannot find information on how different GSC cells (448, X01, 83, are these mouse #?) were isolated and what were the objective criteria to classify them into invasive vs. non-invasive GSCs? There is no description in the Methods section about the RNA-seq. The raw data of RNA-seq data set were not available.

Response: We apologize for the missing several critical technical details, especially for the statement “Statistics and reproducibility” in the Method part. We have included this part in our revised manuscript. We confirm that three technical replicates were performed for all experiments for reproducibility.

➤ Page 32 lines 12--20

“Statistics and reproducibility.

The Kaplan-Meier method was used to plot survival curves. In the case of patients who were alive at the time of last follow-up, survival records were censored in our analysis. The statistical analyses were performed with GraphPad PRISM software (v8.0, CA, USA). In the case of mouse experiments, results of multiple datasets were compared by analysis of variance (ANOVA) using the log-rank (Mantel-Cox) test. The results of two-dataset experiments were compared using a two-tailed Student’s t-test. *P* values < 0.05 were considered statistically significant; individual *P* values are provided in the figure legends. Three technical replicates were performed for all experiments for reproducibility.”

We hope that we could well answer the reviewer’s questions with two parts as follows:

Part I: The same point of these 4 GSCs (X01, 448, 131 and 83) were all primary cultured from human patient GBM tissue under a stem cell required condition without FBS. The differences and the details of GSCs (X01, 448, 131 and 83) could be explained as follows:

- 1) X01 GSC was isolated by Dr. Naoki Oka in 2007, they had identified X01 GSCs express CSC marker Nestin and exhibited extensive infiltrative capacity².
- 2) 448 GSC was dissociated by Dr. Do-Hyun Nam in 2013, which was classified as proneural subtype³.
- 3) GSC 131 was obtained from Dr. Do-Hyun Nam⁴, which was first published as mesenchymal type of glioblastoma by upregulation of the mesenchymal related genes.
- 4) 83 GSCs were isolated by Dr. Ichiro Nakano, they published 83 GSC as Mesenchymal subtype with enrichment of stemness marker CD44⁵.

We included related references in the revised Methods of our manuscript. Please see page 21, line 3 in the revised manuscript.

The classification of invasive or non-invasive GSCs was dependent on the phenotypic characterization according to the hematoxylin and eosin (H&E) staining of GSC-derived orthotopic xenograft models. 448 and X01 GSCs formed invasive tumors that spread into the brain through the corpus callosum, whereas 83 and 131 GSCs exhibited strong localization with a clear boundary, indicating non-invasive localization.

Part II: Once more, we apologize for the RNA-seq method missing, the detail of RNA-seq have been added in our revised manuscript. To provide you a simple explanation for the raw data set of RNA seq, we actually uploaded the raw data set of RNA sequence followed the editor's suggestion when this study was under consideration. However, we have no idea why you couldn't get our raw data before. Currently, NCBI has released our RNA-Seq raw result. Accordingly, you could get our RNA-Seq raw data from Raw omics data in the SRA database (<https://www.ncbi.nlm.nih.gov/sra>) under the study ID PRJNA732258. Also, we have included the details of the RNA-Seq analysis method in the revised manuscript Methods section.

➤ Page 30 line 20--Page 31 line 19

“High-throughput RNA sequencing (RNA-seq)

The library was prepared with 1 µg of total RNA for each sample by a TruSeq Stranded mRNA LT Sample Prep Kit (Illumina, Inc., San Diego, CA, USA). The first step in the workflow involves purifying the poly-A-containing mRNA molecules using poly-T-attached magnetic beads. Following purification, the mRNA was fragmented into small pieces using divalent cations under elevated temperatures. The cleaved RNA fragments were copied into first-strand cDNA using SuperScript II reverse transcriptase (Invitrogen) and random primers. This was followed by second-strand cDNA synthesis using DNA Polymerase I, RNase H and dUTP. These cDNA fragments were then subjected to an end-repair process, the addition of a single ‘A’ base, and adapter ligation. The products were then purified and enriched with PCR to create the final cDNA library. Indexed libraries were then paired-end sequenced with an Illumina HiSeq 4000 (Illumina, Inc., San Diego, CA, USA) at Macrogen Incorporated.

RNA-seq data analysis

Raw fastq files were aligned to the human reference genome (hg19) using the STAR program (<https://github.com/alexdobin/STAR>). The aligned sam files were converted to bam files and sorted by coordinate using the Samtools program (<http://www.htslib.org>). Duplicate reads were removed using Picard (<https://github.com/broadinstitute/picard>). The gene expression values were calculated based on the read per kilobase of exon per million (RPKM) value. Genes that had greater than 30% missing values were discarded. The expression levels of the filtered genes were globally normalized with the Quantile normalization method using the R limma package. The enrichment score (ES) of four GBM subtypes⁷³ with each molecular signatures in 772 GSCs was analyzed using single sample gene set enrichment analysis (ssGSEA).”

In vivo xenograft data in Fig. 3i. There is no description in the Methods section about the xenograft experiment. It is unclear what were the biological replica. The quantification data is not presented.

Response: We really appreciate the reviewer for pointing out the missed experimental details of xenograft data from Fig. 3i. We have revised the statement of Histology and immunohistochemistry (IHC) staining method in the revised manuscript as follows:

➤ Page 28 line 5--Page 29 line 2

“Histology, Immunohistochemistry (IHC) and immunofluorescence staining.

For histological observations, the brains were removed, fixed with 4% paraformaldehyde for 24 h at 4 °C, sectioned at a thickness of 4 μm using an essential microtome (Leica RM2125 RTS). For histological observations, haematoxylin and eosin (H&E) stains were carried out with 1% hematoxylin (DaKo) and 0.25% eosin (Merck). Stained sections were dried and mounted in an organic mounting medium.

Prior to immunohistochemical (IHC) staining for pHER2 Y1248 (1:100, R&D Systems), pROR1 Tyr786 (1:100, Thermo Fisher) and pCREB Ser133 (1:100, Cell Signaling Technology), the sections were subjected to an antigen retrieval process using citrate buffer (pH 6.0), and endogenous peroxidase was blocked by incubating with 3% hydrogen peroxide. The tissue sections were then incubated overnight at 4 °C in a humidified chamber with the primary antibody, diluted with antibody diluent buffer (IHC World). The tissue sections for 3,3'-diaminobenzidine (DAB) staining were developed using DAB (Vector Laboratories) as the chromogen.

For immunofluorescence staining against GFP, the tissue sections were subjected to an antigen retrieval process using citrate buffer (pH 6.0). Then incubated overnight at 4 °C with anti-GFP antibody (1:500, Abcam) in antibody diluent buffer (IHC World). Secondary staining was performed at RT for 2 h with fluorochrome-conjugated antibody (Alexa 568, Thermofisher Scientific) and 4',6-diamidino-2-phenylindole (DAPI; Sigma, 1:5,000). Fluorescence images were acquired with Zeiss LSM 780 confocal laser-scanning microscope (Carl Zeiss, Thornwood, NY, USA).”

Additionally, we calculated the number of GFP positive cells which infiltrated into the corpus callosum in the brains of mice. The number of infiltrated cells that had spread outside the tumor mass was extremely rare in Vector-overexpressed 83 GSC bearing mice. However, the expression was over 100 counts representing GFP-positive cells (Mean = 120) in ectopically IGFBP5 overexpressed 83 GSC bearing mice (Response Fig. 19). This quantitative data supports our conclusion that IGFBP5 promotes GSC invasion.

Response Fig. 19. Quantification of GFP-positive (GFP⁺) cell numbers spread into the corpus

callosum. GFP⁺ cells indicate infiltrating into the corpus callosum outside of tumor mass in orthotopic xenografts of 83 GSCs infected with vector control or IGFBP5 lentivirus. Data are presented as mean ± SEM (n = 3). **** $P < 0.0001$, *t*-test.

2) The conclusion that IGFBP5 acts as ROR1 ligand was based on the data shown in Fig. 4. Unfortunately, all biochemical assays lacked quantified results (e.g., Fig. 4a-4h). Were these experiments performed only one time? Quantitative and reproducible biochemical data are needed. Same question with Fig. 7k.

Response: We would like to explain that three technical replicates were actually performed for all experiments for reproducibility and have included “Statistics and reproducibility” section in the revised manuscript (details as mentioned above in Response to major Comment 1). In the response to major Comment 5, we have demonstrated the specificity of IGFBP5 binding with ROR1 by determining the Kd value (Please see details as below in Response to major Comment 5).

As suggested, the quantification of biochemical assays, immunoblot, and IHC staining has been provided in response Fig. 16 and 20. Kindly remind that we have replaced the siRNA of HER2 and ROR1 by shHER2 or shROR1 in X01 GSCs in response Fig. 21 for our revised Fig.4 k, l, which showed higher efficacy than the siRNA transfected cell, since reviewer 1 suggested us to conduct shROR1 and shHER2 for the further experiments.

Response Fig. 16 (Similar question from Reviewer 1). Quantification of RTK/Kinase Array. Quantitative analysis of the mean pixel density ratio of vehicle control (Veh, blue) to rIGF (100ng/ml recombinant IGFBP5 protein treated 6 h, red) for RTK array from manuscript Fig.4a (a), for Kinase array from Fig.4b (b) in 83 GSCs by using ImageJ software. * $P < 0.05$, ** $P < 0.01$, *t*-test. Refer Spots were used as loading controls.

Response Fig. 20. Quantification of IB analysis in Fig. 4. Bar graph for IB analysis to calculate the gray value of each band, shown as the ratio of phosphorylation to total CREB/HER2/ROR1 expression level from original Fig.4c (a), original Fig.4d (b), original Fig.4e (c), revised Fig.4k, l (d), original Fig.7k (e) using ImageJ software. * $P < 0.05$, ** $P < 0.01$, *** $P < 0.001$, n.s. $P > 0.05$, t -test.

Response Fig. 21 (Revised Fig. 4k, l).

3) It is also troublesome that contradictory results (e.g., Fig.4d, different effects on ROR1 levels; Fig. 4e different effects ROR1 levels) were ignored. How was the staining specificity validated?

Response: Considering the different expressions of basic ROR1 in non-invasive GSCs (Fig. 4d) and invasive GSCs (Fig. 4e), the ratio of phosphorous to total protein (ROR1, HER2, CREB) was estimated from immunoblot results, which showed statistical significance (Response Fig. 20b, c).

According to anti-ROR1 antibody datasheet (#16540, Cell Signaling), ROR1 has been detected at 130 and 135 kDa, validating the accurate molecular weight of ROR1 and its decreased expression under

ROR1 knockdown by shROR1. (Response Fig. 20 and 21).

4) Fig. 4k, Fig. 6e, 6g: Only one section image is shown per group. What were the biological replicates? Are these statistically different changes? How was the staining specificity validated? Are these GSC cells and how do you know?

Response: We would like to explain that three technical replicates were actually performed for all experiments for reproducibility and have included “Statistics and reproducibility” section in the revised manuscript (details as mentioned above in Response to major Comment 1).

Additionally, the representative images of IHC staining for original Fig. 4k, Fig. 4l, and Fig. 6g (Response Fig. 22-24) were provided and quantified. And for the specificity of the primary antibody, we employed IgG and secondary antibodies in each brain slides of each experimental groups to confirm the negative controls and auto-fluorescence.

As the reviewer suggested, we have detected and quantified the expression of stemness marker Nestin in the brain of X01 GSCs-bearing mice (Response Fig. 25) to confirm the stemness of GSCs after orthotopically injected GSCs into the mice. The result of Nestin staining showed knocking down IGFBP5 reduced Nestin expression in the tumor region of X01 GSCs orthotopic xenograft mice models.

Response Fig. 22. Immunohistochemistry (IHC) staining of pCREB, pHER2, pROR1 in orthotopic xenografts of X01 GSCs infected with shCtrl, shIGFBP5-1, or shIGFBP5-2 lentivirus. **a**, IHC staining of pCREB in orthotopic xenograft of X01 GSCs infected with shCtrl, shIGFBP5-1, or shIGFBP5-2 lentivirus. The 3 independent experiments of the staining labeling as pCREB-1, pCREB-2 and pCREB-3 from the top to the bottom (scale bar, 100 μ m). **b**, the graph shows the mean percentage of pCREB-positive (pCREB+) cells in the total cell numbers \pm SEM (n = 3). ** $P < 0.01$, *** $P < 0.001$, t -test. **c**, IHC staining of pHER2 in orthotopic xenograft of X01 GSCs infected with shCtrl, shIGFBP5-1, or shIGFBP5-2 lentivirus. The 3 independent experiments of the staining labeling as pHER2-1, pHER2-2 and pHER2-3 from the top to the bottom (scale bar, 100 μ m). **d**, the graph shows the mean percentage of pHER2-positive (pHER2+) cells in the total cell numbers \pm SEM (n = 3). * $P < 0.05$, ** $P < 0.01$, t -test. **e**, IHC staining of pROR1 in orthotopic xenograft of X01 GSCs infected with shCtrl, shIGFBP5-1, or shIGFBP5-2 lentivirus. The 3 independent experiments of the staining labeling as pROR1-1, pROR1-2 and pROR1-3 from the top to the bottom (scale bar, 100 μ m). **f**, the graph shows the mean percentage of pROR1-positive (pROR1+) cells in the total cell numbers \pm SEM (n = 3). ** $P < 0.01$, *** $P < 0.001$, t -test.

1, shIGFBP5-2 lentivirus. The 3 independent experiments of the staining labeling as pHER2-1, pHER2-2 and pHER2-3 from the top to the bottom (scale bar, 100 μ m). **d**, the graph shows the mean percentage of pHER2-positive (pHER2+) cells in the total cell numbers \pm SEM (n = 3). * $P < 0.05$, *t*-test. **e**, IHC staining of pROR1 in orthotopic xenografts of X01 GSCs infected with shCtrl, shIGFBP5-1, shIGFBP5-2 lentivirus. The 3 independent experiments of the staining labeling as pROR1-1, pROR1-2 and pROR1-3 from the top to the bottom (scale bar, 100 μ m). **f**, the graph shows the mean percentage of pROR1-positive (pROR1+) cells in the total cell numbers \pm SEM (n = 3). ** $P < 0.01$, *t*-test.

Response Fig. 23. IHC staining of pCREB, pHER2, pROR1 in orthotopic of 83 GSCs infected with vector control or IGFBP5 lentivirus. **a**, IHC staining of pCREB in orthotopic xenografts of 83 GSCs infected with vector control and IGFBP5 lentivirus. The 3 independent experiments of the staining labeling as pCREB-1, pCREB-2 and pCREB-3 from the top to the bottom (scale bar, 100 μ m). **b**, the graph shows the mean percentage of pCERB-positive (pCREB+) cells in the total cell numbers \pm SEM (n = 3). * $P < 0.05$, *t*-test. **c**, IHC staining of pHER2 in orthotopic xenografts of 83 GSCs infected with vector control and IGFBP5 lentivirus. The 3 independent experiments of the staining labeling as pHER2-1, pHER2-2 and pHER2-3 from the top to the bottom (scale bar, 100 μ m). **d**, the graph shows the mean percentage of pHER2-positive (pHER2+) cells in the total cell numbers \pm SEM (n = 3). * $P < 0.05$, *t*-test. **e**, IHC staining of pROR1 in orthotopic xenografts of 83 GSCs infected with vector control and IGFBP5 lentivirus. The 3 independent experiments of the staining labeling as pROR1-1, pROR1-2 and pROR1-3 from the top to the bottom (scale bar, 100 μ m). **f**, the graph shows the mean percentage of pROR1-positive (pROR1+) cells in the total cell numbers \pm SEM (n = 3). *** $P < 0.001$, *t*-test.

Response Fig. 24. IHC staining of pCREB, pHER2, pROR1 in orthotopic xenografts of 772 GSCs infected with shCtrl, shIGFBP5-1, or shIGFBP5-2 lentivirus. **a**, IHC staining of pCREB in orthotopic xenografts of 772 GSCs infected with shCtrl, shIGFBP5-1, shIGFBP5-2 lentivirus. The 3 independent experiments of the staining labeling as pCREB-1, pCREB-2 and pCREB-3 from the top to the bottom (scale bar, 100 μ m). **b**, the graph shows the mean percentage of pCREB-positive (pCREB+) cells in the total cell numbers \pm SEM (n = 3). ** $P < 0.01$, *** $P < 0.001$, *t*-test. **c**, IHC staining of pHER2 in orthotopic xenografts of 772 GSCs infected with shCtrl, shIGFBP5-1, shIGFBP5-2 lentivirus. The 3 independent experiments of the staining labeling as pHER2-1, pHER2-2 and pHER2-3 from the top to the bottom (scale bar, 100 μ m). **d**, the graph shows the mean percentage of pHER2-positive (pHER2+) cells in the total cell numbers \pm SEM (n = 3). *** $P < 0.001$, **** $P < 0.0001$, *t*-test. **e**, IHC staining of pROR1 in orthotopic xenograft of 772 GSCs infected with shCtrl, shIGFBP5-1, shIGFBP5-2 lentivirus. The 3 independent experiments of the staining labeling as pROR1-1, pROR1-2 and pROR1-3 from the top to the bottom (scale bar, 100 μ m). **f**, the graph shows the mean percentage of pROR1-positive (pROR1+) cells in the total cell numbers \pm SEM (n = 3). ** $P < 0.01$, *t*-test.

Response Fig. 25. Immunofluorescence staining of stemness marker Nestin in orthotopic xenografts of X01 GSCs infected with shCtrl, shIGFBP5-1, or shIGFBP5-2 lentivirus. **a**, Immunofluorescence staining of stemness marker Nestin in orthotopic xenograft of X01 GSCs infected with shCtrl, shIGFBP5-1, shIGFBP5-2 lentivirus. The 3 independent experiments of the staining tumor

tissue were labeling as Nestin-1, Nestin-2 and Nestin-3 from the top to the bottom (scale bar, 100 μm). Nestin was labeled in red, and Nuclei were counterstained with DAPI (blue). **b**, the graph shows the mean percentage of Nestin-positive (Nestin⁺) cells in the total cell numbers \pm SEM (n = 3). ** $P < 0.01$, *t*-test.

5) To convincingly support the notion that IGFBP5 causes ROR1/HER2 heterodimer formation and increases gene expression and cell invasion vis CREB, more rigorous, independent genetic (dominant negative & constitutive active) and pharmacological experiments are needed. It will also be critical to demonstrate the specificity of IGFBP5 binding and the K_d value.

Response: We appreciate the reviewer's comment. As suggested, we have investigated the role of each molecule in invasion of GSCs to strengthen ROR1/HER2/CREB mechanism by i) knocking down HER2 using shRNA or pharmacological inhibitor, Irbinitinib (Response Fig. 1); ii) knocking down ROR1 using shRNA (Response Fig. 2a-d); iii) modulating ROR1 activation by overexpressing ROR1 wild type (ROR1 WT) and kinase domain deletion type (ROR1 mut) (Response Fig. 2e-h) in GSCs, including rescue of ROR1 WT or mut expression in IGFBP5 knocking down GSCs; iv) overexpressed CREB in non-invasive GSCs and under loss function of ROR1 or HER2. Results were shown as follows:

- i) To determine the functional role of HER2 and ROR1 in GSC invasion, we silenced HER2 and ROR1 in invasive GSCs 448 and X01 with shRNA. As shown in response Fig. 1 and 2, both CREB phosphorylation and the invasive capacity of GSCs 448 and X01 were downregulated by knocking down HER2 (Response Fig. 1a-d) and ROR1 (Response Fig. 2a-d).
- ii) The pharmacological HER2 inhibitor, Irbinitinib reduced the downstream of CREB phosphorylation (Response Fig. 1e, f) and the GSCs invasion (Response Fig. 1g, h) as well.
- iii) We also modulated ROR1 activation by overexpressing ROR1 wild type (ROR1 WT) and kinase deletion type (ROR1 mut) in the non-invasive GSCs 83 and 131. Our results showed that overexpression of ROR1 (ROR1 WT) increased phosphorylation of CREB (Response Fig. 2e, f) and CREB downstream genes including ETV5 and FBXW9 (Response Fig. 2g-j), but not in ROR1 mut. More importantly, overexpression of ROR1 (ROR1 WT) but not ROR1 mut rescued knockdown of IGFBP5 mediated repression of HER2 and CREB phosphorylation (Response Fig. 2k) and GSCs invasion (Response Fig. 2l).
- iv) We overexpressed CREB in non-invasive GSC and determined the role of CREB in regulating GSC invasion. Immunoblot analysis verified a successful ectopic CREB overexpression in non-invasive GSCs (83 and 131) (Response Fig. 3a-b). CREB overexpression significantly increased downstream genes ETV5 and FBXW9 expression, and GSC invasive capacity (Response Fig. 3c-h). To investigate the downstream of CREB in IGFBP5-ROR1/HER2

signaling pathway, we ectopic overexpressed CREB in IGFBP5 knockdown GSCs, and CREB sufficiently rescued GSC invasion (Response Fig. 3i, j). Furthermore, overexpression of CERB also rescued HER2 or ROR1 knockdown mediated repression of GSC invasion (Response Fig. 4a-h). These results indicated CREB is the key molecule in IGFBP5-mediated ROR1/HER2 signaling axis.

Taken together, these results suggest an essential role of IGFBP5/ROR1/HER2-CREB signaling axis in GSC invasion.

Response Fig. 1 (Revised Fig. 4k and New Extended Data Fig. 4d, f-i). Inhibition of HER2 expression suppresses GSCs invasion and the downstream CREB signaling. **a-b**, Immunoblot (IB) analysis of pROR1, pHER2, pCREB, ROR1, HER2, and CREB in 448 (**a**) and in X01 (**b**) GSCs infected with shCtrl, shHER2-1, or shHER2-2 lentivirus. GAPDH was used as a loading controls. **c**, Invasion assays of 448 GSCs infected with shCtrl, shHER2-1, or shHER2-2 lentivirus. Images (left) taken after 24 h of invasion are representative of three independent experiments (scale bar, 100 µm), and the graph (right) shows the mean number of invasive cells ± SEM (n = 3). ** $P < 0.01$, t -test. **d**, Invasion assays of X01 GSCs infected with shCtrl, shHER2-1, or shHER2-2 lentivirus. Images (left) taken after 24 h of invasion are representative of three independent experiments (scale bar, 100 µm), and the graph (right) shows the mean number of invasive cells ± SEM (n = 3). *** $P < 0.001$, t -test. **e-f**, IB analysis of pROR1, pHER2, pCREB, ROR1, HER2, and CREB in 448 (**e**) and in X01 (**f**) GSCs after 24 h Irbitinib treatment. GAPDH was used as a loading controls. **g**, Invasion assays of 448 GSCs treated with Voronoi with 1 µM, 2 µM or vehicle control for 24 h. Images (left) taken after 24 h of invasion are representative of three independent experiments (scale bar, 100 µm), and the graph (right) shows the mean number of invasive cells ± SEM (n = 3). *** $P < 0.001$, t -test. **h**, Invasion assays of X01 GSCs treated with Irbitinib with 1 µM, 2 µM or vehicle control for 24 h. Images (left) taken after 24 h of invasion are representative of three independent experiments (scale bar, 100 µm), and the graph (right) shows the mean number of invasive cells ± SEM (n = 3). *** $P < 0.001$, t -test.

Response Fig. 2 (Revised Fig. 4l and New Extended Data Fig. 4e, f-l, similar question from Reviewer 1). ROR1 regulates GSCs invasion and the downstream CREB signaling. a-b, IB analysis of pROR1, pHER2, pCREB, ROR1, HER2, and CREB in 448 (**a**) and in X01 (**b**) GSCs infected with shCtrl, shROR1-1, or shROR1-2 lentivirus. GAPDH was used as a loading control. **c-d**, Invasion assays of 448 (**c**) and X01 (**d**) GSCs infected with shCtrl, shROR1-1, or shROR1-2 lentivirus. Images (left) taken after 24 h of invasion are representative of three independent experiments (scale bar, 100 μ m), and the graph (right) shows the mean number of invasive cells \pm SEM (n = 3). ** $P < 0.01$, *** $P < 0.001$, t -test. **e-f**, IB analysis of pROR1, pHER2, pCREB, ROR1, HER2, and CREB in 83 (**e**) and in 131 (**f**) GSCs infected with vector control, ROR1 WT, or ROR1 mut lentivirus. GAPDH was used as a loading control. **g-h**, RT-qPCR analysis of CREB targets ETV5 (**g**) and FBXW9 (**h**) expression in 83 GSCs infected with vector control, ROR1 WT, or ROR1 mut lentivirus. Data are presented as mean \pm SEM (n = 3). * $P < 0.05$, t -test. **i-j**, RT-qPCR analysis of CREB targets ETV5 (**i**) and FBXW9 (**j**) expression in 131 GSCs infected with vector control, ROR1 WT, or ROR1 mut lentivirus. Data are presented as mean \pm SEM (n = 3). * $P < 0.05$, ** $P < 0.01$, t -test. **k**, IB analysis of pROR1, pHER2, pCREB, ROR1, HER2, and CREB in X01 GSCs infected with shCtrl, shIGFBP5-1 lentivirus, and then infected with vector control, ROR1 WT, or ROR1 mut lentivirus. GAPDH was used as a loading control. **l**, Invasion assays of X01 GSCs infected with shCtrl, shIGFBP5-1 lentivirus, and then infected with vector control, ROR1 WT, or ROR1 mut lentivirus. Images (left) taken after 24 h of invasion are representative of three independent experiments (scale bar, 100 μ m), and the graph (right) shows the mean number of invasive cells \pm SEM (n = 3). ** $P < 0.01$, t -test.

Response Fig. 3 (Revised Extended Data Fig. 6d-n; similar question from Reviewer 1). Overexpression of CREB enhances GSCs invasion. a-b, IB analysis of pCREB, and CREB in 83 (a) and in 131 (b) GSCs infected with vector control, or CREB OE lentivirus. GAPDH was used as a loading control. **c-d**, RT-qPCR analysis of CREB targets ETV5 (c) and FBXW9 (d) expression in 83 GSCs infected with vector control, CREB OE lentivirus. Data are presented as mean \pm SEM (n = 3). ** $P < 0.01$, *t*-test. **e-f**, RT-qPCR analysis of CREB targets ETV5 (e) and FBXW9 (f) expression in 131 GSCs infected with vector control, CREB OE lentivirus. Data are presented as mean \pm SEM (n = 3). * $P < 0.05$, ** $P < 0.01$, *t*-test. **g-h**, Invasion assays of 83 (g) and 131 (h) GSCs infected with vector control, or CREB OE lentivirus. Images (left) taken after 48 h of invasion are representative of three independent experiments (scale bar, 100 μ m), and the graph (right) shows the mean number of invasive cells \pm SEM (n = 3). * $P < 0.05$, *t*-test. **i**, IB analysis of pROR1, pHER2, pCREB, ROR1, HER2, and CREB in X01 GSCs infected with shCtrl, shIGFBP5-1 lentivirus, and then infected with vector control, CREB OE lentivirus. GAPDH was used as a loading control. **j**, Invasion assays of X01 GSCs infected with shCtrl, shIGFBP5-1 lentivirus, and then infected with vector control, CREB OE lentivirus. Images (left) taken after 24 h of invasion are representative of three independent experiments (scale bar, 100 μ m), and the graph (right) shows the mean number of invasive cells \pm SEM (n = 3). ** $P < 0.01$, *t*-test.

Response Fig. 4 (New Extended Data Fig. 7; similar question from Reviewer 1). Overexpression of CERB rescues HER2 or ROR1 knockdown-mediated repression of GSC invasion. a-b, IB analysis of pROR1, pHER2, pCREB, ROR1, HER2, and CREB in 448 (a) and X01 (b) GSCs infected with shCtrl, shHER2-1 lentivirus, and then infected with vector control, CREB OE lentivirus. GAPDH was used as a loading control. **c-d,** Invasion assays of 448 (c) and X01 (d) GSCs infected with shCtrl, shHER2-1 lentivirus, and then infected with vector control, CREB OE lentivirus. Images (left) taken after 24 h of invasion are representative of three independent experiments (scale bar, 100 μ m), and the graph (right) shows the mean number of the invasive cells \pm SEM (n = 3). * $P < 0.05$, ** $P < 0.01$, t -test. **e-f,** IB analysis of pROR1, pHER2, pCREB, ROR1, HER2, and CREB in 448 (e) and X01 (f) GSCs infected with shCtrl, shROR1-1 lentivirus, and then infected with vector control, CREB OE lentivirus. GAPDH was used as a loading control. **g-h,** Invasion assays of 448 GSCs (g) and X01 (h) GSCs infected with shCtrl, shROR1-1 lentivirus, and then infected with vector control, CREB OE lentivirus. Images (left) taken after 24 h of invasion are representative of three independent experiments (scale bar, 100 μ m), and the graph (right) shows the mean number of invasive cells \pm SEM (n = 3). * $P < 0.05$, ** $P < 0.01$, t -test.

For the suggestion on specificity of IGFBP5 binding and the K_d value, we have conducted microscale thermophoresis assay (MST) to identify the specificity binding between IGFBP5 and HER2/ROR1 dose responses, and also calculated the K_d value through the K_d fitting mode using MO Affinity Analysis software. According to the analysis of IGFBP5-ROR1 binding affinity curve, we concluded that IGFBP5 and ROR1 could specifically bind with K_d value of 157.5 nM (Response Fig. 26). However, we found that IGFBP5 and HER2 could not bind with no affinity, since the signal to noise ratio less than 5 (Response Fig. 27). Taken together, our results suggest that IGFBP5 serves as a novel ligand for ROR1.

Response Fig. 26 (Revised Fig. 4j). Measurement of IGFBP5-ROR1 binding affinity *in vitro*. The *in vitro* binding affinity between IGFBP5 and ROR1 was determined by MST assay. The concentration of IGFBP5 proteins is kept constant at 50 nM, while the ROR1 concentration varies from 1.45 μ M to 0.04 nM. The binding curve yields a K_d of 157.5 nM. Inset, thermophoretic movement of fluorescently labeled proteins. $F_{norm} = F_1/F_0$ (F_{norm} : normalized fluorescence; F_1 : fluorescence after thermosdiffusion; F_0 : initial fluorescence or fluorescence after T-jump). K_d , dissociation constant.

Response Fig. 27 (New Extended Data Fig. 4c). Measurement of IGFBP5-HER2 binding affinity *in vitro*. The *in vitro* binding affinity between IGFBP5 and HER2 was tested by MST assay. The concentration of IGFBP5 proteins is kept constant at 50 nM, while the HER2 concentration varies from 1.00 μ M to 0.03 nM at response time of 10 sec (a) and 2.5 sec (b).

6) Fig. 7m – I could not see much difference between the scrambled control and sgIGFBP5 group. Yet, it was stated that *in vivo* nano capsule-mediated IGFBP5 disruption dramatically increase mouse survival. Did I miss something? Did the authors actually detect any indel in the IGFBP5 gene in these mice?

Response: We have provided the detailed information regarding this as below:

In the Fig. 7m, the median survival time of mice in the PBS treatment group is 23 days, sgScramble group is 25 days, while sgIGFBP5 group is prolonged up to 64 days. Accordingly, survival time of mice treated by sgIGFBP5 increased around 200% comparing to these of mice treated with PBS or sgScramble.

For your other concern about indel in the IGFBP5 gene editing in the mice, our previous work using the same nanocapsules except for the target gene PLK1 instead of IGFBP5, showed these nanocapsules induced 38.1% indel frequencies in GBM-bearing mice after successive intravenous injections, which is the highest gene editing efficiency in the brain via non-invasive treatment to the best of our knowledge⁸. To further evaluate whether the efficient gene editing of these nanocapsules resulted from the blood-brain barrier (BBB) penetration and tumor accumulation, we also conducted tumor penetration experiments of Ang-SS-Cas9/sgRNA and free Cas9/sgRNA. Confocal images showed that our Ang-SS-Cas9/sgRNA nanocapsules have much stronger brain targeting ability than free Cas9/sgRNA (Response Fig. 28). Moreover, immunoblot analysis of IGFBP5 in tumor tissue from mice treated with Ang-SS-Cas9/sgIGFBP5 showed much lower expression level than Ang-SS-Cas9/sgScramble or PBS (Original Fig. 7k). Taken together, these data suggest that the Ang-SS-Cas9/sgRNA nanocapsules can efficiently deliver Cas9/sgIGFBP5 to the brain tumor and suppress IGFBP5 gene expression in the tumor.

Response Fig. 28. CLSM images on tumor penetration of Ang-SS-Cas9/sgRNA and free Cas9/sgRNA. Nuclei were stained with DAPI (blue) and AF647-Cas9 was red. Dotted lines indicate tumor boundary (Scale bars, 20 μ m).

Minor comments:

1) Fig. 7. Did the Cas9/sgIGFBP5 treatment actually reduced IGFBP5 levels in X01 and 448 cells? Is yes, how much?

Response: Yes, Cas9/sgIGFBP5 not only reduced the IGFBP5 protein expression in X01 and 448 GSCs

(Response Fig. 17), but also depressed IGFBP5 expression level in the xenograft tumor tissue (Original Fig. 7k). Successful knockdown of IGFBP5 was validated by ELISA analysis after Cas9/sgIGFBP5 treatment (Response Fig. 17a, b). Immunoblot (IB) results showed that knockdown of IGFBP5 decreased pROR1, pHER2 and pCREB expression level (Response Fig. 17c, d).

Response Fig. 17 (Similar question from Reviewer 1). Cas9/sgIGFBP5 treatment reduces IGFBP5 protein expression levels. a-b, ELISA analysis of IGFBP5 in conditioned media (CM) from 448 (a) and X01 (b) GSCs after Cas9/sgIGFBP5 treatment. Data are presented as mean \pm SEM (n = 3). * $P < 0.05$, ** $P < 0.01$, t -test. **c-d,** IB analysis of IGFBP5, pROR1, pHER2, pCREB, ROR1, HER2, and CREB in 448 (c) and X01 (d) GSCs after Cas9/sgIGFBP5 treatment. GAPDH was used as a loading control.

2) Fig. 7k, what was analyzed here, GSCs or the entire tumor?

Response: Fig.7k showed the IB analysis of entire tumor tissue lysates.

3) The effects of IGFBP5 on cell migration/invasion have been reported since the late 1990s. These papers should be cited.

Response: We appreciate reviewer's helpful comments. We have cited the papers as you suggested.

➤ Page 6 lines 11-- 13

“IGFBP5 is a secreted protein of the IGFBP family that mainly regulates the specific binding of insulin-like growth factors (IGFs) to IGF receptors²¹⁻²⁴, which showed a high correlation with the migration of breast cancer^{25,26}.”

4) For in vitro invasion assays using shRNA-mediated knockdown, the possible off-target effects were not addressed. A simple experiment would be to test if adding back rIGFBP5 to the knocked down cells can rescue the cell invasion/gene expression.

Response: We appreciate reviewer's helpful comments. As suggested, we treated recombinant IGFBP5

back into the IGFBP5 knockdown 448 and X01 GSCs. The result of invasion assay and immunoblot showed rIGFBP5 treatment can rescue the phosphorylation of HER2, ROR1 and CREB (Response Fig. 29a, b) and GSCs invasive ability (Response Fig. 29c, d).

Response Fig. 29. Effects of IGFBP5 knockdown on GSC invasion are rescued by recombinant IGFBP5 protein treatment. a-b, IB analysis of pROR1, pHER2, pCREB, ROR1, HER2, and CREB in 448 (a) and in X01 (b) GSCs treated with recombinant IGFBP5 (rIGF) 100 ng/ml or vehicle control for 24 h after infected with shIGFBP5-1 lentivirus. β -actin was used as a loading control. c-d, Invasion assays of 448 GSCs (c) and X01 GSCs (d) treated with recombinant IGFBP5 (rIGF) 100 ng/ml or vehicle control for 24 h after infected with shCtrl or shIGFBP5-1 lentivirus. rIGFBP5 100 ng/ml treated from the 24 h to 48 h after shCtrl or shIGFBP5-1 lentivirus transduction into 448 or X01 GSCs. Images (left) taken after 24 h of invasion are representative of three independent experiments (scale bar, 100 μ m), and the graph (right) shows the mean number of the invasive cells \pm SEM (n = 3). * $P < 0.05$, *** $P < 0.001$, *t*-test.

Reviewer #3 (Remarks to the Author):

This interesting study shows a clear effect of IGFBP-5 in promoting GBM stem cell invasion and provides a plausible mechanism involving ROR1, HER2, and CREB activation, together with the CREB-dependent expression of ETV5 and FBXW9. Strengths: In general, the experiments are clearly presented and well interpreted, and the development of an IGFBP-5-targeting nanocapsule that significantly impairs GBM tumor growth and prolongs survival in mice with orthotopic GBM tumors is very impressive. Weaknesses: Although IGFBP-5 is immunoprecipitated using an anti-ROR1 antibody its direct binding to ROR1, as proposed, is not demonstrated, and the existence of a functional HER2-ROR1-IGFBP-5 complex is unclear since it is not precipitated using an anti-HER2 antibody (see specific comment #4). Therefore the precise mechanism of IGFBP-5 action is in some doubt, and a role for IGF1R phosphorylation is not explicitly excluded by the single negative phosphoarray experiment (see specific comment #7).

Nevertheless this work adds significantly to current knowledge of the role of IGFBP-5 in GBM.

Specific comments

1. Line 116: “a small number of studies”. Since high IGFBP-5 expression in GBM has been reported previously (as far back as 2006, doi: 10.1177/153303460600500303), some of these prior studies should be cited here.

Response: We thank the reviewer for raising this point. We have cited the relative references in our revised manuscript.

➤ Page 17 lines 16--17

“Three IGFBP isoforms, IGFBP2, IGFBP3, and IGFBP5, are reportedly overexpressed in clinical biopsies of high-grade diffuse glioma⁴¹⁻⁴³.”

2. Fig. 4c. The pROR1 and pCREB responses seem weaker in 131 cells than 83 cells. These data would benefit from quantitation of phospho/total ratios from several independent assays, and statistical evaluation.

Response: As suggested, we quantified the immunoblot results of Fig.4c, and calculated the ratio of phosphorous to total protein (CREB, HER2, ROR1) estimated from immunoblot results (Response Fig. 30). As the bars showed, the treatment of recombinant IGFBP5 protein (rIGF) to GSCs significantly increased phospho/total ratio compared to control groups.

Response Fig. 30. Quantification of IB analysis in Fig. 4c. Bar graph for IB analysis to calculate the gray value of each band, shown as the ratio of phosphorylation to total CREB/HER2/ROR1 expression level from manuscript Fig. 4c (left 83, right 131) using ImageJ software. * $P < 0.05$, ** $P < 0.01$, *** $P < 0.001$, t -test.

3. Fig. 4f, g: Some IP bands are stronger than the input. State what fraction of the total lysate was analyzed in the input lanes.

Response: We apologize for missing the statement of IP in Methods part. We have provided more detailed information about IP in our revised manuscript. For your information here, the protein amount of input lane was around 4% of the total cell lysate.

➤ Page 26 lines 3--16

“Co-immunoprecipitation.

For co-immunoprecipitation (Co-IP) experiments with endogenous proteins, X01 or 448 GSCs were washed twice with ice-cold PBS and fixed with 1 mM dithiobis (succinimidyl propionate, DSP;

TOPSCIENCE) at room temperature for 30 min, followed by lysis with IP buffer (Thermo Fisher Scientific) in the presence of 1 % protease inhibitor. Then, 4 % protein lysate was used as input and the remaining cell lysates were mixed with protein A/G Dynabeads (Thermo Fisher Scientific) preincubated with normal rabbit IgG (Merck) or the indicated antibody at 4 °C overnight. For co-IP of exogenous proteins, 293T cells were transfected with the indicated plasmids 48 h before harvest, fixation, and lysis. Again, 4 % protein lysate was used as input and the remaining cell lysates were mixed with Flag/HA-Magnetic (Bimake) protein A/G Dynabeads (Thermo Fisher Scientific) at 4 °C overnight. The beads were then washed 5-7 times with IP buffer (Thermo Fisher Scientific) supplemented with 1% protease inhibitor. Immunoprecipitates were boiled for 10 min at 98 °C in protein loading buffer (EpiZyme) for further immunoblotting analysis.”

4. Lines 185-186. If a ROR1-IGFBP-5 complex interacts with HER2, IP with a HER2 antibody should precipitate both ROR1 and IGFBP-5, but this is not seen in Fig. 4f. One interpretation is that IGFBP-5 can bind to ROR1, but this binary complex does not interact with HER2. This is different from the model shown in Fig. 4m. Further interpretation of the coIP data is required, as well as some coIPs using anti-IGFBP-5 as the precipitating Ab.

Response: We greatly appreciate the reviewer for pointing this out. As suggested, the model in Fig. 4m was not precise model and deleted in the revised Fig. 4. We have conducted additional experiments to support our theory, including revised Fig. 4 and a New Extended Fig. 4.

(1) immunoprecipitation with HER2 or ROR1 antibody in X01 and 448 GSCs, it was found that the IP of HER2 antibody could precipitate ROR1, and the IP of ROR1 antibody could precipitate HER2 (Response Fig. 31a-d), indicating that there was interaction between HER2 and ROR1.

(2) exogenous expression of HA-ROR1 and Flag-IGFBP5 in 293T cells, HA and Flag were used as precipitation antibodies for Co-IP. Co-IP using Flag as the precipitating antibody showed interactions between IGFBP5 and ROR1, and ROR1 interacts with both IGFBP5 and endogenous HER2 when using HA as the precipitating antibody (Response Fig. 31e, f).

(3) the purified His-IGFBP5 and ROR1 recombinant proteins were used for in vitro MST assay to detect the affinity between ROR1 and IGFBP5. The results showed a clear binding curve to IGFBP5-ROR1 with a Kd of 157.5 nM (Response Fig. 31g), but no binding affinity between IGFBP5 and HER2 (Response Fig. 27a, b), which suggested that IGFBP5 serves as a novel ligand for ROR1 to trigger downstream signaling transduction.

We described this part in the revised manuscript.

Please see page 9, line 11--page 10, line 3 in our revised manuscript.

Response Fig. 31 (Revised Fig. 4f-j and New Extended Data Fig. 4a, b). IGFBP5 is a novel ROR1 ligand in GSCs. a-b, Co-IP of X01 GSCs with antibodies targeting HER2 (a), ROR1 (b) or normal IgG. **c-d,** Co-IP of 448 GSCs with antibodies targeting HER2 (c), ROR1 (d) or normal IgG. **e,** Co-IP analysis for the interaction of IGFBP5 and ROR1 in 293T cells transfected with Flag-tagged IGFBP5 and HA-tagged ROR1. Cell lysates were precipitated with anti-Flag antibody. **f,** Co-IP analysis for the interaction of IGFBP5 and ROR1 in 293T cells transfected with HA-tagged ROR1 and Flag-tagged IGFBP5. Cell lysates were precipitated with anti-HA antibody. **g,** *In vitro* binding affinity between IGFBP5 and ROR1 was tested by MST assay. The concentration of IGFBP5 proteins is kept constant at 50 nM, while the ROR1 concentration varies from 1.45 μ M to 0.04 nM. The binding curve yields a K_d of 157.5 nM. Inset, thermophoretic movement of fluorescently labeled proteins. $F_{norm} = F_1/F_0$ (F_{norm} : normalized fluorescence; F_1 : fluorescence after thermodiffusion; F_0 : initial fluorescence or fluorescence after T-jump). K_d , dissociation constant.

Response Fig. 27 (New Extended Data Fig. 4c; similar question from Reviewer 2). Measurement of IGFBP5-HER2 binding affinity *in vitro*. The *in vitro* binding affinity between IGFBP5 and HER2 was tested by MST assay. The concentration of IGFBP5 proteins is kept constant at 50 nM, while the HER2 concentration varies from 1.00 μ M to 0.03 nM at response time of 10 sec (a) and 2.5 sec (b).

5. Fig. 4h suggests that both ROR1 and HER2 have a role in CREB phosphorylation, but it provides no information about the role of IGFBP-5 in this process. Neither do Figs. 4i and j provide information about the role of IGFBP-5. So line 190, “.. implying that IGFBP-5 promotes ...”, overinterprets the data.

Response: We thank the reviewer for this comment. We determined overexpression of ROR1 wild type

(ROR1 WT) but not ROR1 mut (Response Fig. 2), and overexpression of CREB (Response Fig. 3) rescued knockdown IGFBP5 mediated repression of HER2-CREB phosphorylation and GSC invasion. As suggested, we deleted overinterpreted sentence “.. implying that IGFBP-5 promotes ...”, and rewrote this paragraph in our revised manuscript.

Please see page 10, lines 4-16 in our revised manuscript.

Response Fig. 2 (Revised Fig. 4l and New Extended Data Fig. 4e, f-l, similar question from Reviewers 1 and 2). **a-b**, IB analysis of pROR1, pHER2, pCREB, ROR1, HER2, and CREB in 448 (**a**) and in X01 (**b**) GSCs infected with shCtrl, shROR1-1, or shROR1-2 lentivirus. GAPDH was used as a loading control. **c-d**, Invasion assays of 448 (**c**) and X01 (**d**) GSCs infected with shCtrl, shROR1-1, or shROR1-2 lentivirus. Images (left) taken after 24 h of invasion are representative of three independent experiments (scale bar, 100 μ m), and the graph (right) shows the mean number of invasive cells \pm SEM (n = 3). ** $P < 0.01$, *** $P < 0.001$, *t*-test. **e-f**, IB analysis of pROR1, pHER2, pCREB, ROR1, HER2, and CREB in 83 (**e**) and in 131 (**f**) GSCs infected with vector control, ROR1 WT, or ROR1 mut lentivirus. GAPDH was used as a loading control. **g-h**, RT-qPCR analysis of CREB targets ETV5 (**g**) and FBXW9 (**h**) expression in 83 GSCs infected with vector control, ROR1 WT, or ROR1 mut lentivirus. Data are presented as mean \pm SEM (n = 3). * $P < 0.05$, *t*-test. **i-j**, RT-qPCR analysis of CREB targets ETV5 (**i**) and FBXW9 (**j**) expression in 131 GSCs infected with vector control, ROR1 WT, or ROR1 mut lentivirus. Data are presented as mean \pm SEM (n = 3). * $P < 0.05$, ** $P < 0.01$, *t*-test. **k**, IB analysis of pROR1, pHER2, pCREB, ROR1, HER2, and CREB in X01 GSCs infected with shCtrl, shIGFBP5-1 lentivirus, and then infected with vector control, ROR1 WT, or ROR1 mut lentivirus. GAPDH was used as a loading control. **l**, Invasion assays of X01 GSCs infected with shCtrl, shIGFBP5-1 lentivirus, and then infected with vector control, ROR1 WT, or ROR1 mut lentivirus. Images (left) taken after 24 h of invasion are representative of three independent experiments (scale bar, 100 μ m), and the graph (right) shows the mean number of invasive cells \pm SEM (n = 3). ** $P < 0.01$, *t*-test.

Response Fig. 3 (Revised Extended Data Fig. 6d-n; similar question from Reviewers 1 and 2). Overexpression of CREB enhances GSCs invasion. a-b, IB analysis of pCREB, and CREB in 83 (**a**) and in 131 (**b**) GSCs infected with vector control, or CREB OE lentivirus. GAPDH was used as a loading control. **c-d**, RT-qPCR analysis of CREB targets ETV5 (**c**) and FBXW9 (**d**) expression in 83 GSCs infected with vector control, CREB OE lentivirus. Data are presented as mean \pm SEM ($n = 3$). ** $P < 0.01$, t -test. **e-f**, RT-qPCR analysis of CREB targets ETV5 (**e**) and FBXW9 (**f**) expression in 131 GSCs infected with vector control, CREB OE lentivirus. Data are presented as mean \pm SEM ($n = 3$). * $P < 0.05$, ** $P < 0.01$, t -test. **g-h**, Invasion assays of 83 (**g**) and 131 (**h**) GSCs infected with vector control, or CREB OE lentivirus. Images (left) taken after 48 h of invasion are representative of three independent experiments (scale bar, 100 μ m), and the graph (right) shows the mean number of invasive cells \pm SEM ($n = 3$). * $P < 0.05$, t -test. **i**, IB analysis of pROR1, pHER2, pCREB, ROR1, HER2, and CREB in X01 GSCs infected with shCtrl, shIGFBP5-1 lentivirus, and then infected with vector control, CREB OE lentivirus. GAPDH was used as a loading control. **j**, Invasion assays of X01 GSCs infected with shCtrl, shIGFBP5-1 lentivirus, and then infected with vector control, CREB OE lentivirus. Images (left) taken after 24 h of invasion are representative of three independent experiments (scale bar, 100 μ m), and the graph (right) shows the mean number of invasive cells \pm SEM ($n = 3$). ** $P < 0.01$, t -test.

6. Lines 193-194. “ ... HER2 knockdown did not alter ROR1 phosphorylation”, etc. It is difficult to be confident about this statement in the absence of a quantitative analysis of Fig. 4h. Phospho/total data should be analyzed for several repeat experiments.

Response: We thank the reviewer’s comment. As reviewer 1 suggested, we have replaced the siRNA of HER2 and ROR1 by shHER2 or shROR1 X01 GSCs, which showed higher efficacy than the siRNA transfected cell. And we quantified the immunoblot results of revised Fig.4k, l (Response Fig. 21). The phosphor/total ratios of CREB decreased in both shHER2 and shROR1 (Response Fig. 32) in X01 GSC. The phosphor/total ratios of HER2 decreased significantly in X01 infected with shROR1-1 or shROR1-2 lentivirus (Response Fig. 32, right). However, the phosphor/total ratios of ROR1 showed no significant difference in X01 infected with shCtrl and shHER2 lentivirus (Response Fig. 32, left),

indicating that ROR1 knockdown regulated HER2 phosphorylation, but HER2 knockdown did not alter ROR1 phosphorylation.

Response Fig. 21 (Revised Fig. 4k, l; similar question from Reviewer 2).

Response Fig. 32. Quantification of IB analysis in Fig. 4h. Bar graph for IB analysis to calculate the gray value of each band, shown as the ratio of phosphorylation to total CREB/HER2/ROR1 expression level from manuscript Fig.4h (left shHER2, right shROR1) using ImageJ software. Ratio of phosphorylation/total protein expression \pm SEM (n = 3). ** $P < 0.01$, n.s. $P > 0.05$, *t*-test.

7.Lines 332~333. Rather than introducing a new result in the Discussion, the authors should indicate the position of pIGF1R in Fig. 4a to illustrate that IGF1R phosphorylation is not promoted by IGFBP-5. Given that IGF1R phosphorylation may have occurred transiently and not been detected on this array, it would be informative to check the acute pIGF1R response to IGFBP-5 (e.g. a time-course starting at 5 or 10 min) by immunoblot and to ensure that the addition of an IGF1R tyrosine kinase inhibitor such as NVP-AEW541 did not block any downstream effects of IGFBP-5.

Response: We thank the reviewer for this constructive comment. We have labeled the position of pIGF1R in revised Fig.4a (Response Fig. 33), which showed very low expression of IGF1R tyrosine

phosphorylation in both Vehicle and rIGFBP5 treated GSCs. Moreover, immunoblot analysis showed no expression changed in pIGF1R expression with rIGFBP5 treatment for 6 h (Response Fig. 34a). Then we tested the short time treatment of rIGFBP5 into 83 and 131 GSCs for 5, 8 and 10 min, respectively, and examined the acute pIGF1R response of IGFBP5. The results showed that the treatment of rIGFBP5 did not affect IGF1R phosphorylation (Response Fig. 34b, c).

Response Fig. 33 (Revised Fig. 4a).

Response Fig. 34. IGF1R phosphorylation is not promoted by IGFBP5. a, IB analysis of p-IGF1R and IGF1R in 83 GSCs treated with rIGFBP5 (100 ng/ml) for 6 h, **b-c,** IB analysis of p-IGF1R and IGF1R in 83 (**b**) and 131 (**c**) non-invasive GSCs treated with rIGFBP5 (100 ng/ml) for 5, 8 and 10 min, respectively. GAPDH was used as a loading control.

Minor point

Line 112: "... adverse roles": I believe the authors mean "diverse" roles.

Response: We thank the reviewer for pointing this out. We have revised the "adverse" to "diverse" at page 6, line 13 of our revised manuscript.

Reviewer #4 (Remarks to the Author):

Lin et al. report a comprehensive study into the role of Insulin-like Growth Factor-Binding Protein 5

(IGFBP5) on glioblastoma infiltration in orthotopic mouse models, in particular, as it relates to GSCs. In a GSC-derived brain tumor model, they conducted RNA seq analysis and found that IGFBP5 was differentially expressed between invasive or non-invasive subtypes. The authors then engaged in detailed mechanistic studies to establish the importance of IGFBP5 for GSC invasion. Finally, they report data suggesting that the nanoparticle-based delivery of CRISPR/Cas9-based IGFBP5 gene editing results in impressive therapeutic benefits, such as significant increase in survival in a GBM mouse model.

Overall, this is a large research study with an impressive attention to details, in particular on the mechanistic aspects. The initial hypothesis related to IGFBP5 is motivated by a comparative study between invasive and non-invasive GBM subtypes. They follow up with both silencing and knock down experiments that clearly establish the link between IGFBP5 and GSC invasion. While this study has clearly merits – in particular as it relates to GBM pathobiology - there are also significant shortcomings that dampen the potential impact. The following aspects should be addressed prior to publication:

1) IGFBP5 is a member of the IGFBP family, and, contrary to what is suggested in the introduction and the discussion sections, a significant bulk of literature clearly links these proteins to GBM infiltration and poor clinical prognosis. The same is also true for IGFBP3, see for example (references not cited by authors): <https://www.ncbi.nlm.nih.gov/pmc/articles/PMC7885861/> or <https://celldiv.biomedcentral.com/articles/10.1186/s13008-020-00061-6>.

Response: We thank the reviewer for this comment. As suggested, we have cited more literatures into revised manuscript.

➤ Page 17 lines 16--17

“Three IGFBP isoforms, IGFBP2, IGFBP3, and IGFBP5, are reportedly overexpressed in clinical biopsies of high-grade diffuse glioma⁴⁰⁻⁴².”

2) There are fundamental limitations of using orthotopic GBM systems to model GBC invasion. For this particular study, genetically engineered spontaneous glioblastoma mouse models would be a better choice. The authors should at least discuss the limitations of the model they selected and how this could affect the scientific premise of their study.

Response: We thank the reviewer for this comment. Patient-derived xenograft models (PDXs) generated from GSC, which generally recapitulate both the genetic and histological profiles of donor patient-derived tumors, such as highly infiltrative, following white matter tracts and spreading over the corpus callosum^{1, 12-14}. As reviewer suggested the PDXs have some limitations, the major disadvantage

of the patient derived GSC xenograft model is that it can only be established in immunodeficient mice such as nude, NOD-SCID, or NOD-SCID-gamma mice. The immune system in these mice differs innately from that of the host; thus, current PDXs do not represent the host immune system.

We have discussed the limitations of our models in Discussion section of the revised manuscript as follows:

➤ Page 19 lines 22--24

“Although, the current PDXs model only can be established in immunodeficient mice such as nude, NOD-SCID, or NOD-SCID-gamma mice. However, these preclinical data offer a proof-of-concept for the treatment of invasive GBM by targeting IGFBP5.”

3) IGFBP5 is overexpressed in a wide range of tissues, and it is unclear how CRISPR/Cas9-based IGFBP5 gene editing would affect these tissues systematically, see, e.g., <https://www.proteinatlas.org/ENSG00000115461-IGFBP5/tissue>. This seems an important aspect that should be addressed by the authors.

Response: We appreciate the reviewer’s nice comment regarding the safety concerns. It is true that IGFBP5 is also expressed in other normal organs including liver and kidney. However, the Ang-SS-Cas9/sgIGFBP5 nanocapsules caused little off-target in these organs even though they accumulated in and excreted from the liver, kidney and spleen. This is mainly attributed to two reasons: (1) the Angiopep-2 decorated nanocapsules specifically target to the low-density lipoprotein-1 (LRP-1) over-expressed endothelial and tumor cells, while could hardly be taken by other normal cells that express much less LRP-1 receptor; (2) even little nanocapsules are taken by the normal cells, they could not induce IGFBP5 gene editing as the Cas9/sgIGFBP5 cannot be released from the nanocapsules due to the low intracellular reduction microenvironment in normal cells. Because the nanocapsules contain abundant disulfide bond which can be specifically cleaved in excessive glutathione tumor microenvironment, thusly triggering the disassemble of nanocapsules and fast Cas9/sgIGFBP5 release for effective gene knockout in tumor sites. The brain delivery system has been well-established in our lab, see more details in our previous work⁸ (*Science Advances*, 2022). Therefore, the selective cellular uptake and specific intracellular drug release in tumor cells could avoid the off targets in other normal tissues.

[REDACTED]

Editorial Note: see Figure 5 A-H of Yan Z., *et al. Sci. Adv.* 2022 8: eabm8011

Response Fig. 35. Mutation frequencies of off-target sites (tumor, normal brain tissue, liver and kidney) in (a-d) U87MG and (e-h) CSC2 GSCs tumor-bearing mice treated with ANCSS(Cas9/sgPLK1) (1.5 mg Cas9 equiv./kg). Each value was determined from a single deep-sequencing library prepared from genomic DNA.

4) The nanoparticle delivery system provides clear benefits in survival, but a more complete characterization would be warranted. What is the biodistribution of the nanoparticles? Where are they accumulating? If they go into liver and spleen, is there any toxicity or off-target activity? In particular, it would be interesting to know if they accumulate in any of the tissues that overexpress IFGBP5.

Response: Thank for the reviewer for this constructive comments. (1) The nanocapsule delivery system was similarly with our recently published work⁸. Detailed characterization of the nanocapsules was presented in that paper, the results showed that the targeting nanocapsules achieved approximately 11.8%ID/g (injected dose per gram) in tumor site in both immune-free and immune-competent tumor mice models (Response Fig. 36). Meanwhile, high accumulation was also observed in the main organs including liver, kidney and spleen, indicating they may be excreted from these tissues that are in line with reported results¹⁵⁻¹⁸.

(2) Indeed, abundant nanocapsules were accumulated in kidney and spleen, while non-significant toxicity was observed in these tissues from the histological analysis (Response Fig. 37). That is mainly attributed to the specific and selective tumor cell uptake and intracellular drug release. Moreover, we have also performed the blood analysis for the nanocapsules, the results showed that these nanocapsules displayed no obvious side effects as compared with PBS treatment (Response Fig. 38). Collectively, these data suggest that the targeted nanocapsules possess good biocompatibility.

(3) Yes, the nanocapsules also accumulated in normal tissues (kidney and liver) that expressed IGFBP5, while they may induce non-specific gene editing in these tissues due to the relative low cell uptake and Cas9/sgIGFBP5 release (refer to comment 3).

[REDACTED]

Editorial Note: see Figure 2G and Supplementary Figure 9B of Yan Z., *et al. Sci. Adv.* 2022 8: eabm8011

Response Fig. 36. Biodistribution quantitation of orthotopic X01-Luc GSCs tumor-bearing nude mice following treatment with Ang-SS-Cas9/sgRNA or FreeCas9/sgRNA. Mice were intravenously injected at a dose of 1.5 mg Cas9 equiv./kg. Biodistribution quantitation of AF647-Cas9 accumulation in different organs taken from (a) immune free and (b) immune competent GBM mice after treated with nanocapsules or free Cas9/sgRNA. AF647-Cas9 levels determined by fluorescence spectroscopy are expressed as % ID/g. Data are mean \pm SD (n = 3, ** $P < 0.01$).

[REDACTED]

Editorial Note: see Supplementary Figure 30 of Yan Z., *et al. Sci. Adv.* 2022 8: eabm8011

Response Fig. 37. Tissue histology of orthotopic X01-Luc human glioblastoma tumor-bearing nude mice following treatment with Ang-SS-Cas9/sgRNA, Ang-SS-Cas9/sgScramble or PBS. Mice were intravenously injected at a dose of 1.5 mg Cas9 equiv./kg on day 10, 12, 14, 16, and 18 post tumor implantations. H&E analysis of the main tissues (heart, liver, spleen, lung and kidney) after treated with the nanocapsules and PBS.

Response Fig. 38. Analysis of hematological and biochemical parameters. Blood parameter analysis of healthy BALB/c mice treated with Ang-SS-Cas9/sgIGFBP5 or PBS at 0, 1, 3, 7 or 14 d after nanocapsule injection. Data are presented as mean \pm SD (n = 5). Albumin (ALB), Alkaline Phosphatase (ALP), Plasma Alanine Aminotransferase (ALT), Aspartate Aminotransferase (AST), Creatinine (CREA), Platelet (PLT), Red Blood Cell (RBC), White Blood Cell (WBC).

5) The CRISPR/Cas9-based IGFBP5 gene editing experiments have only been done in the X1 subtype, but at least one non-invasive subtype should be included for comparison.

Response: As suggested by the reviewer, we have carried out the anti-tumor evaluation in a non-invasive 83 GSCs mice model. The treatment course was similarly with that in X01 GSCs. The tumor bioluminescence of mice treated with Ang-SS-Cas9/sgIGFBP5 was increased dramatically, which is comparable to that with PBS and non-specific sequence control Ang-SS-Cas9/sgScr (Response Fig. 6a). Furthermore, the body weight of all the three groups were reduced during the treatment period (Response Fig. 6b). Importantly, the nanocapsules Ang-SS-Cas9/sgIGFBP5 did not prolong the survival time of the mice bearing 83 GSCs (Response Fig. 6c). All these results demonstrate that Ang-SS-Cas9/sgIGFBP5 was ineffective in a non-invasive mice model, supporting that the IGFBP5 plays a key role in the invasion of GSCs.

Accordingly, we have added several sentences in our revised manuscript as follows:

➤ Page 15 lines 19--23

“To further confirm the specificity of IGFBP5 regulatory role for invasive GSCs, we performed Ang-SS-Cas9/sgIGFBP5 in a non-invasive 83-Luc (stable luciferase-expressing 83 GSCs) GSC-bearing mice model. The results showed the identical outcomes of tumor growth and mouse survival among Ang-SS-Cas9/sgIGFBP5, Ang-SS-Cas9/sgScramble and PBS (Extended Data Fig. 9a-c).”

Response Fig. 6 (New Extended Data Fig. 9; similar question from Reviewer 1). Effect of Cas9/sgIGFBP5 nanocapsules on tumorigenesis in 83-Luc GSCs-bearing mice. a, Luminescence images of orthotopic 83-Luc GSCs-bearing mice following treatment with Ang-SS-Cas9/sgIGFBP5, Ang-SS-Cas9/sgScramble or PBS. Mice were intravenously injected at a dose of 1.5 mg Cas9 equiv./kg on day 10, 12, 14, 16, and 18 post tumor implantations. **b,** Body weight changes in mice following different treatments. **c,** Kaplan-Meier survival curves of mice implanted with 1×10^4 83-Luc GSCs and treated with Ang-SS-Cas9/sgIGFBP5, Ang-SS-Cas9/sgScramble or PBS ($n = 7$). ns $P > 0.05$, log-rank test.

6) What was the dose used for the CRISPR/Cas9-based IGFBP5 gene editing experiments? A dose

escalation study should be included. What is the fraction of the total dose delivered to the brain tumor? How much was delivered to the brain? What is the GBM/brain ratio?

Response: We appreciate the reviewer's constructive comment. The dosage is 1.5 mg Cas9 equiv./kg for the anti-tumor evaluation of the nanocapsules. The mice treated with the nanocapsules showed negligible side effects as observed from both the histological and blood analysis. Thusly, we deduce that these nanocapsules have good biocompatibility *in vivo*. Considering the high expense of Cas9 protein, we did not carry out the dose escalation study, but the constitutions of the nanocapsules are non-toxic materials, including Cas9/sgRNA complex, polyethylene glycol (PEG), angiopep-2 peptide, acrylate guanidine (AG) and cysteamine bisacrylamide (CBA).

The injected dosage of the nanocapsules is 1.5 mg Cas9 equiv./kg, which contains approximately 30 µg Cas9, 36 µg sgRNA, 990 µg Angiopep-2 decorated PEG, 9 µg AG and 9 µg CBA. More than 92% of the nanocapsules are based on peptide and PEG, which have excellent biocompatibility.

As calculated from the biodistribution analysis, more than 11.8% ID/g Cas9/sgRNA was delivered into the tumor, which was 11.5-fold higher than that into the normal brain (1.03% ID/g). Therefore, the ratio of GBM/brain is 11.5, indicating the specific tumor targeting capability of these nanocapsules.

7) The authors claim for their CRISPR-Cas9 system “a much preferable outcome on tumor growth as well as mouse expectancy, comparing to the well-known siRNA-based therapies targeting the conventional GBM targets (i.e., PLK134 or STAT3)”. However, when consulting their cited references, it appears as the opposite is the case, i.e., their survival results, also significant, don't match up with the one achieved by the other therapies.

Response: We thank the reviewer for pointing out this. As mentioned that siRNA-based therapies in GBM models^{17, 18} also showed significant survival results, the reason we mentioned as CRISPR-Cas9 system “a much preferable outcome on tumor growth...” is that we compared the survival curve differences of our Cas9/sgIGFBP5 with siRNA targeted PLK1 or STAT3. Targeting PKL1 by siRNA improved the survival days of GBM mice model from about 28 days to 52 days. Although the conventional GBM therapy of targeting STAT3 SNPs + IR prominent improved of survival curve of GBM models, the mono-targeting STAT3 by siRNA only showed around 5 day-difference in survival length¹⁷. However, targeting IGFBP5 by CRISPR-Cas9 system in our Fig. 7m promoted survival of around 60-day in GBM mice models.

Additionally, we also performed IGFBP5 siRNA therapeutic efficacy towards CSC-2 GSC orthotopic xenograft model. Although *in vivo* nanocapsule-mediated siIGFBP5 knockdown significantly increased mouse survival (Response Fig. 6f), the improvement of median survival time was less than 10 days.

Taken together, our results and related references suggest that Cas9/sgIGFBP5 nano-therapy is a more potent method for GBM targeting therapy.

Response Fig. 7. Nanocapsule-mediated delivery of siIGFBP5 suppresses GSCs invasion. **a-b**, RT-qPCR analysis of IGFBP5 (**a**) and ETV5 (**b**) expression in CSC2 GSCs treated with Ang-SS-siIGFBP5 nanocapsules (1 ng/ml) or vehicle control for 72 h. Data are represented as mean \pm SEM ($n = 3$). ** $P < 0.01$, t -test. **c-d**, Invasion assay in CSC2 GSCs treated with Ang-SS-siIGFBP5 nanocapsules or vehicle control for 48 h. (**c**) Images are representative of three independent experiments (scale bar, 100 μm), and (**d**) the graph shows the mean number of invasive cells \pm SEM ($n = 3$). ** $P < 0.01$, t -test. **e**, H&E staining of the whole brain and IHC analysis of pHER2, pROR1, and pCREB in orthotopic xenografts of CSC2 GSCs treated with Ang-SS-siIGFBP5 nanocapsules every other day for 10 days (scale bar, 100 μm). **f**, Kaplan-Meier survival curves of mice implanted with 1×10^4 CSC2-Luc cells and treated with Ang-SS-siIGFBP5 nanocapsules (1 mg/kg) or vehicle control ($n = 5$). $P = 0.0035$, log-rank test.

References

1. Jeongwu L., *et al.* Tumor stem cells derived from glioblastomas cultured in bFGF and EGF more closely mirror the phenotype and genotype of primary tumors than do serum-cultured cell lines. *Cancer Cell* **9**: 391-403 (2006). (Revised manuscript refer 11)
2. Naoki O., *et al.* VEGF promotes tumorigenesis and angiogenesis of human glioblastoma stem cells. *Biochem. Biophys. Res. Commun.* **360**: 553-559 (2007). (Revised manuscript refer 68)
3. Kyeung M. J., *et al.* Patient-Specific Orthotopic Glioblastoma Xenograft Models Recapitulate the Histopathology and Biology of Human Glioblastomas In Situ. *Cell Rep.* **3**: 260-273 (2013). (Revised manuscript refer 69)
4. Eunhee K., *et al.* Phosphorylation of EZH2 activates STAT3 signaling via STAT3 methylation and promotes tumorigenicity of glioblastoma stem-like cells. *Cancer Cell* **23**: 839-852 (2013). (Revised manuscript refer 70)
5. Ping M., *et al.* Mesenchymal glioma stem cells are maintained by activated glycolytic metabolism involving aldehyde dehydrogenase 1A. *PNAS* **110**: 8647-8649 (2013). (Revised manuscript refer 13)
6. Frank B. F., *et al.* Malignant astrocytic glioma: Genetics, biology, and paths to treatment. *Genes & Dev.* **21**: 2683-2710 (2017).
7. Chengyuan D., *et al.* IGFBP5 increases cell invasion and inhibits cell proliferation by EMT and Akt signaling pathway in Glioblastoma multiforme cells. *Cell Div.* **15**: 1-9 (2020). (Revised manuscript refer 42)
8. Yan Z., *et al.* Blood-brain barrier–penetrating single CRISPR-Cas9 nanocapsules for effective and safe glioblastoma gene therapy. *Sci. Adv.* **8**: eabm8011 (2022). (Revised manuscript refer 38)
9. Vincent D., *et al.* Intracellular levels and secretion of insulin-like-growth-factor-binding proteins in MCF-7/6, MCF-7/AZ and MDA-MB-231 breast cancer cells Differential modulation by estrogens in serum-free medium. *Eur. J. Biochem.* **232**: 47-53 (1995). (Revised manuscript refer 25)
10. Han., *et al.* The expression of insulin-like growth factor (IGF) and IGF-binding protein (IGFBP) genes in the human placenta and membranes: evidence for IGF-IGFBP interactions at the fetomaternal interface. *J. Clin. Endocrinol. Metab.* **81**: 2680-2693 (1996). (Revised manuscript refer 26)
11. Huamin W., *et al.* Overexpression of IGFBP5, but not IGFBP3, Correlates with the Histologic Grade of Human Diffuse Glioma: A Tissue Microarray and Immunohistochemical Study. *Technol. Cancer Res. Treat.* **5**: 195-199 (2006). (Revised manuscript refer 41)

12. Peter C. H., *et al.* In vivo models of primary brain tumors: Pitfalls and perspectives. *Neuro. Oncol.* **14**: 979-993 (2012).
13. Hiroaki W., *et al.* Maintenance of primary tumor phenotype and genotype in glioblastoma stem cells. *Neuro. Oncol.* **14**: 13-44 (2012). (Revised manuscript refer 14)
14. Ruihui C., *et al.* A hierarchy of self-renewing tumor-initiating cell types in glioblastoma. *Cancer Cell* **17**: 362-375 (2010).
15. Bing W., *et al.* Acc. Metabolism of Nanomaterials in Vivo: Blood Circulation and Organ Clearance. *Chem. Res.* **46**: 761-769 (2013).
16. Bujie D., *et al.* Transport and interactions of nanoparticles in the kidneys. *Nat. Rev. Mater.* **3**: 358-374 (2018).
17. Gregory, J.V., *et al.* Systemic brain tumor delivery of synthetic protein nanoparticles for glioblastoma therapy. *Nat. Commun.* **11**: 5687 (2020). (Revised manuscript refer 67)
18. Zou, Y., *et al.* Single siRNA Nanocapsules for Effective siRNA Brain Delivery and Glioblastoma Treatment. *Adv. Mater.* **32**: 2000416 (2020). (Revised manuscript refer 37)

REVIEWER COMMENTS

Reviewer #1 (Remarks to the Author):

The authors have sufficiently addressed the major mechanistic and functional concerns raised in my previous review. I disagree with their continued use of the "Glioma Stem Cell" designation for a GBM cell cultured in neurosphere conditions. These are simply conditions that may enrich for putative glioma stem cells, and cannot on its own define the population as being exclusively comprised of GSCs.

Reviewer #2 (Remarks to the Author):

The new data provided by the authors have addressed my concerns.

Reviewer #3 (Remarks to the Author):

The authors are to be commended for the extensive improvements made to this interesting manuscript. The question of possible modulation of IGF1R signaling by IGFBP-5 still remains to be answered (points 2 and 3 below)..

Specific points

1. The quantitative analysis of Fig. 4c (response Fig. 30) should be added to the paper, e.g. as supplementary data. Similarly for some other quantitative summaries of immunoblot data that are missing from the manuscript.
2. Given the role of IGFBP-5 in modulating IGF-1 and IGF-2 signaling through IGF1R, the authors need to address IGF1R activation in the paper. As noted in my first review, the failure to detect phospho-IGF1R in the phospho-array (Fig 4a) provides no information, and in fact the very strong phospho-IGF1R signals in Response Fig. 34 contradict the absence of phospho-IGF1R signal on the phospho-array. Can the authors suggest an explanation for this discrepancy?
3. It is unusual to see such a strong constitutive phospho-IGF1R signal (Response Fig. 34), described in the non-invasive lines. Is there a high concentration of IGF-1 in the medium? Phospho-IGF1R should also be examined in the invasive cell lines. And most importantly (as in my first review), the authors should determine how IGF1R tyrosine kinase inhibition (which should abolish the strong phospho-IGF1R signal) affects the signaling responses attributed to IGFBP-5 in these cells. There should be a brief discussion of IGF1R-dependent and independent effects of IGFBP-5.

Reviewer #4 (Remarks to the Author):

The authors have clearly addressed the majority of the reviewers' comments. I would recommend publishing the manuscript in its current form.

<Responses to reviewers' comments>

Reviewer #1 (Remarks to the Author)

The authors have sufficiently addressed the major mechanistic and functional concerns raised in my previous review. I disagree with their continued use of the "Glioma Stem Cell" designation for a GBM cell cultured in neurosphere conditions. These are simply conditions that may enrich for putative glioma stem cells, and cannot on its own define the population as being exclusively comprised of GSCs.

Response: We thank the reviewer for this comment. There is no doubt that the culture condition of stem cells needs to be improved. However, the current neurosphere culture condition (DMEM/F12 serum-free supplemented with EGF, bFGF, and B27) is utilized as one of the most common culture conditions for glioma stem cell studies not only in our team but also in many other laboratories¹⁻⁹. Furthermore, previous studies determined that numerous stemness-associated genes were upregulated under the neurosphere culture condition, which promoted maintaining the criteria of stem cells both *in vitro* and *in vivo* compared with the FBS culture condition^{1,2}. The four cell lines (X01, 448, 83, and 131) we used in this study were identified as GSCs with upregulation of stemness markers (Nestin, CD44 or CD133) by Oka et al.; Joo et al.; Kim et al. and Lee et al. in different studies^{1, 3-5}. Additionally, in our results, we also showed the detection of stemness marker Nestin (Response Fig. 8a, b, and Response Fig. 9b) in X01, 448, 83, 131, 772 and stemness marker CD133 in 772 (Response Fig. 9a) in protein or mRNA levels. Taken together, we believe that the cells we used in this study under neurosphere culture condition, could be termed glioblastoma stem cells.

Reviewer #2 (Remarks to the Author)

The new data provided by the authors have addressed my concerns.

Response: We again appreciate the kindness of the reviewer in helping improve our manuscript.

Reviewer #3 (Remarks to the Author)

The authors are to be commended for the extensive improvements made to this interesting manuscript. The question of possible modulation of IGF1R signaling by IGFBP-5 still remains to be answered (points 2 and 3 below)..

Specific points

1. The quantitative analysis of Fig. 4c (response Fig. 30) should be added to the paper, e.g. as

supplementary data. Similarly for some other quantitative summaries of immunoblot data that are missing from the manuscript.

Response: We thank the reviewer for this comment. We have added the relative quantitative analysis into the revised Extended Data Fig. 4a-d with accompanying description (Page 8, line 15—Page 10, line 23) and Extended Data Figure Legend 4 (Page 1, line 21—Page 2, line 20) in our revised manuscript.

Revised Extended Data Fig. 4.

2. Given the role of IGFBP-5 in modulating IGF-1 and IGF-2 signaling through IGF1R, the authors need to address IGF1R activation in the paper. As noted in my first review, the failure to detect phospho-IGF1R in the phospho-array (Fig 4a) provides no information, and in fact the very strong phospho-IGF1R signals in Response Fig. 34 contradict the absence of phospho-IGF1R signal on the phospho-array. Can the authors suggest an explanation for this discrepancy?

Response: We thank the reviewer for this kind comment. We would like to explain this as below: First, we observed no IGF1R phosphorylation in 83 GSCs treated with human rIGFBP5 protein using the human phospho-RTK array (R&D system, ARY001B), which allows screening 49 different phosphorylated RTKs at the same time. Compared to the pHER2 and pROR1, the expression of pIGF1R was very weak, resulting in high difficulties in detecting the expression of pIGF1R in the same human phospho-RTK array. Second, we couldn't use the same antibody in the verification experiment. We contacted the technical support call center of R&D system and asked if they can offer the same antibodies in the human phospho-RTK antibody array (R&D system, ARY001B), but unfortunately, they cannot provide the same RTK antibody products. Under this situation, we had to choose other antibodies (Sangon Biotech: IGF1R, D155189; pIGF1R, D155037), to confirm if IGF1R phosphorylation responses to rIGFBP5 treatment, following the reviewer's suggestion. Given that the antibodies used in immunoblot analysis (Response Fig. 34a) were different from the human phospho-RTK array (Fig. 4a), the sensitivities of different antibodies may cause showing unequal expressions of pIGF1R. This can be considered as another reason for pIGF1R expressional differences in human phospho-RTK antibody array and immunoblot analysis. We hope the reviewer will understand this situation and that our explanation can dispel your concerns.

3. It is unusual to see such a strong constitutive phospho-IGF1R signal (Response Fig. 34), described in the non-invasive lines. Is there a high concentration of IGF-1 in the medium? Phospho-IGF1R should also be examined in the invasive cell lines. And most importantly (as in my first review), the authors should determine how IGF1R tyrosine kinase inhibition (which should abolish the strong phospho-IGF1R signal) affects the signaling responses attributed to IGFBP-5 in these cells. There should be a brief discussion of IGF1R-dependent and independent effects of IGFBP-5.

Response: We thank the reviewer for the constructive comment. First, we confirmed that the culture media used for culturing 4 GSCs X01, 448, 83, and 131 is a DMEM/F12 medium supplemented with EGF, bFGF, B27, but no serum addition. Following the components information of DMEM/F12 (WELGENE, Cat. LM002-04), there is no IGF1 or IGF2.

Second, as the reviewer suggested, we performed immunoblot analysis to detect the expression of pIGF1R in both invasive GSCs (X01 and 448) and non-invasive GSCs (83 and 131). Results showed higher basic expression levels of pIGF1R in non-invasive GSC 83 compared to GSCs X01, 448, and 131 (New Response Fig. 1), suggesting that 83 GSCs might have an upregulation of autocrine IGF1 or IGF2 signaling.

Third, following the suggestion by the reviewer, we further examined the expression of pIGF1R, pROR1, pHER2, pCREB, IGF1R, ROR1, HER2, CREB, and IGFBP5 in X01 GSCs after treatment with IGF1R tyrosine kinase inhibitor (NVP-AEW541) for 1h or 3h (New Response Fig. 2). The immunoblot analysis indicates that the inhibition of IGF1R signaling by NVP-AEW541 does not affect the expression of pROR1, pHER2, and pCREB in GSCs, either. In breast and prostate cancers, IGFBP5-stimulated IGF receptor signaling was determined to implicate cancer cell survival or migration^{10, 11, 14}. On the other hand, IGF-independent IGFBP5 showed inhibitory, proapoptotic effects on cell growth in head and neck squamous cell carcinoma, melanoma, and breast cancer cells¹⁰⁻¹⁵, through another signaling (p38/AKT, p38/MAPK). In this study, our results suggest that IGFBP5 triggers IGF-independent oncogenic signaling in GSC invasion.

New Response Fig. 1. IGF1R and pIGF1R expression in 131, 83, X01 and 448 GSCs. IB analysis of IGF1R and pIGF1R in X01, 448, 83, 131 GSCs. GAPDH was used as a loading control.

Response Fig. 2. IB analysis of pIGF1R, pROR1, pHER2, pCREB, IGF1R, ROR1, HER2, CREB, and IGFBP5 in X01 GSCs after 1h and 3h treatment with 10 μ M NVP-AEW541. GAPDH was used as a loading control.

Reviewer #4 (Remarks to the Author)

The authors have clearly addressed the majority of the reviewers' comments. I would recommend publishing the manuscript in its current form.

Response: We thank the reviewer for acknowledging our efforts.

References

1. Lee J., *et al.* Tumor stem cells derived from glioblastomas cultured in bFGF and EGF more closely mirror the phenotype and genotype of primary tumors than do serum-cultured cell lines. *Cancer Cell* **9**: 391-403 (2006). (Revised manuscript refer 11)
2. Ahmad M., *et al.* How stemlike are sphere cultures from Long-term cancer cell lines? Lessons from mouse glioma models. *J. Neuropathol. Exp. Neurol.* **17**: 1062-1077 (2014).
3. Oka N., *et al.* VEGF promotes tumorigenesis and angiogenesis of human glioblastoma stem cells. *Biochem. Biophys. Res. Commun.* **360**: 553-559 (2007). (Revised manuscript refer 68)
4. Joo K. M., *et al.* Patient-specific orthotopic glioblastoma xenograft models recapitulate the histopathology and biology of human glioblastomas in situ. *Cell Rep.* **3**: 260-273 (2013). (Revised manuscript refer 69)
5. Kim E., *et al.* Phosphorylation of EZH2 activates STAT3 signaling via STAT3 methylation and promotes tumorigenicity of glioblastoma stem-like cells. *Cancer Cell* **23**: 839-852 (2013). (Revised manuscript refer 70)
6. Mao P., *et al.* Mesenchymal glioma stem cells are maintained by activated glycolytic metabolism involving aldehyde dehydrogenase 1A. *PNAS* **110**: 8647-8649 (2013). (Revised manuscript refer 13)
7. Galli R., *et al.* Isolation and characterization of tumorigenic, stem-like neural precursors from human glioblastoma. *Cancer Res.* **64**: 7011-7021 (2004).
8. Pollard S. M., *et al.* Glioma stem cell lines expanded in adherent culture have tumor-specific phenotypes and are suitable for chemical and genetic screens. *Cell Stem Cell* **4**: 568-580 (2009).
9. Zhang L., *et al.* The necessity for standardization of glioma stem cell culture: a systematic review. *Stem Cell Res. & Ther.* **13**: 470 (2022).
10. Dittmer J., *et al.* Biological effects and regulation of IGFBP5 in breast cancer. *Front. Endocrinol.* **13**: 983796 (2022).
11. Beattie J., *et al.* Insulin-like growth factor-binding protein-5 (IGFBP-5): a critical member of the IGF axis. *Biochem. J.* **395**: 1-19 (2007). (Revised manuscript refer 46)
12. Ding M., *et al.* Secreted IGFBP5 mediates mTOC1-dependent feedback inhibition of IGF-1 signaling. *Nat. Cell Biol.* **18**: 319-327 (2016).
13. Tripathi G., *et al.* IGF-independent effects of insulin-like growth factor binding protein (Igfbp5) in vivo. *FASEB J.* **23**: 2616-2626 (2009).
14. Güllü G., *et al.* Functional roles and clinical values of insulin-like growth factor-binding protein-5 in different types of cancers. *Chin. J. Cancer* **31**: 266-280 (2012).
15. Wang J., *et al.* Insulin-like growth factor-binding protein-5 (IGFBP5) functions as a tumor suppressor

in human melanoma cells. *Oncotarget* **6**: 20636-20649 (2015).

Reviewer #1 comments:

While I appreciate the authors' response to my concerns regarding the use of the term "glioblastoma stem cell (GSC)", it is recognized in the field that GSCs cannot be defined by any specific molecular marker (e.g., Nestin, CD133, etc) or simply the ability to grow in the neurosphere culture media. Conversely, the most accepted literature states that a GSC is defined as "the ability to generate a tumor upon intracranial implantation that recapitulates the cellular heterogeneity present in the parent tumor" (Lathia et. al, Genes and Development, 2015). They further state "while the ability to grow as spheres is also evident of [GSCs], it is not by default the defining feature of a self-renewing population of cells". A more recent review from Dr. Jeremy Rich, perhaps the most recognized investigator in the GSC space, concurs with the previous review where he states GSCs "occupy a functionally defined state characterized by self-renewal and tumour-initiating capacity with inherent plasticity that is agnostic to cell of origin, frequency, proliferation rate or specific molecular markers"(Gimple et. al., Nature Reviews Cancer 2022).

Therefore, according to these criteria above, if the authors are set on using this term, I would suggest additional experiments including limiting dilution in vivo transplantation assays as well as an evaluation of the regeneration of the cellular heterogeneity of the parent tumor.

Reviewer #3 comments:

The authors have provided some useful additional data to the reviewers in this reponse and the previous one, but they have not made this information available to the reader. In this reviewer's opinion an examination of the possible involvement of IGF1R modulation in the observed effects of IGFBP-5 is integral to understanding how IGFBP-5 works in these cells, since IGF binding and IGF1R modulation are canonical roles of IGFBP-5, and the authors accept that at least some GSC lines "might have an upregulation of autocrine IGF1 or IGF2 signaling". Therefore the relevant data should be included in the paper so readers can understand the authors' reasoning.

In all versions of this manuscript, the first mention of IGF1R occurs in the Discussion on page 18, even though this is clearly a relevant result ("we observed no IGF1R phosphorylation in GSCs treated with human rIGFBP5 protein") and therefore should be included in the Results section.

Suggested specific remedies:

1. The authors have revealed that their phospho-RTK array lacks the necessary sensitivity to detect pIGF1R in their cells, since Response Fig. 43a (to which I no longer have access) showed a strong pIGF1R signal when the array showed no signal. The Results section of this paper should include brief data to demonstrate their finding of "no IGF1R phosphorylation in GSCs treated with human rIGFBP5 protein" and this data should use an antibody capable of detecting pIGF1R (as in Response Fig. 43a).
2. The paper should also include in the Results section the data shown in New Response Figure 2 – a demonstration that IGF1R blockade has no apparent effect on the phosphorylation responses of the pROR1, pHER2, and pCREB. Ideally this result would be shown in cells both untreated and treated with IGFBP-5 stimulation, since the point of the experiment is to determine whether there is IGF1R involvement in the response to IGFBP-5, not in the basal phosphorylation state.

<Responses to reviewers' comments>

Reviewer #1 comments:

While I appreciate the authors' response to my concerns regarding the use of the term "glioblastoma stem cell (GSC)", it is recognized in the field that GSCs cannot be defined by any specific molecular marker (e.g., Nestin, CD133, etc) or simply the ability to grow in the neurosphere culture media. Conversely, the most accepted literature states that a GSC is defined as "the ability to generate a tumor upon intracranial implantation that recapitulates the cellular heterogeneity present in the parent tumor" (Lathia et. al, Genes and Development, 2015). They further state "while the ability to grow as spheres is also evident of [GSCs], it is not by default the defining feature of a self-renewing population of cells". A more recent review from Dr. Jeremy Rich, perhaps the most recognized investigator in the GSC space, concurs with the previous review where he states GSCs "occupy a functionally defined state characterized by self-renewal and tumour-initiating capacity with inherent plasticity that is agnostic to cell of origin, frequency, proliferation rate or specific molecular markers"(Gimple et. al., Nature Reviews Cancer 2022). Therefore, according to these criteria above, if the authors are set on using this term, I would suggest additional experiments including limiting dilution in vivo transplantation assays as well as an evaluation of the regeneration of the cellular heterogeneity of the parent tumor.

Response: As suggested by the reviewer, we would like to change the term glioblastoma stem cell (GSC) to glioblastoma stem-like cell (GSC). The related statements were revised in the manuscript and supplemental information. To further respond to the reviewer's concerns about glioblastoma stem cells, we would like to provide more stem-like associated evidence for X01, 448, 83, and 131 cells and some new data for 772 cells as following:

First: The cells we chose were already published as glioma stem cells or glioblastoma stem-like cells (GSCs) in several previous studies in different journals¹⁻⁸. In these previous papers, the four cell lines (X01, 448, 83, and 131) were determined as GSCs with the upregulation of stemness markers (Nestin, CD44, or CD133), tumor initiation, differentiation and self-renewal abilities¹⁻⁴. Dr. Naoki Oka and Dr. Kyeong Min Joo demonstrated that orthotopic xenograft tumors from X01, and 448 were recapitulating the pheno-copy of original GBM's extensive infiltrative capacity¹⁻², which matched with our infiltrative phenotype classification of GSCs. In recent years, more following publications started working on various mechanisms of GBM initiation and progression by using GSCs (X01, 448, 83, and 131)⁹⁻¹⁴ to understand the detailed molecular mechanism of tumor infiltration and cancer stemness. Our team also showed positive data for stemness marker detection, limiting dilution analysis, and differentiation ability of 448, X01, and 83 cells⁵⁻

⁸, but still, we need more proof to describe these cells as glioma stem cells. However, the characteristics of these cells are well mimicking the infiltrative phenotype of patient-derived glioblastoma cells, we can use these cells as a model system to prove our new findings *in vitro* and *in vivo*. Therefore, we will exchange the term GSC (glioblastoma stem cell) for glioblastoma stem-like cell.

Then: To give more evidence for the stem-like features of 772 GSCs, we detected the stemness marker CD133 and Nestin expression in mRNA level (Second Revision Response Fig. 1), also determined the differentiation marker and self-renewal ability of 772 cells by RT-PCR and *in vitro* limiting dilution assay. The sphere forming ability of 772 extremely decreased, and the astrocyte marker GFAP induced expression in FBS culture condition (Second Revision Response Fig. 2). Although we did not perform the *in vivo* transplantation assay at limiting dilution, we produced 772 tumor model with 1×10^5 and 3×10^5 cells/mouse. The median survival time of 1×10^5 772 cells transplanted mice was around 5 months (Original Fig. 6f), but 3×10^5 772 cells injected mice was around 3 months (Second Revision Response Fig. 3c). The MRI image showed tumor growth after 10 or 19 weeks of 772 cells transplantation (Second Revision Response Fig. 3a, b). Moreover, the cellular heterogeneity of GSCs (448, X01, 83, 131, and 772) was also determined by RNA-Seq analysis of our original manuscript, showing as all the GSCs we used exhibited different subtypes of GBM (Revised manuscript Fig. 6b; Extended Data Fig. 1b-d). But still, we need more proof to describe 772 as glioblastoma stem cells. So, we would like to classify 772 as a glioblastoma stem-like cell.

Last: Given it is not realistic to perform the *in vivo* transplantation assay at limiting dilution due to the restriction from COVID-19 in our place, we summarized the cell numbers of tumor formation in the previously published studies (current study included) instead. The xenograft mouse tumors were performed by injecting different cell numbers listed as 1×10^5 and 2×10^5 cells/mouse of GSC 488^{2, 7}; 1×10^4 , 1×10^5 and 2×10^5 cells/mouse of GSC X01^{1, 7-10}; 1×10^3 , 1×10^4 and 1×10^5 cells/mouse of 83 GSC^{4, 6, 8, 11, 12}; and 5×10^4 , 1×10^5 and 5×10^5 cells/mouse of GSC 131^{3, 13, 14}. The numbers of transplanted GSC cells in the xenograft model are comparable to or less than the other glioblastoma stem cell-related studies¹⁵⁻¹⁷, but we agree with the reviewer's point about 'glioblastoma stem cells' definition. Thus, we will change our concept for GSC (glioblastoma stem cell) to glioblastoma stem-like cell.

Collectively, we have revised the term 'glioblastoma stem cell (GSC)' to 'glioblastoma stem-like cell (GSC)' in our revised manuscript, which is based on the evidence of stem-like natures we stated above. Please see page 2, line 3, and line 18; page 3, line 9 in our revised manuscript.

Second Revision Response Fig. 1 (Response Fig. 9 in 1st revision). Stemness marker expression in 772 GSCs. a-b, RT-qPCR analysis of CD133 (a) and Nestin (b) mRNA expression in 772 GSCs and astrocyte cell HA1800. Data are presented as mean \pm SEM ($n = 3$). ** $P < 0.01$, t -test.

Second Revision Response Fig. 2. Differentiation and self-renewal capability of 772 GSCs. a-b, RT-qPCR analysis of Nestin, CD133 (a) and GFAP (b) mRNA expression in 772 cells with CSC full medium (GSC) or serum contained medium 48 h (DIFF). Data are presented as mean \pm SEM ($n = 3$). ** $P < 0.01$, t -test. **c**, Limiting dilution analysis (LDA) performed using 772 cells with GSC or DIFF medium, calculated the sphere formation numbers at day 6 after seeding. ** $P < 0.01$, t -test.

Second Revision Response Fig. 3. 772 GSCs orthotopic xenografts transplanted with different cell numbers. **a**, Magnetic resonance imaging (MRI) of mice bearing orthotopic xenografts of patient-derived 772 GSCs with 3×10^5 (left) and 1×10^5 (right) cells/mouse after 10 weeks of cell injection ($n = 3$). **b**, MRI of mice bearing orthotopic xenografts of patient-derived 772 GSCs with 1×10^5 cells/mouse after 19 weeks ($n = 3$). **c**, Kaplan-Meier survival curves of mice implanted with 3×10^5 772 GSCs ($n = 9$).

Reviewer #3 comments:

The authors have provided some useful additional data to the reviewers in this response and the previous one, but they have not made this information available to the reader. In this reviewer's opinion an examination of the possible involvement of IGF1R modulation in the observed effects of IGFBP-5 is integral to understanding how IGFBP-5 works in these cells, since IGF binding and IGF1R modulation are canonical roles of IGFBP-5, and the authors accept that at least some GSC lines "might have an upregulation of autocrine IGF1 or IGF2 signaling". Therefore the relevant data should be included in the paper so readers can understand the authors' reasoning.

In all versions of this manuscript, the first mention of IGF1R occurs in the Discussion on page 18, even though this is clearly a relevant result ("we observed no IGF1R phosphorylation in GSCs treated with human rIGFBP5 protein") and therefore should be included in the Results section.

Suggested specific remedies:

1. The authors have revealed that their phospho-RTK array lacks the necessary sensitivity to detect pIGF1R in their cells, since Response Fig. 43a (to which I no longer have access) showed a strong pIGF1R signal when the array showed no signal. The Results section of this paper should include brief data to demonstrate their finding of "no IGF1R phosphorylation in GSCs treated with human rIGFBP5 protein" and this data should use an antibody capable of detecting pIGF1R (as in Response Fig. 43a).

Response: Based on the statement of "no IGF1R phosphorylation in GSCs treated with human rIGFBP5 protein(as in Response Fig. 43a)" from the reviewer, but we only listed 38 Response Figures in our first rebuttal letter. So, we realize that the reviewer may request us to include the result of Response Fig. 34a in the Results section of this paper. Thus, we added Response Fig. 34a into our revised manuscript as Extended Data Fig. 10a. Please see the Response part below for further statement.

2. The paper should also include in the Results section the data shown in New Response Figure 2 – a demonstration that IGF1R blockade has no apparent effect on the phosphorylation responses of the pROR1, pHER2, and pCREB. Ideally this result would be shown in cells both untreated and treated with

IGFBP-5 stimulation, since the point of the experiment is to determine whether there is IGF1R involvement in the response to IGFBP-5, not in the basal phosphorylation state.

Response: We thank the reviewer for this comment. As suggested, we added Response Fig. 34a and the new IB analysis result of comment 2 into our revised manuscript as new Extended Data Fig. 10a and b. Please see page 18, lines 4—9 in our revised manuscript and page 5, lines 13—18 in our Extended Data information. Finally, we appreciated the reviewer’s valuable and insightful comments and suggestions during the revision process.

Second Revision Response Fig. 4 (New Extended Data Fig. 10). IGF1R does not regulate the IGFBP5 mediated signaling axis in GSCs. a, IB analysis of pIGF1R and IGF1R in 83 GSCs treated with rIGFBP5 (100 ng/ml) for 6 h. **b,** IB analysis of pIGF1R, pROR1, pHER2, pCREB, IGF1R, ROR1, HER2 and CREB in 83 GSCs with IGF1R blockade (10 μ M NVP-AEW541, 3 h) under IGFBP5 stimulation (100ng/ml rIGFBP5, 6 h). GAPDH was used as the loading control.

References

1. Naoki O., *et al.* VEGF promotes tumorigenesis and angiogenesis of human glioblastoma stem cells. *Biochem. Biophys. Res. Commun.* **360**: 553-559 (2007). (revised manuscript refer 69)
2. Kyeung M. J., *et al.* Patient-Specific Orthotopic Glioblastoma Xenograft Models Recapitulate the Histopathology and Biology of Human Glioblastomas In Situ. *Cell Rep.* **3**: 260-273 (2013). (revised manuscript refer 70)
3. Eunhee K., *et al.* Phosphorylation of EZH2 activates STAT3 signaling via STAT3 methylation and promotes tumorigenicity of glioblastoma stem-like cells. *Cancer Cell* **23**: 839-852 (2013). (revised manuscript refer 71)

4. Ping M., *et al.* Mesenchymal glioma stem cells are maintained by activated glycolytic metabolism involving aldehyde dehydrogenase 1A3. *PNAS* **110**: 8647-8649 (2013). (revised manuscript refer 13)
5. Yin, J., *et al.* ARS2/MAGL signaling in glioblastoma stem cells promotes self-renewal and M2-like polarization of tumor-associated macrophages. *Nat Commun.* **11**, 2978 (2020). (revised manuscript refer 7)
6. Yin, J., *et al.* Transglutaminase 2 Inhibition Reverses Mesenchymal Transdifferentiation of Glioma Stem Cells by Regulating C/EBP β Signaling. *Cancer Res.* **77**, 4973-4984 (2017). (revised manuscript refer 8)
7. Jun-hee, H., *et al.* Modulation of Nogo receptor I expression orchestrates myelin-associated infiltration of glioblastoma. *Brain* **144**, 636-654 (2021).
8. Yin, J., *et al.* Pigment Epithelium-Derived Factor (PEDF) Expression Induced by EGFRvIII Promotes Self-renewal and Tumor Progression of Glioma Stem Cells. *PLoS Biol* **13**, e1002152 (2015). (revised manuscript refer 12)
9. Akio, S., *et al.* Epidermal Growth Factor Plays a Crucial Role in Mitogenic Regulation of Human Brain Tumor Stem Cells. *J. Biol. Chem.* **283**, 10958-10966 (2008).
10. Akio, S., *et al.* The Evidence of Glioblastoma Heterogeneity. *Sci. Rep.* **27**, 7979 (2015).
11. Delphine, G., *et al.* Divergent evolution of temozolomide resistance in glioblastoma stem cells is reflected in extracellular vesicles and coupled with radiosensitization. *Neuro. Oncol.* **20**, 236-248 (2017).
12. Hacer, G., *et al.* Impairment of Glioma Stem Cell Survival and Growth by a Novel Inhibitor for Survivin-Ran Protein Complex. *Clin. Cancer. Res.* **19**, 613-642 (2013).
13. Jin-Ku, L., *et al.* USP1 targeting impedes GBM growth by inhibiting stem cell maintenance and radioresistance. *Neuro. Oncol.* **18**, 37-47 (2016).
14. Jason K. S., *et al.* *In vivo* RNAi screen identifies NLK as a negative regulator of mesenchymal activity in glioblastoma. *Oncotarget* **6**, 20145-20159 (2015).
15. Weiwei, T., *et al.* Dual role of WISP1 in maintaining glioma stem cells and tumor-supportive macrophages in glioblastoma. *Nat Commun.* **11**, 3015 (2020).
16. Zhizhong L., *et al.* Hypoxia-inducible factors regulate tumorigenic capacity of glioma stem cells. *Cancer Cell* **15**: 501-513 (2009).
17. Bao, S., *et al.* Glioma stem cells promote radioresistance by preferential activation of the DNA damage response. *Nature* **444**, 756-760 (2006). (revised manuscript refer 6)

18. Jeongwu L., *et al.* Tumor stem cells derived from glioblastomas cultured in bFGF and EGF more closely mirror the phenotype and genotype of primary tumors than do serum-cultured cell lines. *Cancer Cell* **9**: 391-403 (2006). (revised manuscript refer 11)

REVIEWERS' COMMENTS

Reviewer #1 (Remarks to the Author):

The authors have adequately addressed my concerns.

Reviewer #3 (Remarks to the Author):

Thanks to the authors for including sufficient data to indicate that IGFBP-5 appears to stimulate pROR1, pHER2, and pCREB independently of IGF1R activation. This is critical to the conclusion that IGFBP-5 acts IGF- and IGF1R-independently, and I am still unclear why the authors have not included it in the Results section. However this is not a reason to further delay publication of this interesting paper.

<Responses to reviewers' comments>

Reviewer #1 (Remarks to the Author)

The authors have adequately addressed my concerns.

Response: We again appreciate the kindness of the reviewer in helping improve our manuscript.

Reviewer #3 (Remarks to the Author)

Thanks to the authors for including sufficient data to indicate that IGFBP-5 appears to stimulate pROR1, pHER2, and pCREB independently of IGF1R activation. This is critical to the conclusion that IGFBP-5 acts IGF- and IGF1R-independently, and I am still unclear why the authors have not included it in the Results section. However this is not a reason to further delay publication of this interesting paper.

Response: We thank the reviewer for acknowledging our efforts, his/her suggestions have remarkably improved our manuscript.